**Integrating faults and past earthquakes into a probabilistic seismic hazard**
**model for peninsular Italy**
Alessandro Valentini[1], Francesco Visini[2] and Bruno Pace[1]
[1] DiSPUTer, Università degli Studi "Gabriele d'Annunzio", Chieti, Italy
[2] Istituto Nazionale di Geofisica e Vulcanologia, L'Aquila, Italy
**Abstract**
*Italy is one of the most seismically active countries in Europe. Moderate to strong earthquakes, with*
*magnitudes of up to ~7, have been historically recorded for many active faults. Currently,*
*probabilistic seismic hazard assessments in Italy are mainly based on area source models, in which*
*seismicity is modelled using a number of seismotectonic zones and the occurrence of earthquakes is*
*assumed uniform. However, in the past decade, efforts have increasingly been directed towards using*
*fault sources in seismic hazard models to obtain more detailed and potentially more realistic patterns*
*of ground motion. In our model, we used two categories of earthquake sources. The first involves*
*active faults, and geological slip rates were used to quantify the seismic activity rate. We produced an*
*inventory of all fault sources with details of their geometric, kinematic and energetic properties. The*
*associated parameters were used to compute the total seismic moment rate of each fault. We*
*evaluated the magnitude-frequency distribution (MFD) of each fault source using two models: a*
*characteristic Gaussian model centred at the maximum magnitude and a Truncated Gutenberg-*
*Richter model. The second earthquake source category involves grid-point seismicity, with a fixed-*
*radius smoothed approach and a historical catalogue were used to evaluate seismic activity. Under*
*the assumption that deformation is concentrated along faults, we combined the MFD derived from the*
*geometry and slip rates of active faults with the MFD from the spatially smoothed earthquake sources*
*and assumed that the smoothed seismic activity in the vicinity of an active fault gradually decreases*
*by a fault size-driven factor. Additionally, we computed horizontal peak ground acceleration maps for*
*return periods of 475 and 2,475 yrs. Although the ranges and gross spatial distributions of the*
*expected accelerations obtained here are comparable to those obtained through methods involving*
*seismic catalogues and classical zonation models, the spatial pattern of the hazard maps obtained*
*with our model is far more detailed. Our model is characterized by areas that are more hazardous*
*and that correspond to mapped active faults, while previous models yield expected accelerations that*
*are almost uniformly distributed across large regions. In addition, we conducted sensitivity tests to*

*determine the impact on the hazard results of the earthquake rates derived from two MFD models for faults and to determine the relative contributions of faults versus distributed seismic activity. We believe that our model represents advancements in terms of the input data (quantity and quality) and methodology used in the field of fault-based regional seismic hazard modelling in Italy.*

## 1. Introduction

In this paper, we present the results of an alternative seismogenic source model for use in a probabilistic seismic hazard assessment (PSHA) for Italy that integrates active fault and seismological data. The use of active faults as an input for seismic hazard analysis is a consolidated approach in many countries characterized by high strain rates and seismic releases, as shown, for example, by Field et al. (2015) in California and Stirling et al. (2012) in New Zealand. Moreover, in recent years, active fault data have also been successfully integrated into seismic hazard studies or models, in regions with moderate-to-low strain rates, such as SE Spain (e.g., Garcia-Mayordomo et al., 2007), France (e.g., Scotti et al., 2014), and central Italy (e.g., Peruzza et al., 2011).

In Europe, a working group of the European Seismological Commission, named *Fault2SHA*, is discussing fault-based seismic hazard modelling (https://sites.google.com/site/linkingfaultpsha/home). The working group, born to motivate exchanges between field geologists, fault modellers and seismic hazard practitioners, and it is a community initiative with long term vision on studying the active faults. The work we are presenting here stems from the activities of the *Fault2SHA* working group.

Combining active faults and background sources is one of the key aspects in this type of approach. Although the methodology remains far from identifying a standard procedure, common approaches combine active faults and background sources by applying a threshold magnitude, generally between 5.5 and 7, above which seismicity is modelled as occurring on faults and below which seismicity is modelled via a smoothed approach (e.g., Akinci et al., 2009; Danciu et al., 2017), area sources (e.g., the so-called FSBG model in the 2013 European Seismic Hazard Model, ESHM13; Woessner et al., 2015) or a combination of the two (Field et al., 2015; Pace et al., 2006).

Another important aspect in the use of active faults to build a seismogenic source
model is the use of an appropriate MFD to characterize the temporal model
describing the seismic activity of faults. Gutenberg-Richter (GR) and characteristic
earthquake models are commonly used, and the choice sometimes depends on the
knowledge of the fault and data availability. Often, the choice of the "appropriate"
MFD for each fault source is a difficult task because palaeoseismological studies are
scarce, and it is often difficult to establish clear relationships between mapped faults
and historical seismicity. Recently, Field et al. (2017) discussed the effects and
complexity of the choice, highlighting how often the GR model results are not
consistent with data; however, in other cases, uncharacteristic behaviour, with rates
smaller than the maximum, are possible. The discussion is open (see for example
the discussion by Kagan et al., 2012) and far from being solved with the available
observations, including both seismological and/or geological/paleoseismological
observations. In this work, we explore the calculations of these two MFD, a
characteristic Gaussian model and a Truncated Gutenberg-Richter model, to explore
the epistemic uncertainties and to consider a *Mixed model* as a so-called "expert
judgment" model. This *Mixed model* approach, in which we assigned one of the two
MFDs to each fault source, is useful for comparison analysis. The rationale of the
choice of the MFD of each fault source is explained in detail later in this paper.
However, this approach obviously does not solve this issue, that can be treated as
epistemic uncertainties using logic tree or random sampling but, in any case, the
choice of MFD remains an open question in fault-based PSHA.
In Italy, the current national PSH model for building code (Stucchi et al., 2011) is
based on area sources and the classical Cornell approach (Cornell, 1968), in which
the occurrence of earthquakes is assumed uniform in the defined seismotectonic
zones. However, we believe that more efforts must be directed towards using
geological data (e.g., fault sources and paleoseismological information) in PSHA to
use slip-rates that describe longer seismic cycles to match the larger magnitudes,
extend the observational time required to capture the recurrence of large-magnitude
events and therefore improve the reliability of seismic hazard assessments. In fact,
as highlighted by the 2016-2017 seismic sequences in central Italy, a zone-based
source model is not able to model local spatial variations in ground motion (Meletti et
al., 2016), whereas a fault-based model can provide insights for aftershock time-
dependent hazard analysis (Peruzza et al., 2016). In conclusion, even if the main
purpose of this work is to integrate active faults into hazard calculations for the Italian
territory, this study does not represent an official update of the seismic hazard model
of Italy.

**2. Source Inputs**
Two earthquake-source inputs are considered in this work. The first is a fault source
input that is based on active faults and uses the geometries and slip rates of known
active faults to compute activity rates over a certain range of magnitudes. The
second is a classical smoothed approach that accounts for the rates of expected
earthquakes with a minimum moment magnitude (Mw) of 4.5 but excludes
earthquakes associated with known faults based on a modified earthquake
catalogue. Note that our seismogenic source requires the combination of the two
source inputs related to the locations of expected seismicity rates into a single
source model. Therefore, these two earthquake-source inputs are not independent
but complementary, in both the magnitude and frequency distribution, and together
account for spatial and temporal distribution of the seismicity in Italy.
In the following subsections, we describe the two source inputs and how they are
combined in the seismogenic source model.
**2.1 Fault Source Input**
In seismic hazard assessment, an active fault is a structure that exhibits evidence of
activity in the late Quaternary, has a demonstrable or potential capability of
generating major earthquakes and is capable of future reactivation (e.g. Machette,
2000, Danciu et al., 2017). The evidence of Quaternary activity can be
geomorphological and/or paleoseismological when activation information from
instrumental seismic sequences and/or association to historical earthquakes is not
available. Fault source data and location are useful for seismic hazard studies, and
we compiled a database for Italy via the analysis and synthesis of neotectonic and
seismotectonic data from approximately 90 published studies of 110 faults across
Italy. Our database included, but was not limited to, the Database of Individual
Seismogenic Sources (DISS vers. 3.2.0, http://diss.rm.ingv.it/diss/), which is already
available for Italy. It is important to highlight that the DISS is currently composed of
two main categories of seismogenic sources: individual and composite sources. The
latter are defined by the DISS' authors as "*simplified and three-dimensional*
*representation of a crustal fault containing an unspecified number of seismogenic*
*sources that cannot be singled out. Composite seismogenic sources are not*
*associated with a specific set of earthquakes or earthquake distribution*", and
therefore are not useful for our PSHA approach; the former is "*a simplified and three-*
*dimensional representation of a rectangular fault plane. Individual seismogenic*
*sources are assumed to exhibit characteristic behaviour with respect to rupture*
*length/width and expected magnitude*" (http://diss.rm.ingv.it/diss/index.php/about/13-
introduction). Even if in agreement with our approach, we note that some of the
individual seismogenic sources in the DISS are based on geological and
paleoseismological information, and many others used the *Boxer* code (Gasperini et
al., 1999) to calculate the epicentre, moment magnitude, size and orientation of a
seismic source from observed macroseismic intensities. We carefully analysed the
individual sources and some related issues: (i) the lack of updating of the geological
information of some individual sources and (ii) the nonconformity between the input
data used by DISS in *Boxer* and the latest historical seismicity (CPTI15) and
macroseismic intensity (DBMI15) publications. Thus, we performed a full review of
the fault database. We then compiled a fault source database as a synthesis of
works published over the past twenty years, including DISS, using all updated and
available geological, paleoseismological and seismological data (see the
supplemental files for a complete list of references). We consider our database as
complete as possible in terms of individual seismogenic sources, and it contains all
the parameters necessary to construct an input dataset for fault-based PSHA.
The resulting database of normal and strike-slip active and seismogenic faults in
peninsular Italy (Fig. 1, Tables 1 and 2; see the supplemental files) includes all the
available geometric, kinematic, slip rate and earthquake source-related information.
In the case of missing data regarding the geometric parameters of dip and rake, we
assumed typical dip and rake values of 60° and -90°, respectively, for normal faults
and 90° and 0° or 180°, respectively, for strike-slip faults. In this paper, only normal
and strike-slip faults are used as fault source inputs. We decided not to include thrust
faults in the present study because, with the methodology proposed in this study (as
discussed later in the text), the maximum size of a single-rupture segment must be
defined, and segmentation criteria have not been established for large thrust zones.
Moreover, our method uses long-term geological slip rates to derive active seismicity
rates, and sufficient knowledge of these values is not available for thrust faults in
Italy. Because some areas of Italy, such as the NW sector of the Alps, Po Valley, the
offshore sector of the central Adriatic Sea, and SW Sicily, may be excluded by this
limitation, we are considering an update to our approach to include thrust faults and
volcanic sources in a future study. The upper and lower boundaries of the
seismogenic layer are mainly derived from the analysis of Stucchi et al. (2011) of the
Italian national seismic hazard model and locally refined by more detailed studies
(Boncio et al., 2011; Peruzza et al., 2011; Ferranti et al., 2014).
Based on the compiled database, we explored three main aspects associated with
defining a fault source input: the slip rate evaluation, the segmentation model and
the expected seismicity rate calculation.
*2.1.1 Slip rates*
Slip rates control fault-based seismic hazards (Main, 1996, Roberts et al., 2004; Bull
et al., 2006; Visini and Pace, 2014) and reflect the velocities of the mechanisms that
operate during continental deformation (e.g., Cowie et al., 2005). Moreover, long-
term observations of faults in various tectonic contexts have shown that slip rates
vary in space and time (e.g., Bull et al., 2006; Nicol et al., 2006, 2010, McClymont et
al., 2009a-b; Gunderson et al., 2013; Benedetti et al., 2013, D'Amato et al., 2016),
and numerical simulations (e.g., Robinson et al., 2009; Cowie et al., 2012; Visini and
Pace, 2014) suggest that variability mainly occurs in response to interactions
between adjacent faults. Therefore, understanding the temporal variability in fault slip
rates is a key point in understanding the earthquake recurrence rates and their
variability.
To evaluate the minimum and maximum slip rates, that we assumed representatives
of the long-term slip rate variability over time, we used slip rates determined in
different ways and at different time scales (e.g., at the decadal scale based on
geodetic data or at longer scales based on the displacement of Holocene or Plio-
Pleistocene horizons). These values were derived from approximately 65 available
neotectonics, palaeoseismology and seismotectonics papers (see the supplemental
files). In this work, we used the mean of the minimum and maximum slip rate values
listed in Table 1 and assumed that they are representative of the long-term
behaviour (over the past 15 ky in the Apennines). Because a direct comparison of
slip rates over different time intervals obtained by different methods may be
misleading (Nicol et al., 2009), we cannot exclude the possibility that uncertainties
and errors compilation could affect the original data in some cases. The discussion
of these possible biases and their evaluation via statistically derived approaches
(e.g., Gardner et al., 1987; Finnegan et al., 2014; Gallen et al., 2015) is beyond the
scope of this paper and will be explored in future work. Moreover, we are assuming
that slip rate values used are representative of seismic movements, and aseismic
factors are not taken into account. Therefore, we believe that investigating the effect
of this assumption could be another issue explored in future work; for example, by
differentiating between aseismic slip factors in different tectonic contexts.
Because 28 faults had no measured slip (or throw) rate (Fig. 1a), we proposed a
statistically derived approach to assign a slip rate to these faults. Based on the slip
rate spatial distribution shown in Figure 1b, we subdivided the fault database into
three large regions–the Northern Apennines, Central-Southern Apennines and
Calabria-Sicilian coast–and analysed the slip rate distribution in these three areas.
Figure 1b indicates that the slip rates tend to increase from north to south. The fault
slip rates in the Northern Apennines range from 0.3 to 0.8 mm/yr, with the most
common values ranging from approximately 0.5-0.6 mm/yr; the slip rates in the
Central-Southern Apennines range from 0.3 to 1.0, and the most common rate is
approximately 0.3 mm/yr; and the slip rates in the southern area (Calabria and Sicily)
range from 0.9 to 1.8, with the most common being approximately 0.9 mm/yr.
Keeping in mind that average and minimum-maximum range of slip rate represents
two different aspects of the slip rate behaviour of a fault (average long-term and its
variability), we analysed them independently to assign values to active faults without
measures.
The first step in assigning an average slip rate and a range of variability to the faults
with unknown values is to identify the most representative distribution among known
probability density functions using the slip rate data from each of the three areas. We
test five well-known probability density functions (*Weibull*, *normal*, *exponential*,
*Inverse Gaussian* and *gamma*) against mean slip rate observations. The resulting
function with the highest log-likelihood is the *normal* function in all three areas. Thus,
the mean value of the *normal* distribution is assigned to the faults with unknown
values. We assign a value of 0.58 mm/yr to faults in the northern area, 0.64 mm/yr to
faults in the Central-Southern area, and 1.10 mm/yr to faults in the Calabria-Sicilian
area. To assign a range of slip rate variability to each of the three areas, we test the
same probability density functions against slip rate variability observations. Similar to
the mean slip rate, the probability density function with the highest log-likelihood is
the *normal* function in all three areas. We assign a variability of 0.25 mm/yr to the
faults in the northern area, 0.29 mm/yr to the faults in the Central-Southern area, and
0.35 mm/yr to the faults in the Calabria-Sicilian area.

*2.1.2 Segmentation rules for delineating fault sources*
An important issue in the definition of a fault source input is the formulation of
segmentation rules. In fact, the question of whether structural segment boundaries
along multisegment active faults act as persistent barriers to a single rupture is
critical to defining the maximum seismogenic potential of fault sources. In our case,
the rationale behind the definition of a fault source is based on the assumption that
the geometric and kinematic features of a fault source are expressions of its
seismogenic potential and that its dimensions are compatible for hosting major (Mw
≥ 5.5) earthquakes. Therefore, a fault source may consist of a fault or an ensemble
of faults that slip together during an individual major earthquake. A fault source is
defined by a *seismogenic master fault* and its surface projection (Fig. 2a).
*Seismogenic master faults* are separated from each other by first-order structural or
geometrical complexities. Following the suggestions by Boncio et al. (2004) and
Field et al. (2015), we imposed the following segmentation rules in our case study: (i)
4-km fault gaps among aligned structures; (ii) intersections with cross structures
(often transfer faults) extending 4 km along strike and oriented at nearly right angles
to the intersecting faults; (iii) overlapping or underlapping en echelon arrangements
with separations between faults of 4 km; (iv) bending ≥ 60° for more than 4 km; (v)
average slip rate variability along a strike greater than or equal to 50%; and (vi)
changes in seismogenic thickness greater than 5 km among aligned structures.
Example applications of the above rules are illustrated in Figure 2a.
By applying the above rules to our fault database, the 110 faults yielded 86 fault
sources: 9 strike-slip sources and 77 normal-slip sources. The longest fault source is
*Castelluccio dei Sauri* (fault number (*id in Table 1)* 42, L = 93.2 km), and the shortest
is *Castrovillari* (*id* 63, L = 10.3 km). The mean length is 30 km. The dip angle varies
from 30° to 90°, and 70% of the fault sources have dip angles between 50° and 60°.
The mean value of seismogenic thickness (ST) is approximately 12 km. The source
with the largest ST is *Mattinata* (*id* 41, ST = 25 km), and the source with the thinnest
ST is *Monte Santa Maria Tiberina* (*id* 9, ST = 2.5 km). This low value is due to the
presence of an east-dipping low angle normal fault, the Alto-Tiberina Fault (Boncio et
al., 2000), located a few kilometres west of the Monte Santa Maria Tiberina fault.
Maximum observed moment magnitude values (MObs) have been assigned to 35
fault sources (based on Table 2), and the values vary from 5.90 to 7.32. The fault
source inputs are shown in Figure 3.

*2.1.3 Expected seismicity rates*
Each fault source is characterized by data, such as kinematic, geometry and slip rate
information, that we use as inputs for the FiSH code (Pace et al., 2016) to calculate
the global budget of the seismic moment rate allowed by the structure. This
calculation is based on predefined size-magnitude relationships in terms of the
maximum magnitude (Mmax) and the associated mean recurrence time (Tmean).
Table 1 summarizes the geometric parameters used as FiSH input parameters for
each fault source (seismogenic box) shown in Figure 3. To evaluate Mmax of each
source, according to Pace et al., (2016) we first computed and then combined up to
five Mmax estimates (see the example of the Paganica fault source in Fig. 2b, details
in Pace et al., 2016). Specifically, these five Mmax estimates are as follows: MM0
based on the calculated scalar seismic moment (M0) and the application of the
standard formula Mw = 2/3 (logM0 – 9.1) (Hanks and Kanamori, 1979; IASPEI,
2005); two magnitude estimates using the Wells and Coppersmith (1994) empirical
relationships for the maximum subsurface rupture length (MRLD) and maximum
rupture area (MRA); a estimate that corresponds to the MObs, if available; and a
estimate (MASP, ASP for aspect ratio) computed by reducing the fault length input if
the aspect ratio (W/L) is smaller than the value evaluated by the relation between the
aspect ratio and rupture length of observed earthquake ruptures, as derived by
Peruzza and Pace (2002) (not in the case of Paganica in Fig. 2b). In some cases,
the use of MObs it was useful to better constrain the seismogenic potential of
individual seismogenic sources. For this reason and to take into account Mobs in the
estimation of Mmax, for each source we (i) calculated the maximum expected
magnitude (Mmax1) and the relative uncertainties using only the scaling
relationships and (ii) compared the maximum of observed magnitudes of the
earthquakes potentially associated with the fault. If MObs was within the range of
Mmax ± 1 standard deviation, we considered the value and recalculated a new
Mmax (Mmax2) with a new uncertainty. If MObs was larger than Mmax1 + 1
standard deviation, we reviewed the fault geometry and/or the earthquake-source
association. Conversely, if Mobs was lower then Mmax1 - 1 standard deviation we
considered a GR behaviour for that source, without using the Mobs in the Mmax2
calculation  As an example, for the Irpinia Fault (id 51 in Tables 1 and 2), the
characteristics of the 1980 earthquake (Mw~6.9) can be used to evaluate Mmax via
comparison with the Mmax derived from scaling relationships.
Because all the empirical relationships, as well as observed historical and recent
magnitudes of earthquakes, are affected by uncertainties, the *MomentBalance* (MB)
function of the FiSH code (Pace et al., 2016) was used to account for these
uncertainties. MB computes a probability density function (PDF) for each magnitude
derived from empirical relationships or observations and summarizes the results as a
maximum magnitude value with a standard deviation. The uncertainties in the
empirical scaling relationship, in FiSH, are taken from the studies of Wells and
Coppersmith (1994), Peruzza and Pace (2002) and Leonard (2010). Currently, the
uncertainty in magnitude associated with the seismic moment is fixed and set to 0.3,
whereas the catalogue defines the uncertainty in MObs. Moreover, to combine the
evaluated maximum magnitudes, MB creates a probability curve for each magnitude
by assuming a normal distribution (Fig. 2). We assumed a two-sides untruncated
normal distribution of magnitudes. MB subsequently sums the probability density
curves and fits the summed curve to a normal distribution to obtain the mean of the
maximum magnitude $M_{max}$ and its standard deviation.
Thus, a unique $M_{max}$ with a standard deviation is computed for each source, and this
value represents the maximum rupture that is allowed by the fault geometry and the
rheological properties.
Finally, to obtain the mean recurrence time of $M_{max}$ (i.e., $T_{mean}$), we use the criterion
of "segment seismic moment conservation" proposed by Field et al. (1999). This
criterion divides the seismic moment that corresponds to $M_{max}$ by the moment rate
for given a slip rate:
$$T_{mean} = \frac{1}{Char\_Rate} = \frac{10^{(1.5\,Mmax+\,9.1)}}{\mu VLW} \ (1)$$

where $T_{mean}$ is the mean recurrence time in years, *Char_Rate* is the annual mean
rate of occurrence, $M_{max}$ is the computed mean maximum magnitude, µ is the shear
modulus, V is the average long-term slip rate, and L and W are along-strike rupture
length and downdip width, respectively. Finally, we calculated the seismic moment
rate corresponding to $M_{max}$ and the MFDs of expected seismicity. For each fault
source, we use two "end-member" MFD models: (i) a *Characteristic Gaussian* (*CHG*)
model, a symmetric Gaussian curve (applied to the incremental MFD values) centred
on the $M_{max}$ value of each fault with a range of magnitudes equal to 1-sigma, and (ii)
a *Truncated Gutenberg-Richter* (*TGR*, Ordaz, 1999; Kagan, 2002) model, with $M_{max}$
as the upper threshold and $M_w$ = 5.5 as the minimum threshold for all sources. We
consider a constant b-value equal to 1.0 for all faults, as single-source events are
insufficient for calculating the required statistics; this value corresponds to the mean
b-value determined from the CPTI15 catalogue. The a-values were computed with
the ActivityRate tool of the FiSH code. ActivityRate calculated activity rates at
magnitudes given by each MFD, balancing the total MFD expected seismic moment
rate with the seismic moment rate that was obtained based on $M_{max}$ and $T_{mean}$
(details in Field et al., 1999; Field et al., 2015; Pace et al., 2016; Woessner et al.,
2015). In Figure 2c, we show an example of the expected seismicity rates in terms of
the annual cumulative rates for the Paganica source using the two above-described
MFD models.
Finally, we create a so-called "expert judgement" model, called the *Mixed* model, to
determine the MFD for each fault source based on the earthquake-source
associations. In this case, we decided that if an earthquake assigned to a fault
source (see Table 2 for earthquake-source associations) has a magnitude lower than
the magnitude range in the curve of the *CHG* model distribution, the *TGR* model is
applied to that fault source. Otherwise, the *CHG* model, which peaks at the
calculated $M_{max}$, is applied. We decided to not use a logic tree for every fault to
capture the model options because one of the aims of this work is to compare the
different MFD choices in terms of results and impact in the hazard curves. Of course,
errors in this approach can originate from the misallocation of historical earthquakes,
and we cannot exclude the possibility that potentially active faults responsible for
historical earthquakes have not yet been mapped. The MFD model assigned to each
fault source in our *Mixed* model is shown in Figure 3.

**2.2 Distributed Source Inputs**
Introducing distributed earthquakes into the seismogenic source model is necessary
because researchers have not been able to identify a causative source (i.e., a
mapped fault) for important earthquakes in the historical catalogue. This lack of
correlation between earthquakes and faults may be related to (i) interseismic strain
accumulation in areas between major faults, (ii) earthquakes occurring on unknown
or blind faults, (iii) earthquakes occurring on unmapped faults characterized by slip
rates lower than the rates of erosional processes, and/or (iv) the general lack of
surface ruptures associated with faults generating $M_w$ < 5.5 earthquakes.
We used the historical catalogue of earthquakes (CPTI15; Rovida et al., 2016; Fig.
4) to model the occurrence of moderate-to-large ($M$w ≥ 4.5) earthquakes. The
catalogue consists of 4,427 events and covers approximately the last one thousand
years from 01/01/1005 to 28/12/2014. Before using the catalogue, we removed all
events not considered mainshocks via a declustering filter (Gardner and Knopoff,
1974). This process resulted in a catalogue composed of 1,839 independent events,
which we denote as the "complete" catalogue. Moreover, to avoid double counting
due to the use of two seismicity sources, i.e., the fault sources and the distributed
seismicity sources, we removed events associated with known active faults from the
CPTI15 earthquake catalogue. If the causative fault of an earthquake is known, that
earthquake does not need to be included in the seismicity smoothing procedure. The
earthquake-source association is based on neotectonics, palaeoseismology and
seismotectonics papers (see the supplemental files) and, in a few cases,
macroseismic intensity maps. In Table 2, we listed the earthquakes with known
causative fault sources. The differences in the smoothed rates given by eq. (2) using
the complete and modified catalogues are shown in Figure 5.
We applied the standard methodology developed by Frankel (1995) to estimate the
density of seismicity in a grid with latitudinal and longitudinal spacing of 0.05°. The
smoothed rate of events in each cell *i* is determined as follows:
$$n_i = \frac{\Sigma_j n_j e^{\frac{-\Delta_{ij}^2}{c^2}}}{\Sigma_j e^{\frac{-\Delta_{ij}^2}{c^2}}} \qquad (2)$$

where $n_i$ is the cumulative rate of earthquakes with magnitudes greater than the
completeness magnitude Mc in each cell *i* of the grid and Δ*ij* is the distance between
the centres of grid cells *i* and *j*. The parameter *c* is the correlation distance. The sum
is calculated in cells *j* within a distance of 3*c* of cell *i*.
To compute earthquake rates, we adopted the completeness magnitude thresholds
over different periods given by Stucchi et al. (2011) for five large zones (Fig. 4).
To optimize the smoothing distance Δ in eq. (2), we divided the earthquake
catalogue into four 10-yr disjoint learning and target periods from the 1960s to the
1990s. For each pair of learning and target catalogues, we used the probability gain
per earthquake to find the optimal smoothing distance (Kagan and Knopoff, 1977;
Helmstetter et al., 2007). After assuming a spatially uniform earthquake density
model as a reference model, the probability gain per earthquake G of a candidate
model relative to a reference model is given by the following equation:
$$G = exp(\frac{L-L_0}{N}) \qquad (3)$$

where N is the number of events in the target catalogue and *L* and $L_0$ are the joint
log-likelihoods of the candidate model and reference model, respectively. Under the
assumption of a Poisson earthquake distribution, the joint log-likelihood of a model is
given as follows:

$$L = \sum_{i_x=1}^{N_x} \sum_{j_y=1}^{N_y} \log p \left[ \lambda(i_x, i_y), \omega \right] \quad (4)$$

where $p$ is the Poisson probability, $\lambda$ is the spatial density, $\omega$ is the number of observed events during the target period, and the parameters $i_x$ and $i_y$ denote each corresponding longitude-latitude cell.

Figure 6 shows that for the four different pairs of learning-target catalogues, the optimal smoothing distance $c$ (the mean curve) ranges from 25-40 km. Finally, the mean of all the probability gains per earthquake yields a maximum smoothing distance of 30 km (Fig. 6), which is then used in eq. (2).

The b-value of the GR distribution is calculated on a regional basis using the maximum-likelihood method of Weichert (1980), which allows multiple periods with varying completeness levels to be combined. Following the approach recently proposed by Kamer and Hiemer (2015), we used a penalized likelihood-based method for the spatial estimation of the GR b-values based on the Voronoi tessellation of space without tectonic dependency. The whole Italian territory has been divided into a grid with a longitude/latitude spacing of 0.05°, and the centres of the grid cells represent the possible centres of Voronoi polygons. We vary the number of Voronoi polygons, Nv, from 3 to 50, generating 1000 tessellations for each Nv. The summed log-likelihood of each obtained tessellation is compared with the log-likelihood given by the simplest model (prior model) obtained using the entire earthquake dataset. We find that 673 random realizations led to better performance than the prior model. Thus, we calculate an ensemble model using these 673 solutions, and the mean b-value of each grid node is shown in Figure 4.

The maximum magnitude $M_{max}$ assigned to each node of the grid, the nodal planes and the depths have been taken from ESHM13 (Woessner et al., 2015). The ESHM13 project evaluated the maximum magnitudes of large areas of Europe based on a joint procedure involving historical observations and tectonic regionalization. We adopted the lowest value of the maximum magnitude distributions proposed by ESHM13, but evaluating the impact of different maximum magnitudes is beyond the scope of this work.

Finally, the rates of expected seismicity for each node of the grid are assumed to
follow the TGR model (Kagan 2002):
$$\lambda(M) = \lambda_0 \frac{\exp(-\beta M) - \exp(-\beta M_u)}{\exp(-\beta M_0) - \exp(-\beta M_u)} \qquad (5)$$

where the magnitude (*M*) is in the range of $M_0$ (minimum magnitude) to $M_u$ (upper or
maximum magnitude); otherwise $\lambda(M)$ is 0. Additionally, $\lambda_0$ is the smoothed rate of
earthquakes at $M_w$ = 4.5 and $\beta$ = b ln(10).
**2.3 Combining Fault and Distributed Sources**
To combine the two source inputs, we introduced a distance-dependent linear
weighting function, such that the contribution from the distributed sources linearly
decreases from 1 to 0 with decreasing distance from the fault. The expected
seismicity rates of the distributed sources start at Mw = 4.5, which is lower than the
minimum magnitude of the fault sources, and the weighting function is only
applicable in the magnitude range overlapping the MFD of each fault. This weighting
function is based on the assumption that faults tend to modify the surrounding
deformation field (Fig. 7), and this assumption is explained in detail later in this
paper.
During fault system evolution, the increase in the size of a fault through linking with
other faults results in an increase in displacement that is proportional to the quantity
of strain accommodated by the fault (Kostrov, 1974). Under a constant regional
strain rate, the activity of fault sections arranged across strike must eventually
decrease (Nicol et al., 1997; Cowie, 1998; Roberts et al., 2004). Using an analogue
modelling, Mansfield and Cartwrigth (2001) showed that faults grow via cycles of
overlap, relay formation, breaching and linkage between neighbouring segments
across a wide range of scales. During the evolution of a system, the merging of
neighbour faults, mostly along strike, results in the formation of major faults, which
accommodate the majority of displacement. These major faults are surrounded by
minor faults, which accommodate lower amounts of displacement. To highlight the
spatial patterns of major and minor faults, Figures 7a and 7b present diagrams from
the Mansfield and Cartwright (2001) experiment in two different stages: the
approximate midpoint of the sequence and the end of the sequence. Numerical
modelling performed by Cowie et al. (1993) yielded similar evolutionary features for
major and minor faults. The numerical fault simulation of Cowie et al. (1993) was
able to reproduce the development of a normal fault system from the early nucleation
stage, including interactions with adjacent faults, to full linkage and the formation of a
large thoroughgoing fault. The model also captures the increase in the displacement
rate of a large linked fault. In Figures 7c and 7d, we focus on two stages of the
simulation (from Cowie et al., 1993): the stage in which the fault segments have
formed and some have become linked and the final stage of the simulation.
Notably, the spatial distributions of major and minor faults are very similar in the
experiments of both Mansfield and Cartwrigth (2001) and Cowie et al. (1993), as
shown in Figures 7a-d. Developments during the early stage of major fault formation
appear to control the location and evolution of future faults, with some areas where
no major faults develop. The long-term evolution of a fault system is the
consequence of the progressive cumulative effects of the slip history, i.e.,
earthquake occurrence, of each fault. Large earthquakes are generally thought to
produce static and dynamic stress changes in the surrounding areas (King et al.,
1994; Stein, 1999; Pace et al., 2014; Verdecchia and Carena, 2016). Static stress
changes produce areas of negative stress, also known as shadow zones, and
positive stress zones. The spatial distributions of decreases (unloading) and
increases (loading) in stress during the long-term slip history of faults likely influence
the distance across strike between major faults. Thus, given a known major active
fault geometrically capable of hosting a Mw ≥ 5.5 earthquake, the possibility that a
future Mw ≥ 5.5 earthquake will occur in the vicinity of the fault, but is not caused by
that fault, should decrease as the distance from the fault decreases. Conversely,
earthquakes with magnitudes lower than 5.5 and those due to slip along minor faults
are likely to occur everywhere within a fault system, including in proximity to a major
fault.
In Figure 7e, we illustrate the results of the analogue and numerical modelling of
fault system evolution and indicate the areas around major faults where it is unlikely
that other major faults develop. In Figure 7f, we show the next step in moving from
geologic and structural considerations. In this step, we combine fault sources and
distributed seismicity source inputs, which serve as inputs of the seismogenic model.
Fault sources are used to model major faults and are represented by a master fault
(i.e., one or more major faults) and its projection at the surface. Distributed seismicity
is used to model seismicity associated with minor, unknown or unmapped faults.
Depending on the positions of distributed seismicity points with respect to the buffer
zones around major faults, the rates of expected distributed seismicity remain
unmodified or decrease and can even reach zero.
Specifically, we introduced a slip rate and a distance-weighted linear function based
on the above reasoning. The probability of the occurrence of an earthquake (Pe) with
a Mw greater than or equal to the minimum magnitude of the fault is as follows:
$$Pe = \begin{cases} 0, & d \leq 1\ km \\ d/d_{max}, & 1\ km < d \leq d_{max} \\ 1, & d > d_{max} \end{cases} \qquad (6)$$

where $d$ is the Joyner-Boore distance from a fault source. The maximum value of $d$
($d_{max}$) is assumed to be controlled by the slip rate of the fault. For faults with slip
rates ≥ 1 mm/yr, we assume $d_{max}$ = L/2 (L is the length along the strike, Fig. 2a); for
faults with slip rates of 0.3 - 1 mm/yr, $d_{max}$ = L/3; and for faults with slip rates of ≤ 0.3
mm/yr, $d_{max}$ = L/4. The rationale for varying $d_{max}$ is given by a simple assumption: the
higher the slip rate is, the larger the deformation field and the higher the value of
$d_{max}$. This linear function has been applied around each fault, without differences
between footwall and hangingwall. We applied eq. (6) to the smoothed occurrence
rates of the distributed seismogenic sources. In Figure 8 we show the annual
cumulative MFD (Fig. 8a) and incremental annual MFD (Fig. 8c) computed for the
red bounded area in Figure 8b. Because we consider three fault source inputs (red
lines in Fig.8): one using only TGR MFD; one using only CHG MFD; and one using
*Mixed model* MFD and because the MFDs of distributed seismicity grid points in the
vicinity of faults are modified with respect to the MFDs of these faults, we obtain
three different inputs of distributed seismicity (blue lines in Fig. 8). These three
distributed seismogenic source inputs differ because the minimum magnitude of the
faults is Mw 5.5 in the TGR model, but this value depends on each fault source
dimension in the CHG and Mixed model. From Mw = 4.5 to Mw = 5.5 the complete
CPT15 is fully described by the MFD of the distributed source input. From Mw = 5.5
to Mw = 6.3 the total MFD (black lines in Fig. 8) computed using only TGR MFD is
higher than the MFD computed using only CHG and Mixed MFD, this because the
annual rates of occurrences of intermediate-magnitude events obtained with TGR
model are higher than the ones obtained with CHG and Mixed model, as shown in
the incremental annual MFD in Figure 8c. From Mw = 6.4 to Mw = 7.3 the total MFDs
computed using only CHG and Mixed MFD are higher the total MFD obtained with
TGR model.
Our approach allows incompleteness in the fault database to be bypassed, which is
advantageous because all fault databases should be considered incomplete. In our
approach, the seismicity is modified only in the vicinity of mapped faults. The
remaining areas are fully described by the *distributed* input. With this approach, we
do not define regions with reliable fault information, and the locations of currently
unknown faults can be easily included when they are discovered in the future.
**3. Results and Discussion**
To probabilistically obtain ground shaking, we assign the calculated seismicity rates,
based on the Poisson hypothesis, to their pertinent geometries, i.e., individual 3D
seismogenic sources for the *fault input* and point sources for the *distributed input*
(Fig. 8). All the computations are performed using the OpenQuake Engine, an open
source software developed recently with the purpose of providing seismic hazard
and risk assessments (Pagani et al., 2014). Moreover, it is widely recognized within
the scientific community for its potential. The ground motion prediction equations
(GMPE) of Akkar et al. (2013), Chiou et al., (2008), Faccioli et al., (2010) and Zhao
et al., (2006) are used, because these GMPEs were selected in the ESHM13
(Woessner et al., 2015) for active shallow crust. In addition, we used the GMPE
proposed by Bindi et al. (2014) and calibrated using Italian data. We combined all
GMPEs into a logic tree with the same weight of 0.2 for each branch. Note that these
GMPEs use different distance metrics: the Joyner and Boore distance for Akkar et al.
(2013), Bindi et al. (2014) and Chiou et al. (2008) and the closest rupture distance
for Faccioli et al. (2010) and Zhao et al. (2006).
The results of the fault source inputs, distributed source inputs, and aggregated
model are expressed in terms of peak ground acceleration (PGA) for exceedance
probabilities of 10% and 2% over 50 years, corresponding to return periods of 475
and 2,475 years, respectively (Fig. 9).
To explore the epistemic uncertainty associated with the MFDs of fault source inputs,
we compared the seismic hazard levels obtained based on the TGR and CHG fault
source inputs (left column in Fig. 9) using the TGR and CHG MFDs for all the fault
sources (details in section 2.1.3). Although both models have the same seismic
moment release, the different MFDs generate clear differences. In fact, for 10%
exceedance probability in 50 yr, in the *TGR* model all faults contribute significantly to
the seismic hazard level, whereas in the *CHG* model, only a few faults located in the
central Apennines and Calabria contribute to the seismic hazard level. This
difference is due to the different shapes of the MFDs in the two models (Fig. 2c). As
shown in Figure 8, the amount of earthquakes with magnitudes between 5.5 and
approximately 6, which are likely the main contributors to these levels of seismic
hazard, is generally higher in the *TGR* model than in the *CHG* model. At a 2%
probability of exceedance in 50 years, all fault sources in the CHG contribute to the
seismic hazard level, but the absolute values are still generally higher in the *TGR*
model.
The *distributed input* (middle column in Fig. 9) depicts a more uniform shape of the
seismic hazard level than that of fault source inputs. A PGA value lower than 0.125 g
at a 10% probability of exceedance over 50 years and lower than 0.225 g at a 2%
probability of exceedance over 50 years encompass a large part of peninsular Italy
and Sicily. Two areas with high levels of ground shaking are located in the central
Apennines and northeastern Sicily.
The overall model, which was obtained by combining the fault and distributed source
inputs, is shown in the right column of Figure 9. Areas with comparatively high
seismic hazard levels, i.e., hazard levels greater than 0.225 g and greater than 0.45
g at 50-yr exceedance probabilities of 10% and 2%, respectively, are located
throughout the Apennines, in Calabria and in Sicily. The fault source inputs
contribute most to the total seismic hazard levels in the Apennines, Calabria and
eastern Sicily, where the highest PGA values are observed.
Figure 10 shows the ratios to the total seismic hazard level by the *fault* and
*distributed* source inputs at a specific site (L'Aquila, 42.400-13.400). Notably, in
Figure 10, *distributed* sources dominate the seismic hazard contribution at
exceedance probabilities greater than ~81% over 50 years, but the contribution of
*fault* sources cannot be neglected. Conversely, at exceedance probabilities of less
than ~10% in 50 years, the total hazard level is mainly associated with *fault* source
inputs. Moreover, note that the contributions are not based on deaggregation but are
computed according to the percentage of each source input in the AFOE value of the
combined model.
Figure 11 presents seismic hazard maps for PGA at 10% and 2% exceedance
probabilities in 50 years for *fault* sources, *distributed* sources and a combination of
the two. These data were obtained using the above-described *Mixed* model, in which
we selected the most "appropriate" MFD model (TGR or CHG) for each fault (as
shown in Figure 3). The results of this model therefore have values between those of
the two end-members shown in Figure 9.
Figure 12 shows the *CHG*, *TGR* and *Mixed* model hazard curves of three sites
(Cesena, L'Aquila and Crotone, Fig. 13c). As previously noted, the results of the
*Mixed* model, due to the structure of the model, are between those of the *CHG* and
*TGR* models. The relative positions of the hazard curves derived from the two end-
member models and the *Mixed* model depend on the number of nearby fault sources
that have been modelled using one of the MFD models and on the distance of the
site from the faults. For example, in the case of the Crotone site, the majority of the
fault sources in the *Mixed* model are modelled using the CHG MFD. Thus, the
resulting hazard curve is similar to that of the *CHG* model. For the Cesena site, the
three hazard curves overlap. Because the distance between Cesena and the closest
fault sources is approximately 60 km, the impact of the fault input is less than the
impact of the *distributed* source input. In this case, the choice of a particular MFD
model has a limited impact on the modelling of *distributed* sources. Notably, for an
annual frequency of exceedance (*AFOE*) higher than $10^{-4}$, the *TGR fault* source
input values are generally higher than those of the *CHG* source input, and the three
models converge at *AFOE* lower than $10^{-4}$, as shown for L'Aquila site. The resulting
seismic hazard estimates depend on the assumed MFD model (*TGR* vs. *CHG*), and
for the investigated range of AFOE, especially on the annual rates of occurrences of
intermediate-magnitude events (5.5 to ~6.5, see Fig. 8). Therefore, the *TGR* model
leads to the highest hazard values because this range of magnitude (5.5 to ~ 6.5)
contributes the most to the hazard level.
In Figure 13, we investigated the influences of the Mixed *fault* source inputs and the
Mixed *distributed* source inputs on the total hazard level of the entire study area, as
well as the spatial variability. The maps in Figure 13a show that the contribution of
*fault* inputs to the total hazard level generally decreases as the exceedance
probability increases from 2% to 81% in 50 years. At a 2% probability of exceedance
in 50 years, the total hazard levels in the Apennines and eastern Sicily are mainly
related to faults, whereas at an 81% probability of exceedance in 50 years, the
contributions of *fault* inputs are high in local areas of central Italy and southern
Calabria.
Moreover, we examined the contributions of *fault* and *distributed* sources along three
E-W-oriented profiles in northern, central and southern Italy (Fig. 13b). In areas with
faults, the hazard level estimated by *fault* inputs is generally higher than that
estimated by the corresponding *distributed* source inputs. Notable exceptions are
present in areas proximal to slow-slipping active faults at an 81% probability of
exceedance in 50 years (profile A), such as those at the eastern and western
boundaries of the fault area in central Italy (profile B), and in areas where the
contribution of the *distributed* source input is equal to that of the *fault* input at a 10%
probability of exceedance in 50 years (eastern part of profile C).
The features depicted by the three profiles result from a combination of the slip rates
and spatial distributions of faults for *fault* source inputs. The proposed approach
requires a high level of expertise in active tectonics and cautious expert judgement
at many levels in the procedure. First, the seismic hazard estimate is based on the
definition of a segmentation model, which requires a series of rules based on
observations and empirical regression between earthquakes and the size of the
causative fault. New data might make it necessary to revise the rules or reconsider
the role of the segmentation. In some cases, expert judgement could permit
discrimination among different fault source models. Alternatively, all models should
be considered branches in a logic tree approach.
Moreover, we propose a fault seismicity input in which the MFD of each fault source
has been chosen based on an analysis of the occurrences of earthquakes that can
be tentatively or confidently assigned to a certain fault. To describe the fault activity,
we applied a probability density function to the magnitude, as commonly performed
in the literature: the TGR model, where the maximum magnitude is the upper
threshold and $M_w = 5.5$ is the lower threshold for all faults, and the characteristic
maximum magnitude model, which consists of a truncated normal distribution
centred on the maximum magnitude. Other MFDs have been proposed to model the
earthquake recurrence of a fault. For example, Youngs and Coppersmith (1985)
proposed a modification to the truncated exponential model to allow for the
increased likelihood of characteristic events. However, we focused only on two
models, as we believe that instead of a "blind" or qualitative characterization of the
MFD of a fault source, future applications of statistical tests of the compatibility
between expected earthquake rates and observed historical seismicity could be used
as an objective method of identifying the optimal MFD of expected seismicity. As
shown in this analyses, fault sources, even if modelled by TGR or CHG MFD, are
able to match occurred seismicity for magnitude ~> 5.5 (see for example Fig. 8) and
so are complementary to other inputs that model seismicity using area sources or
smoothing approaches.
To focus on the general procedure for spatially integrating faults with sources
representing distributed (or off-fault) seismicity, we did not investigate the impact of
other smoothing procedures on the distributed sources, and we used fixed kernels
with a constant bandwidth (as in the works of Kagan and Jackson, 1994; Frankel et
al. 1997; Zechar and Jordan, 2010). The testing of adaptive bandwidths (e.g., Stock
and Smith, 2002; Helmstetter et al., 2006, 2007; Werner et al., 2010; Hiemer et al,
2014) or weighted combinations of both models has been reserved for future studies.
Finally, we compared, as shown in Figure 14, the 2013 European Seismic Hazard
Model (ESHM13) developed within the SHARE project, the current Italian national
seismic hazard map (MPS04) and the results of our model (Mixed model) using the
same GMPEs as used in this study. Specifically, for ESHM13, we compared the
results to the fault-based hazard map (FSBG model) that accounts for fault sources
and background seismicity. The figure shows how the impact of our fault sources is
more evident than in FSBG-ESHM13, and the comparison with MPS04 confirms a
similar pattern, but with some significant differences at the regional to local scales.
The strength of our approach lies in the integration of different levels of information
regarding the active faults in Italy, but the final result is unavoidably linked to the
quality of the relevant data. Our work focused on presenting and applying a new
approach for evaluating seismic hazards based on active faults and intentionally
avoided the introduction of uncertainties due to the use of different segmentation
rules or other slip rate values of faults. Moreover, the impact of ground motion
predictive models is important in seismic hazard assessment but beyond the scope
of this work. Future steps will be devoted to analysing these uncertainties and
evaluating their impacts on seismic hazard estimates.

## 703   4. Conclusions

We presented a seismogenic source model for Italy, which summarizes and
integrates the fault-based models developed within the last decade (Pace et al.,

706    2006).

The model proposed in this study combines fault source inputs based on over 110
faults grouped into 86 fault sources and distributed source inputs. For each fault
source, the maximum magnitude and its uncertainty were derived by applying
scaling relationships, and the rates of seismic activity were derived by applying slip
rates to seismic moment evaluations and balancing these seismic moments using
two MFD models.
To account for unknown faults, a distributed seismicity input was applied following
the well-known Frankel (1995) methodology to calculate seismicity parameters.
The fault sources and gridded distributed seismicity sources have been integrated
via a new approach based on the idea that deformation in the vicinity of an active
fault is concentrated along the fault and that the seismic activity in the surrounding
region is reduced. In particular, a distance-dependent linear weighting function has
been introduced to allow the contribution of distributed sources (in the magnitude
range overlapping the MFD of each fault source) to linearly decrease from 1 to 0 with
decreasing distance from a fault. The strength of our approach lies in the ability to
integrate different levels of available information for active faults that actually exist in
Italy (or elsewhere), but the final result is unavoidably linked to the quality of the
relevant data. We think that our seismogenic source model includes significant
advances in the use of integrated active fault and seismological data.
The probabilistically estimated ground shaking maps produced using our model
show a hazard pattern similar to that of the current maps at the national scale, but
some significant differences in hazard level are present at the regional to local scales
(Figure 13).
Moreover, the impact of using different MFD models to derive seismic activity rates
has on the hazard maps was investigated. The PGA values in the hazard maps
obtained with the *TGR* model are higher than those in the hazard maps based on the
*CHG* model. This difference is because the rates of earthquakes with magnitudes
from 5.5 to approximately 6 are generally higher in the *TGR* model than in the *CHG*
model. Moreover, the relative contributions of fault source inputs and distributed
source inputs have been identified in maps and profiles in three sectors of the study
area. These profiles show that the hazard level is generally higher where fault inputs
are used, and for high probabilities of exceedance, the contribution of *distributed*
inputs equals that of *fault* inputs.
Finally, the *Mixed* model was created by selecting the most appropriate MFD model
for each fault. All data, including the locations and parameters of fault sources, are
provided in the supplemental files of this paper.
It shall be noted that our new seismogenic source model is not intended to replace,
integrate or assess the current official national seismic hazard model of Italy. While
some aspects remain to be implemented in our approach (e.g., the integration of
reverse/thrust faults in the database, sensitivity tests for the distance-dependent
linear weighting function parameters, sensitivity tests for potential different
segmentation models, and fault source inputs that account for fault interactions), the
proposed model represents advancements in terms of input data (quantity and
quality) and methodology based on a decade of research in the field of fault-based
approaches to regional seismic hazard modelling.

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

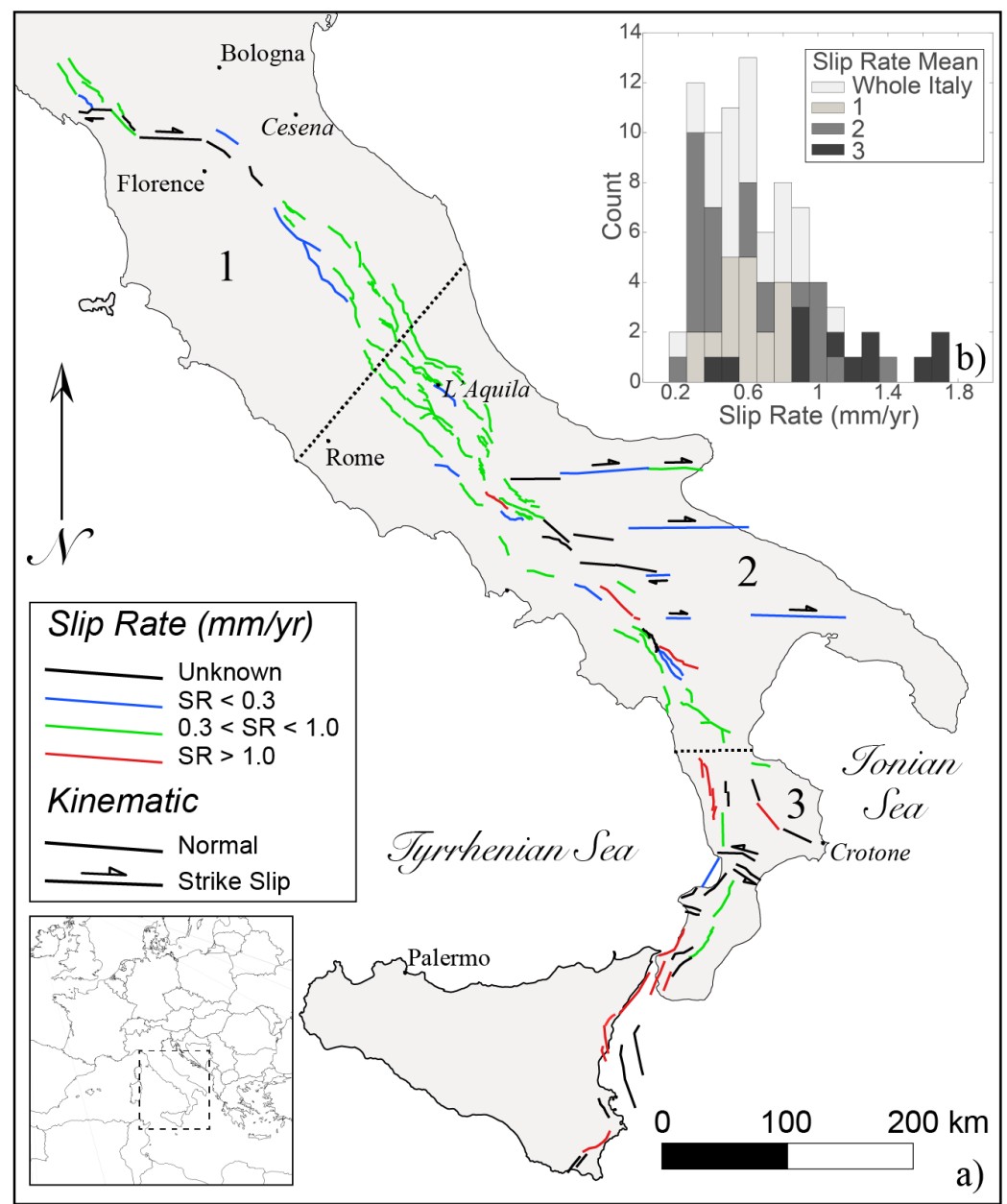

Fig. 1 a) Map of normal and strike-slip active faults used in this study. The colour scale indicates the slip rate. b) Histogram of the slip rate distribution in the entire study area and in three subsectors. The numbers 1, 2 and 3 represent the Northern Apennines, Central-Southern Apennines and Calabria-Sicilian coast regions, respectively. The dotted black lines are the boundaries of the regions.

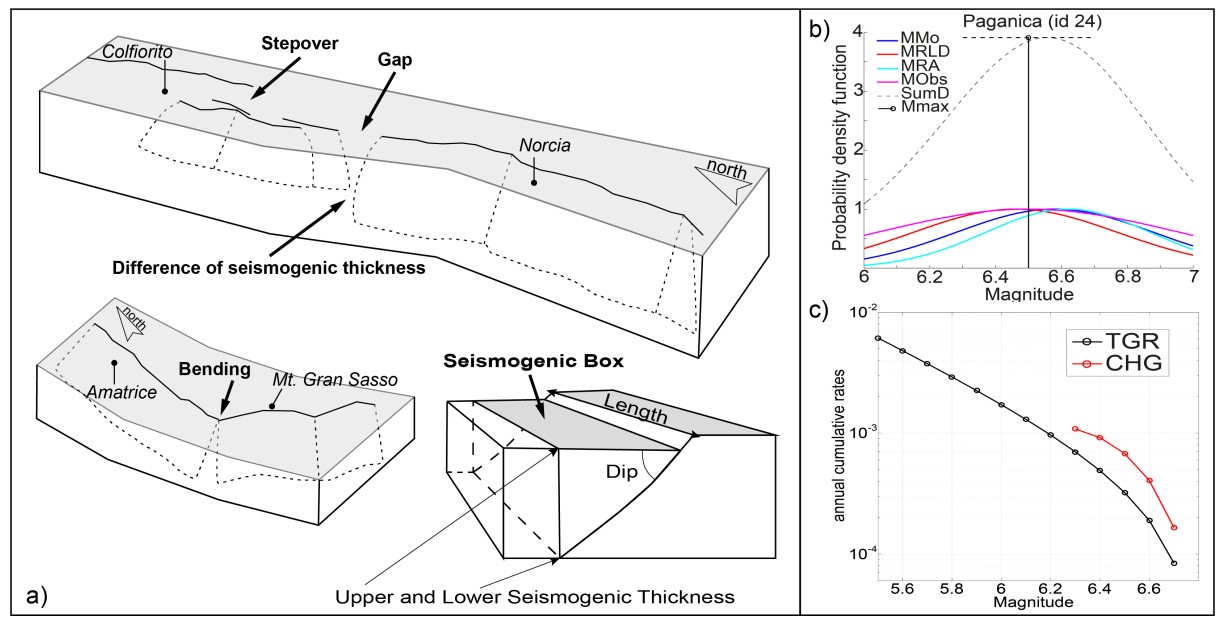

Fig. 2 a) Conceptual model of active faults and segmentation rules adopted to define
a fault source and its planar projection, forming a seismogenic box [modified from
Boncio et al., 2004]. b) Example of FiSH code output (see Pace et al., 2016 for
details) for the Paganica fault source showing the magnitude estimates from
empirical relationships and observations, both of which are affected by uncertainties.
In this example, four magnitudes are estimated: MMo (blue line) is from the standard
formula (IASPEI, 2005); MRLD (red line) and MRA (cyan line) correspond to
estimates based on the maximum subsurface fault length and maximum rupture area
from the empirical relationships of Wells and Coppersmith (1994) for length and
area, respectively; and Mobs (magenta line) is the largest observed moment
magnitude. The black dashed line represents the summed probability density curve
(SumD), the vertical black line represents the central value of the Gaussian fit of the
summed probability density curve (Mmax), and the horizontal black dashed line
represents its standard deviation (σMmax). The input values that were used to
obtain this output are provided in Table 1. c) Comparison of the magnitude–
frequency distributions of the Paganica source, which were obtained using the CHG
model (red line) and the TGR model (black line).

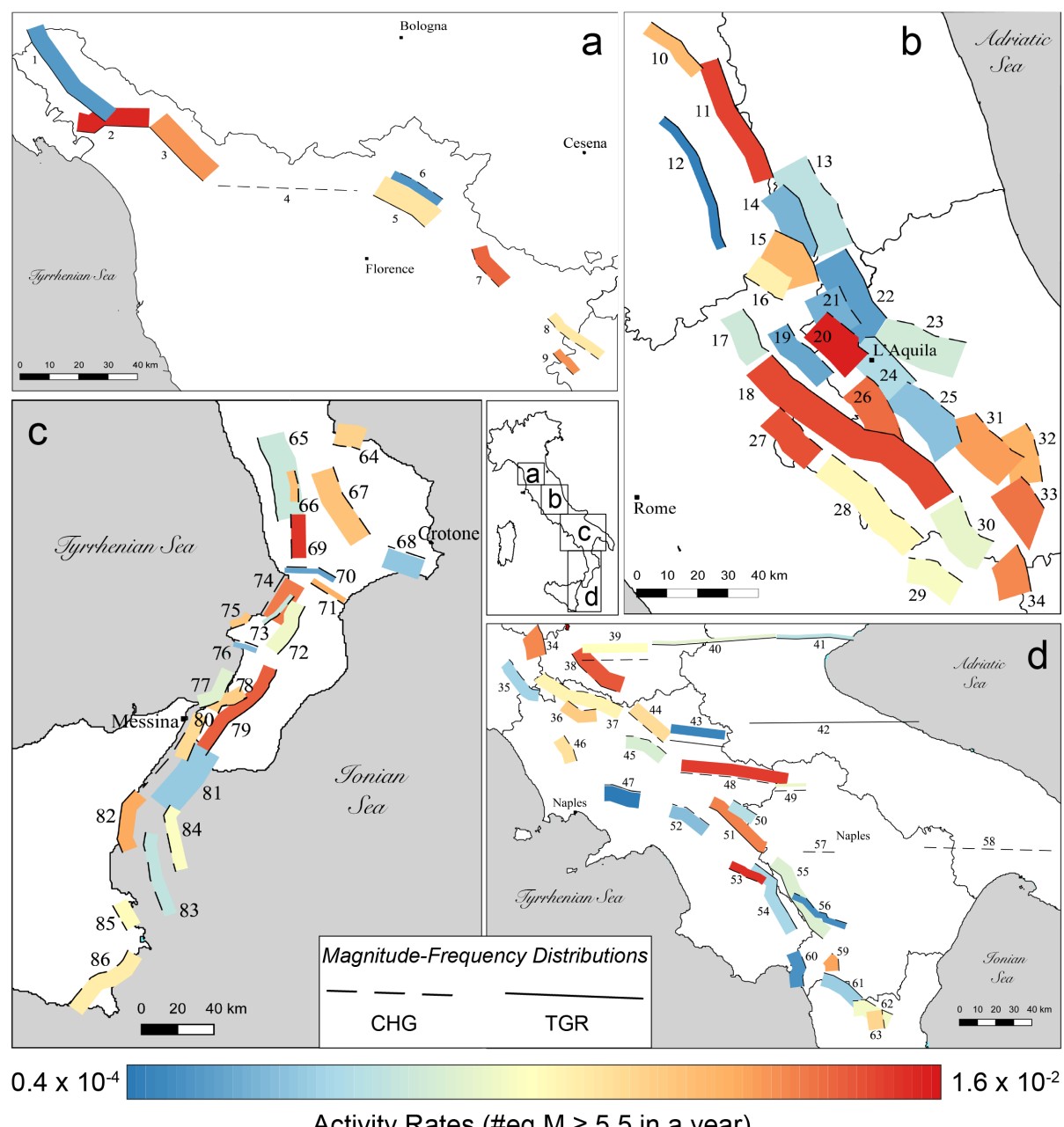

Fig. 3 Maps showing the fault source inputs as seismogenic boxes (see Fig. 2a). The
colour scale indicates the activity rate. Solid and dashed lines (corresponding to the
uppermost edge of the fault) are used to highlight our choice between the two end-
members of the MFD model adopted in the so-called *Mixed* model.

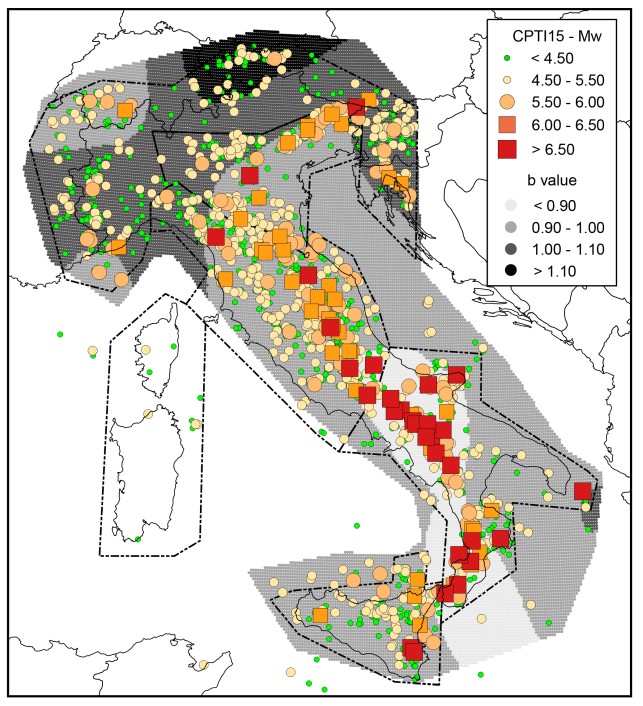


Fig. 4 Historical earthquakes from the most recent version of the historical
parametric Italian catalogue (CPTI15, Rovida et al., 2016), the spatial variations in b-
values and the polygons defining the five macroseismic areas used to assess the
magnitude completeness intervals (Stucchi et al, 2011).

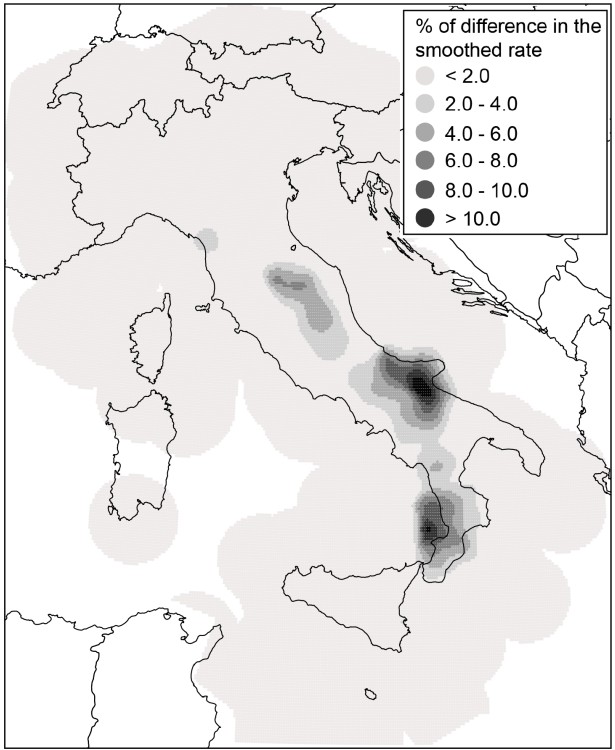


Fig. 5 Differences in percentages between the two smoothed rates computed with
eq. (2) using the complete catalogue and the modified catalogue without events
associated with known active faults (*TGR* model)

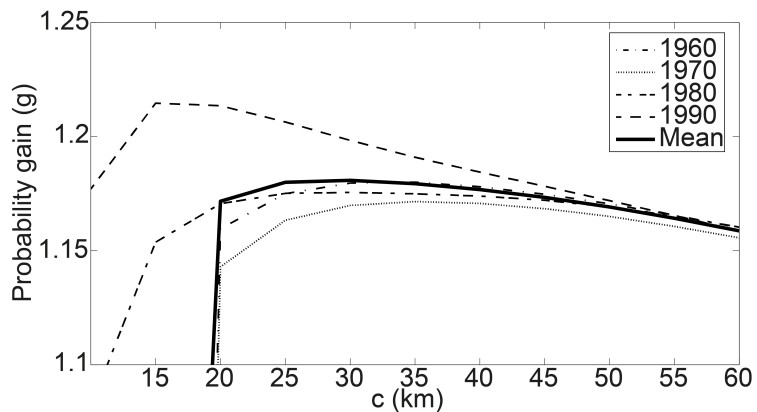


Fig. 6 Probability gain per earthquake (see eq. 3) versus correlation distance $c$, used
to determine the best radius for use in the smoothed seismicity approach (eq. 2)




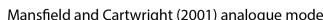

around master fault there is an area (between dahed lines) where no more than 1 single major fault is likely to be developed

minor faults

major faults (line weight proportional to throw)

a)

b)

~@ half of the timeline evolution of the experiment

@ end of the timeline evolution of the experiment

Cowie et al. (1993) numeric model

c)

d)

~@ half of the timeline evolution of the simulation

~@ end of the timeline evolution of the simulation

e)

major faults

buffer area around master fault, where no more than 1 single major fault is likely

minor faults

f)

major faults became master fault (Seismogenic Source model)

surface projection of the master fault at depth. Width depends on the dip angle

seismicity of the minor faults is modeled into the Distributed Seismicity model

rates

point located outside master fault buffer area

magnitude

rates

point located inside master fault buffer area

rates >= Mmin of the Seismogenic source are reduced

magnitude

rates

point located inside master fault surface projection

rates truncated at magnitude >= Mmin of the Seismogenic source

magnitude


Fig. 7 Fault system evolution and its implications for our model. a) and b) Diagrams
from the Mansfield and Cartwright (2001) analogue experiment in two different
stages: the approximate midpoint of the sequence and the end of the sequence.
Areas exist around master faults where no more than a single major fault is likely to
develop. c) and d) Diagrams from numerical modelling conducted by Cowie et al.
(1993) in two different stages. This experiment shows the similar evolutional features
of major and minor faults. e) and f) Application of the analogue and numerical
modelling of fault system evolution to the fault source input proposed in this paper. A
buffer area is drawn around each fault source, where it is unlikely for other major
faults to develop, accounting for the length and slip rate of the fault source. This
buffer area is useful for reducing or truncating the rates of expected distributed
seismicity based on the position of a distributed seismicity point with respect to the
buffer zone (see the text for details).

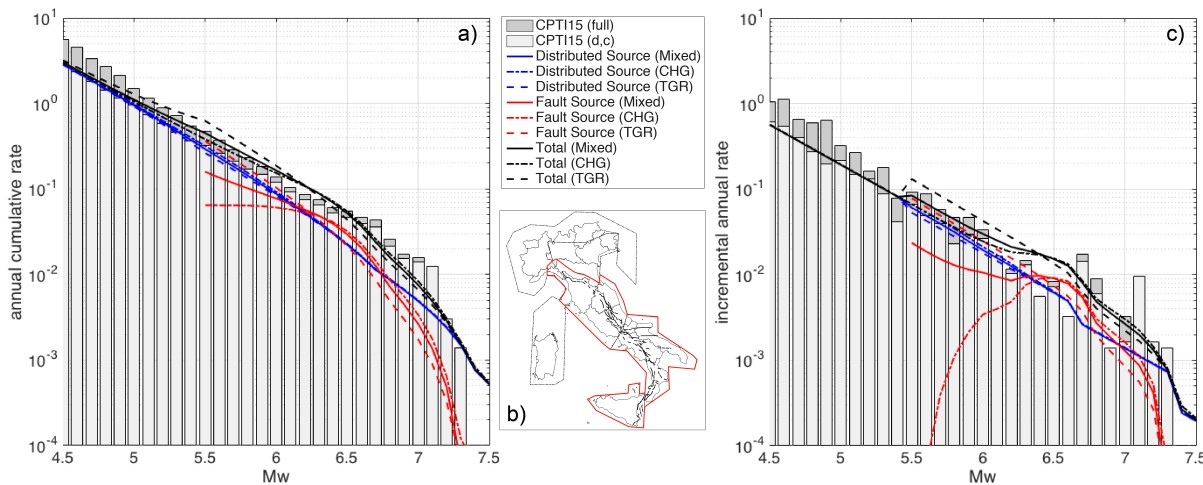


Fig. 8 a) Annual cumulative MFD and c) incremental annual MFD computed for the
red bounded area in b). The rates have been computed using: (i) the full CPTI15
catalogue; (ii) the declustered and complete catalogue (CPTI15 (d, c) in the legend)
obtained using the completeness magnitude thresholds over different periods of time
given by Stucchi et al. (2011) for five large zones; (iii) the distributed sources; (iv) the
fault sources; and (v) summing fault and distributed sources (Total).

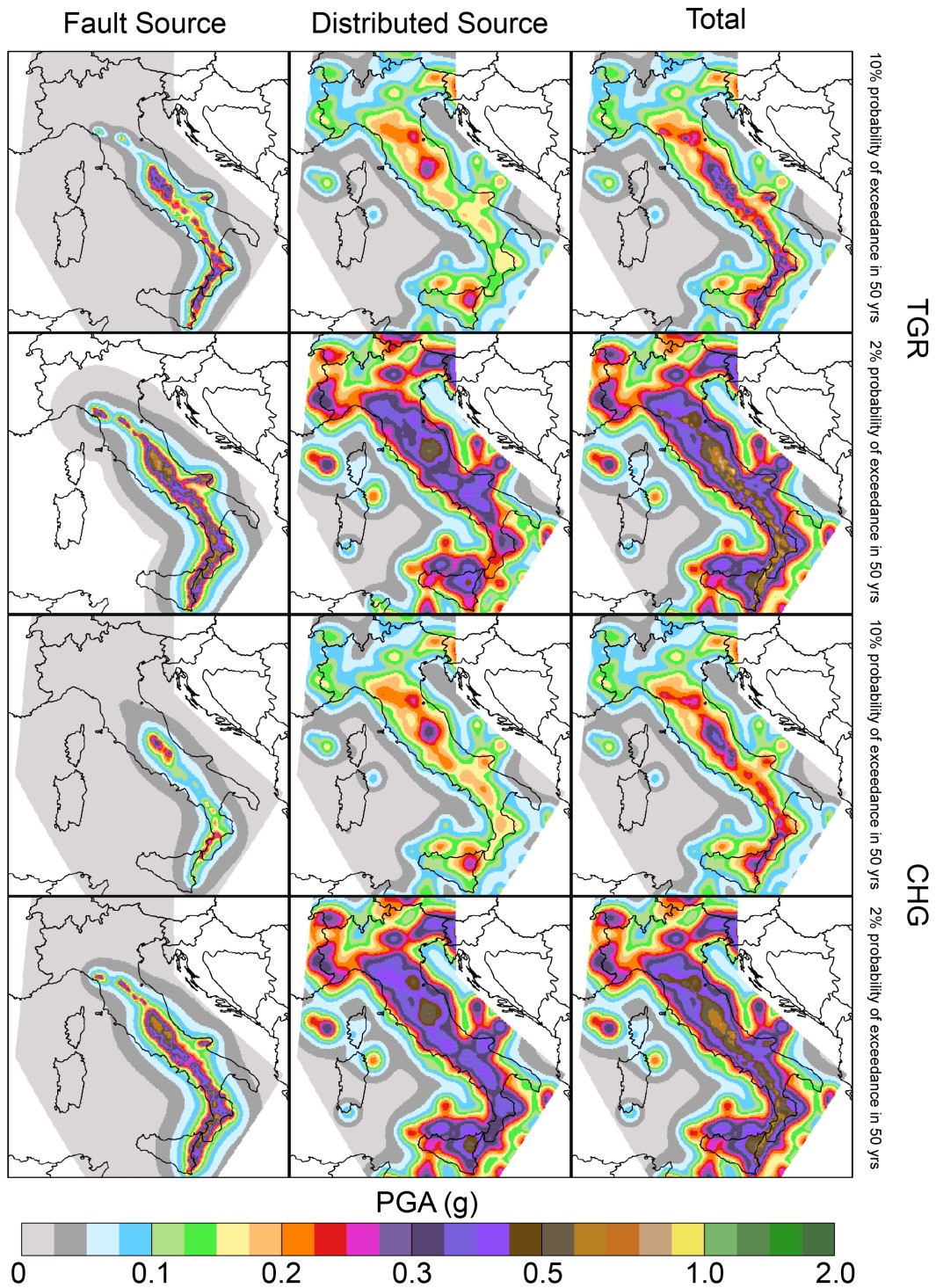

Fig. 9 Seismic hazard maps for the *TGR* and *CHG* models expressed in terms of peak ground acceleration (PGA) and computed for a latitude/longitude grid spacing of 0.05°. The first and second rows show the fault source, distributed source and total maps of the *TGR* model computed for 10% probability of exceedance in 50 years and 2% probability of exceedance in 50 years, corresponding to return periods of 475 and 2475 years, respectively. The third and fourth rows show the same maps for the *CHG* model.

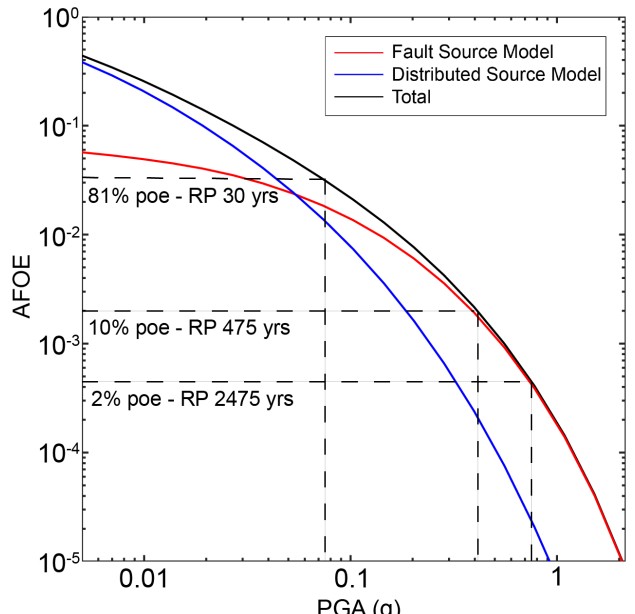


Fig. 10 An example of the contribution to the total seismic hazard level (black line), in
terms of hazard curves, by the *fault* (red line) and *distributed* (blue line) source inputs
for one of the 45,602 grid points (L'Aquila, 42.400-13.400). The dashed lines
represent the 2%, 10% and 81% probabilities of exceedance (poes) in 50 years.

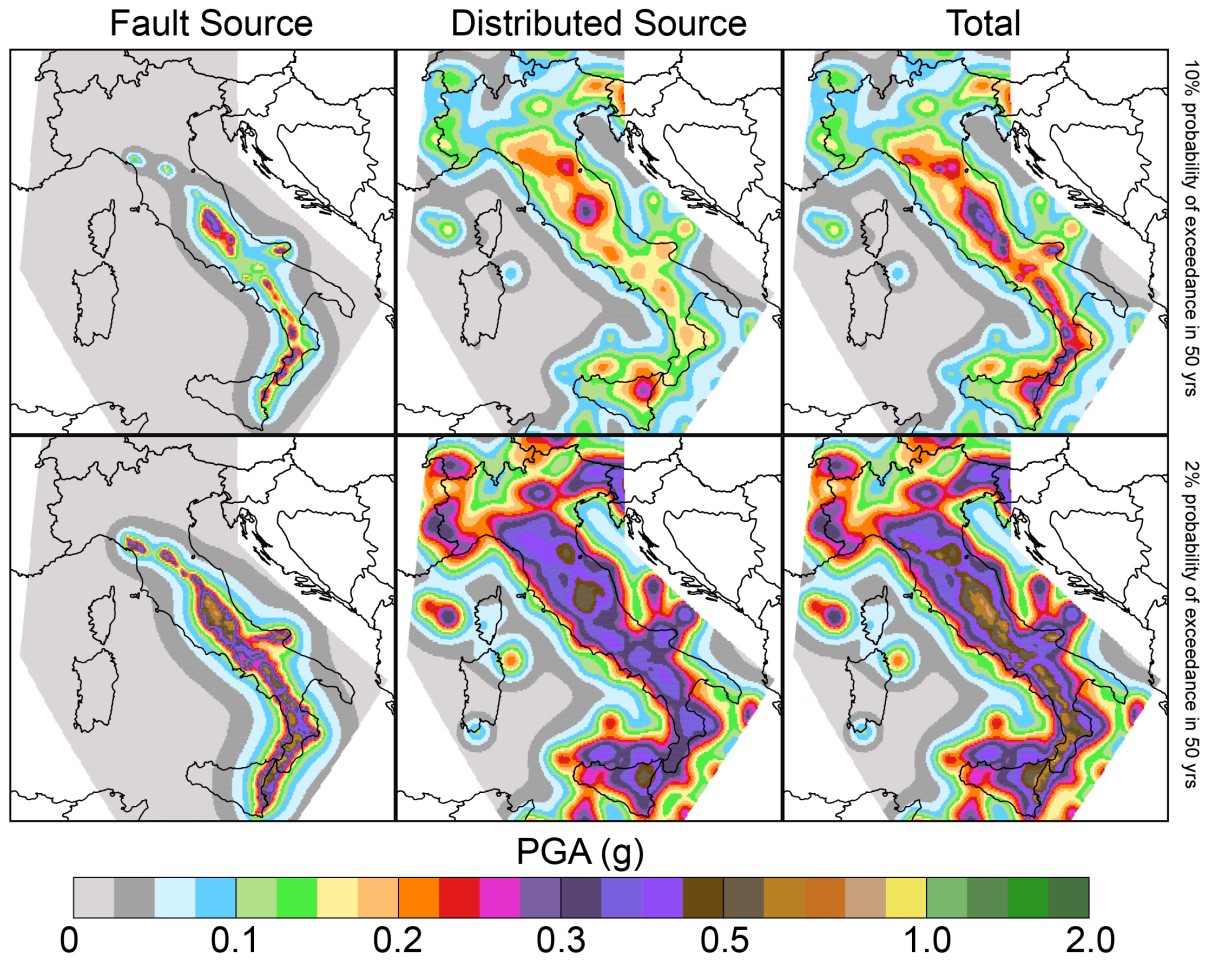

Fig. 11 Seismic hazard maps for the *Mixed* model. The first row shows the fault source, distributed source and total maps computed for 10% probability of exceedance in 50 years, and the second row shows the same maps but computed for 2% probability of exceedance in 50 years, corresponding to return periods of 475 and 2475 years, respectively. The results are expressed in terms of peak ground acceleration (PGA).

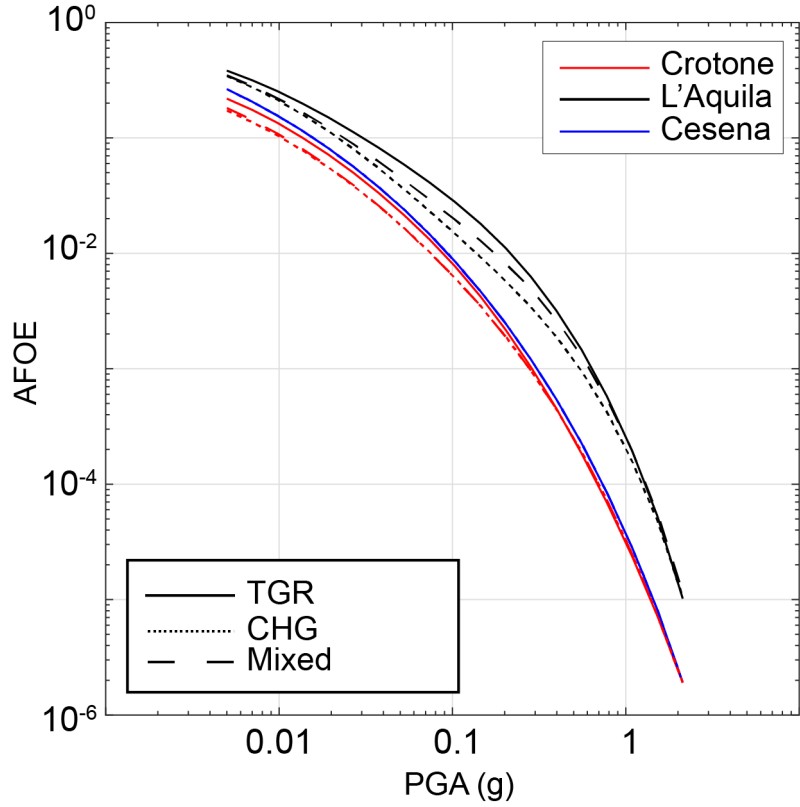


Fig. 12 *CHG* (dotted line), *TGR* (solid line) and *Mixed* model (dashed line) hazard
curves for three sites (see Fig. 13 for the location): Cesena (red line), L'Aquila (black
line) and Crotone (blue line)

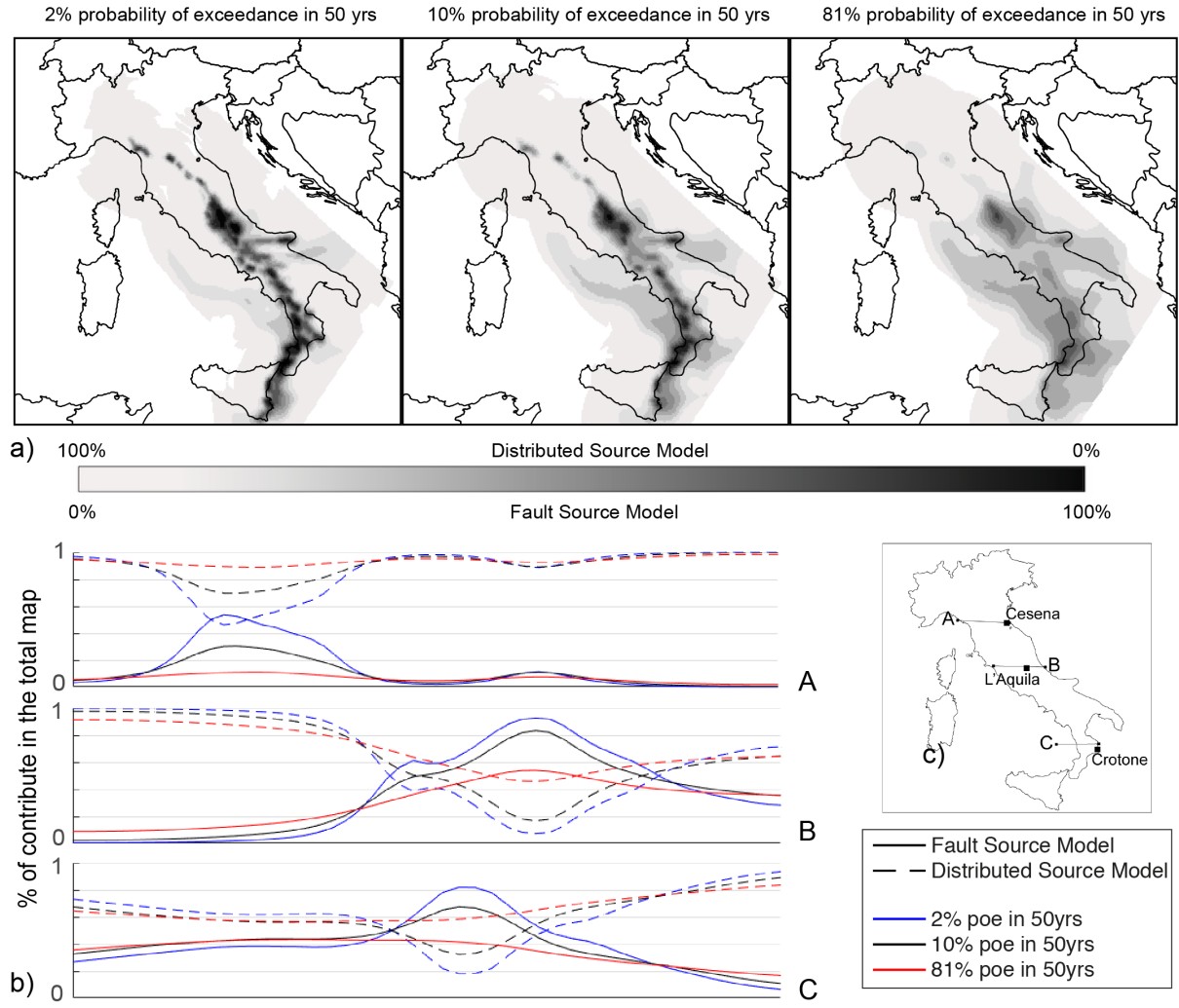

Fig. 13 a) Contribution maps of the Mixed *fault* and *distributed* source inputs to the total hazard level for three probabilities of exceedance: 2%, 10% and 81%, corresponding to return periods of 2475, 475 and 30 years, respectively. b) Contributions of the Mixed *fault* (solid line) and *distributed* (dashed line) source inputs along three profiles (A, B and C in Fig. 13c) for three probabilities of exceedance: 2% (blue line), 10% (black line) and 81% (red line).


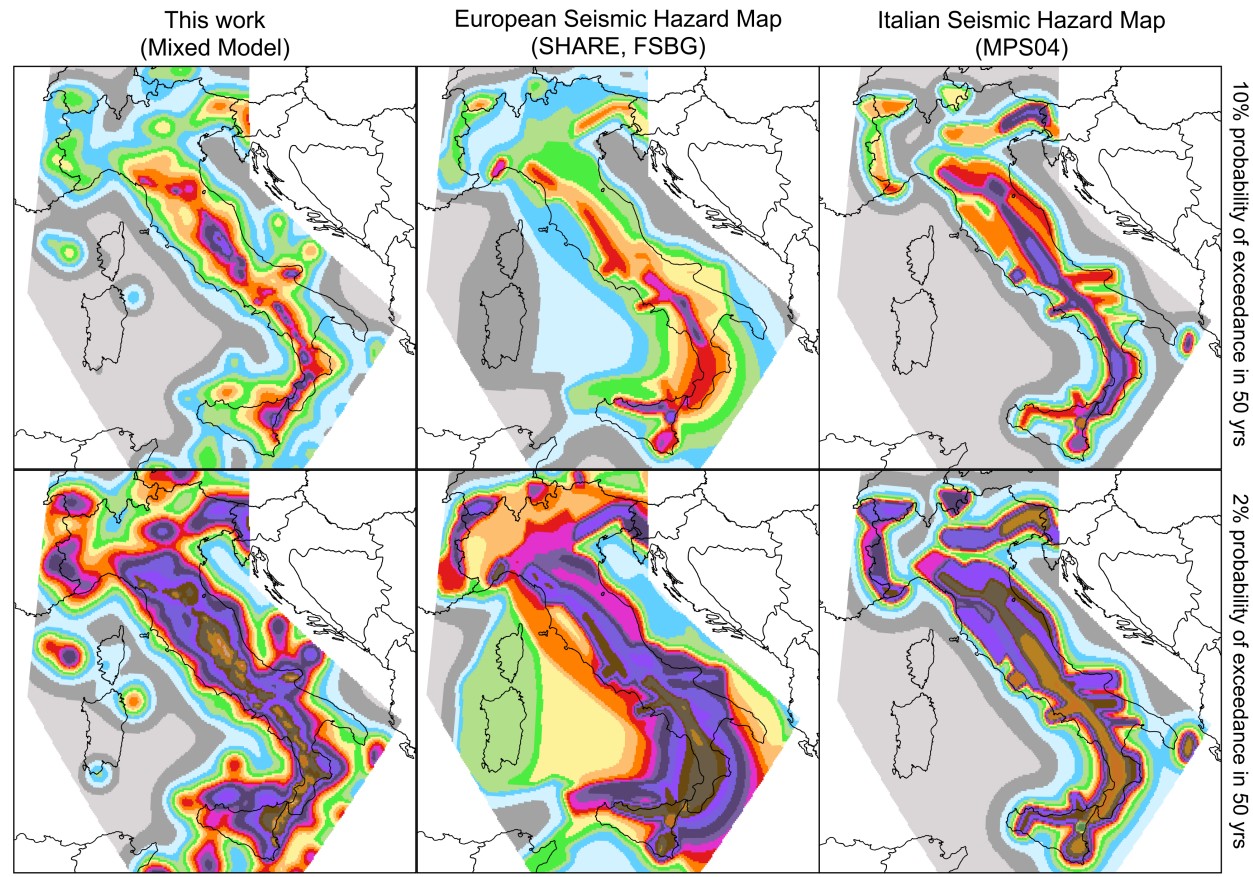

| This work (Mixed Model) | European Seismic Hazard Map (SHARE, FSBG) | Italian Seismic Hazard Map (MPS04) | |

10% probability of exceedance in 50 yrs

2% probability of exceedance in 50 yrs

Fig. 14 Seismic hazard maps expressed in terms of Peak Ground Acceleration
(PGA) and computed for a latitude/longitude grid spacing of 0.05° based on rock site
conditions. The figure shows a comparison of our model (*Mixed* model, on the left),
the ESHM13 model (FSBG logic tree branch, in the middle) and the current Italian
national seismic hazard map (MPS04, on the right). The same combination of
GMPEs (Akkar et al. 2013, Chiou et al., 2008, Faccioli et al., 2010 and Zhao et al.,
2006 and Bindi et al. 2014), were used for all models to obtain and compare the
maps*.





| ID | Fault Sources | L (km) | Dip (°) | Upper (km) | Lower (km) | SR$_{min}$ (mm/yr) | SR$_{max}$ (mm/yr) |
|----|---------------|--------|---------|------------|------------|------------|------------|
| 1 | Lunigiana | 43.8 | 40 | 0 | 5 | 0.28 | 0.7 |
| 2 | North Apuane Transfer | 25.5 | 45 | 0 | 7 | 0.33 | 0.83 |
| 3 | Garfagnana | 26.9 | 30 | 0 | 4.5 | 0.35 | 0.57 |
| 4 | Garfagnana Transfer | 47.1 | 90 | 2 | 7 | 0.33 | 0.83 |
| 5 | Mugello | 21.0 | 40 | 0 | 7 | 0.33 | 0.83 |
| 6 | Ronta | 19.3 | 65 | 0 | 7 | 0.17 | 0.5 |
| 7 | Poppi | 17.1 | 40 | 0 | 4.5 | 0.33 | 0.83 |
| 8 | Città di Castello | 22.9 | 40 | 0 | 3 | 0.25 | 1.2 |
| 9 | M.S.M. Tiberina | 10.5 | 40 | 0 | 2.5 | 0.25 | 0.75 |
| 10 | Gubbio | 23.6 | 50 | 0 | 6 | 0.4 | 1.2 |
| 11 | Colfiorito System | 45.9 | 50 | 0 | 8 | 0.25 | 0.9 |
| 12 | Umbra Valley | 51.1 | 55 | 0 | 4.5 | 0.4 | 1.2 |
| 13 | Vettore-Bove | 35.4 | 50 | 0 | 15 | 0.2 | 1.05 |
| 14 | Nottoria-Preci | 29.0 | 50 | 0 | 12 | 0.2 | 1 |
| 15 | Cascia-Cittareale | 24.3 | 50 | 0 | 13.5 | 0.2 | 1 |
| 16 | Leonessa | 14.9 | 55 | 0 | 12 | 0.1 | 0.7 |
| 17 | Rieti | 17.6 | 50 | 0 | 10 | 0.25 | 0.6 |
| 18 | Fucino | 82.3 | 50 | 0 | 13 | 0.3 | 1.6 |
| 19 | Sella di Corno | 23.1 | 60 | 0 | 13 | 0.35 | 0.7 |
| 20 | Pizzoli-Pettino | 21.3 | 50 | 0 | 14 | 0.3 | 1 |
| 21 | Montereale | 15.1 | 50 | 0 | 14 | 0.25 | 0.9 |
| 22 | Gorzano | 28.1 | 50 | 0 | 15 | 0.2 | 1 |
| 23 | Gran Sasso | 28.4 | 50 | 0 | 15 | 0.35 | 1.2 |
| 24 | Paganica | 23.7 | 50 | 0 | 14 | 0.4 | 0.9 |
| 25 | Middle Aternum Valley | 29.1 | 50 | 0 | 14 | 0.15 | 0.45 |
| 26 | Campo Felice-Ovindoli | 26.2 | 50 | 0 | 13 | 0.2 | 1.6 |
| 27 | Carsoli | 20.5 | 50 | 0 | 11 | 0.35 | 0.6 |
| 28 | Liri | 42.5 | 50 | 0 | 11 | 0.3 | 1.26 |
| 29 | Sora | 20.4 | 50 | 0 | 11 | 0.15 | 0.45 |
| 30 | Marsicano | 20.0 | 50 | 0 | 13 | 0.25 | 1.2 |
| 31 | Sulmona | 22.6 | 50 | 0 | 15 | 0.6 | 1.35 |
| 32 | Maiella | 21.4 | 55 | 0 | 15 | 0.7 | 1.6 |
| 33 | Aremogna C.Miglia | 13.1 | 50 | 0 | 15 | 0.1 | 0.6 |
| 34 | Barrea | 17.1 | 55 | 0 | 13 | 0.2 | 1 |
| 35 | Cassino | 24.6 | 60 | 0 | 11 | 0.25 | 0.5 |
| 36 | Ailano-Piedimonte | 17.6 | 60 | 0 | 12 | 0.15 | 0.35 |
| 37 | Matese | 48.3 | 60 | 0 | 13 | 0.2 | 1.9 |
| 38 | Bojano | 35.5 | 55 | 0 | 13 | 0.2 | 0.9 |
| 39 | Frosolone | 36.1 | 70 | 11 | 25 | 0.35 | 0.93 |
| 40 | Ripabottoni-San Severo | 68.3 | 85 | 6 | 25 | 0.1 | 0.5 |
| 41 | Mattinata | 42.3 | 85 | 0 | 25 | 0.7 | 1 |
| 42 | Castelluccio dei Sauri | 93.2 | 90 | 11 | 22 | 0.1 | 0.5 |
| 43 | Ariano Irpino | 30.1 | 70 | 11 | 25 | 0.35 | 0.93 |
| 44 | Tammaro | 25.0 | 60 | 0 | 13 | 0.35 | 0.93 |
| 45 | Benevento | 25.0 | 55 | 0 | 10 | 0.35 | 0.93 |
| 46 | Volturno | 15.7 | 60 | 1 | 13 | 0.23 | 0.57 |
| 47 | Avella | 20.5 | 55 | 1 | 13 | 0.2 | 0.7 |
| 48 | Ufita-Bisaccia | 59.0 | 64 | 1.5 | 15 | 0.35 | 0.93 |

| | | | | | | | |
|---|---|---|---|---|---|---|---|
| 49 | Melfi | 17.2 | 80 | 12 | 22 | 0.1 | 0.5 |
| 50 | Irpinia Antithetic | 15.0 | 60 | 0 | 11 | 0.2 | 0.53 |
| 51 | Irpinia | 39.7 | 65 | 0 | 14 | 0.3 | 2.5 |
| 52 | Volturara | 23.7 | 60 | 1 | 13 | 0.2 | 0.35 |
| 53 | Alburni | 20.4 | 60 | 0 | 8 | 0.35 | 0.7 |
| 54 | Caggiano-Diano Valley | 46.0 | 60 | 0 | 12 | 0.35 | 1.15 |
| 55 | Pergola-Maddalena | 50.6 | 60 | 0 | 12 | 0.20 | 0.93 |
| 56 | Agri | 34.9 | 50 | 5 | 15 | 0.8 | 1.3 |
| 57 | Potenza | 17.8 | 90 | 15 | 21 | 0.1 | 0.5 |
| 58 | Palagianello | 73.3 | 90 | 13 | 22 | 0.1 | 0.5 |
| 59 | Monte Alpi | 10.9 | 60 | 0 | 13 | 0.35 | 0.9 |
| 60 | Maratea | 21.6 | 60 | 0 | 13 | 0.46 | 0.7 |
| 61 | Mercure | 25.8 | 60 | 0 | 13 | 0.2 | 0.6 |
| 62 | Pollino | 23.8 | 60 | 0 | 15 | 0.22 | 0.58 |
| 63 | Castrovillari | 10.3 | 60 | 0 | 15 | 0.2 | 1.15 |
| 64 | Rossano | 14.9 | 60 | 0 | 22 | 0.5 | 0.6 |
| 65 | Crati West | 49.7 | 45 | 0 | 15 | 0.84 | 1.4 |
| 66 | Crati East | 18.4 | 60 | 0 | 8 | 0.75 | 1.45 |
| 67 | Lakes | 43.6 | 60 | 0 | 22 | 0.75 | 1.45 |
| 68 | Fuscalto | 21.1 | 60 | 2 | 22 | 0.75 | 1.45 |
| 69 | Piano Lago-Decollatura | 25.0 | 60 | 1 | 15 | 0.23 | 0.57 |
| 70 | Catanzaro North | 29.5 | 80 | 3 | 20 | 0.75 | 1.45 |
| 71 | Catanzaro South | 21.3 | 80 | 3 | 20 | 0.75 | 1.45 |
| 72 | Serre | 31.6 | 60 | 0 | 15 | 0.7 | 1.15 |
| 73 | Vibo | 23.0 | 80 | 0 | 15 | 0.75 | 1.45 |
| 74 | Sant'Eufemia Gulf | 24.8 | 40 | 1 | 11 | 0.11 | 0.3 |
| 75 | Capo Vaticano | 13.7 | 60 | 0 | 8 | 0.75 | 1.45 |
| 76 | Coccorino | 13.3 | 70 | 3 | 11 | 0.75 | 1.45 |
| 77 | Scilla | 29.7 | 60 | 0 | 13 | 0.8 | 1.5 |
| 78 | Sant'Eufemia | 19.2 | 60 | 0 | 13 | 0.75 | 1.45 |
| 79 | Cittanova-Armo | 63.8 | 60 | 0 | 13 | 0.45 | 1.45 |
| 80 | Reggio Calabria | 27.2 | 60 | 0 | 13 | 0.7 | 2 |
| 81 | Taormina | 38.7 | 30 | 3 | 13 | 0.9 | 2.6 |
| 82 | Acireale | 39.4 | 60 | 0 | 15 | 1.15 | 2.3 |
| 83 | Western Ionian | 50.1 | 65 | 0 | 15 | 0.75 | 1.45 |
| 84 | Eastern Ionian | 39.3 | 65 | 0 | 15 | 0.75 | 1.45 |
| 85 | Climiti | 15.7 | 60 | 0 | 15 | 0.75 | 1.45 |
| 86 | Avola | 46.9 | 60 | 0 | 16 | 0.8 | 1.6 |

Table 1 Geometric Parameters of the Fault Sources. L, along-strike length; Dip, inclination angle of the fault plane; Upper and Lower, the thickness bounds of the local seismogenic layer; SRmin and SRmax, the minimum and maximum slip rates assigned to the sources using the references available (see the supplemental files); and *ID,* the fault number identifier.

| ID | Fault Sources | Historical Earthquakes | | | | | Instrumental Earthquakes | |
|----|---------------|------------|-------|-------|-------|-----|------------|-------|
| | | yyyy/mm/dd | $I_{Max}$ | $I_0$ | $M_w$ | sD | yyyy/mm/dd | $M_w$ |
| 1 | Lunigiana | 1481/05/07 | VIII | VIII | 5.6 | 0.4 | | |
| | | 1834/02/14 | IX | IX | 6.0 | 0.1 | | |
| 2 | North Apuane Transfer | 1837/04/11 | X | IX | 5.9 | 0.1 | | |
| 3 | Garfagnana | 1740/03/06 | VIII | VIII | 5.6 | 0.2 | | |
| | | 1920/09/07 | X | X | 6.5 | 0.1 | | |
| 4 | Garfagnana Transfer | | | | | | | |
| 5 | Mugello | 1542/06/13 | IX | IX | 6.0 | 0.2 | | |
| | | 1919/06/29 | X | X | 6.4 | 0.1 | | |
| 6 | Ronta | | | | | | | |
| 7 | Poppi | | | | | | | |
| 8 | Città di Castello | 1269 | | | 5.7 | | | |
| | | 1389/10/18 | IX | IX | 6 | 0.5 | | |
| | | 1458/04/26 | VIII-IX | VIII-IX | 5.8 | 0.5 | | |
| | | 1789/09/30 | IX | IX | 5.9 | 0.1 | | |
| 9 | M.S.M. Tiberina | 1352/12/25 | IX | IX | 6.3 | 0.2 | | |
| | | 1917/04/26 | IX-X | IX-X | 6.0 | 0.1 | | |
| 10 | Gubbio | | | | | | 1984/04/29 | 5.6 |
| 11 | Colfiorito System | 1279/04/30 | X | IX | 6.2 | 0.2 | 1997/09/26 | 5.7 |
| | | 1747/04/17 | IX | IX | 6.1 | 0.1 | 1997/09/26 | 6 |
| | | 1751/07/27 | X | X | 6.4 | 0.1 | | |
| 12 | Umbra Valley | 1277 | | VIII | 5.6 | 0.5 | | |
| | | 1832/01/13 | X | X | 6.4 | 0.1 | | |
| | | 1854/02/12 | VIII | VIII | 5.6 | 0.3 | | |
| 13 | Vettore-Bove | | | | | | 2016/10/30 | 6.5 |
| 14 | Nottoria-Preci | 1328/12/01 | X | X | 6.5 | 0.3 | 1979/09/19 | 5.8 |
| | | 1703/01/14 | XI | XI | 6.9 | 0.1 | | |
| | | 1719/06/27 | VIII | VIII | 5.6 | 0.3 | | |
| | | 1730/05/12 | IX | IX | 6.0 | 0.1 | | |
| | | 1859/08/22 | VIII-IX | VIII-IX | 5.7 | 0.3 | | |
| | | 1879/02/23 | VIII | VIII | 5.6 | 0.3 | | |
| 15 | Cascia-Cittareale | 1599/11/06 | IX | IX | 6.1 | 0.2 | | |
| | | 1916/11/16 | VIII | VIII | 5.5 | 0.1 | | |
| 16 | Leonessa | | | | | | | |
| 17 | Rieti | 1298/12/01 | X | IX-X | 6.3 | 0.5 | | |
| | | 1785/10/09 | VIII-IX | VIII-IX | 5.8 | 0.2 | | |
| 18 | Fucino | 1349/09/09 | IX | IX | 6.3 | 0.1 | | |
| | | 1904/02/24 | IX | VIII-IX | 5.7 | 0.1 | | |
| | | 1915/01/13 | XI | XI | 7 | 0.1 | | |
| 19 | Sella di Corno | | | | | | | |
| 20 | Pizzoli-Pettino | 1703/02/02 | X | X | 6.7 | 0.1 | | |
| 21 | Montereale | | | | | | | |
| 22 | Gorzano | 1639/10/07 | X | IX-X | 6.2 | 0.2 | | |
| | | 1646/04/28 | IX | IX | 5.9 | 0.4 | | |
| 23 | Gran Sasso | | | | | | | |
| 24 | Paganica | 1315/12/03 | VIII | VIII | 5.6 | 0.5 | 2009/06/04 | 6.3 |
| | | 1461/11/27 | X | X | 6.5 | 0.5 | | |
| 25 | Middle Aternum Valley | | | | | | | |
| 26 | Campo Felice-Ovindoli | | | | | | | |
| 27 | Carsoli | | | | | | | |
| 28 | Liri | | | | | | | |
| 29 | Sora | 1654/07/24 | X | IX-X | 6.3 | 0.2 | | |
| 30 | Marsicano | | | | | | | |
| 31 | Sulmona | | | | | | | |
| 32 | Maiella | | | | | | | |
| 33 | Aremogna C.Miglia | | | | | | | |
| 34 | Barrea | | | | | | 1984/05/07 | 5.9 |
| 35 | Cassino | | | | | | | |
| 36 | Ailano-Piedimonte | | | | | | | |
| 37 | Matese | 1349/09/09 | X-XI | X | 6.8 | 0.2 | | |

| | | | | | | | |
|---|---|---|---|---|---|---|---|
| 38 | Bojano | 1805/07/26 | X | X | 6.7 | 0.1 | | |
| 39 | Frosolone | 1456/12/05 | XI | XI | 7 | 0.1 | | |
| 40 | Ripabottoni-San Severo | 1627/07/30 | X | X | 6.7 | 0.1 | 2002/10/31 | 5.7 |
| | | 1647/05/05 | VII-VIII | VII-VIII | 5.7 | 0.4 | | |
| | | 1657/01/29 | IX-X | VIII-IX | 6.0 | 0.2 | | |
| 41 | Mattinata | 1875/12/06 | VIII | VIII | 5.9 | 0.1 | | |
| | | 1889/12/08 | VII | VII | 5.5 | 0.1 | | |
| | | 1948/08/18 | VII-VIII | VII-VIII | 5.6 | 0.1 | | |
| 42 | Castelluccio dei Sauri | 1361/07/17 | X | IX | 6 | 0.5 | | |
| | | 1560/05/11 | VIII | VIII | 5.7 | 0.5 | | |
| | | 1731/03/20 | IX | IX | 6.3 | 0.1 | | |
| 43 | Ariano Irpino | 1456/12/05 | | | 6.9 | 0.1 | | |
| | | 1962/08/21 | IX | IX | 6.2 | 0.1 | | |
| 44 | Tammaro | 1688/06/05 | XI | XI | 7 | 0.1 | | |
| 45 | Benevento | | | | | | | |
| 46 | Volturno | | | | | | | |
| 47 | Avella | 1499/12/05 | VIII | VIII | 5.6 | 0.5 | | |
| 48 | Ufita-Bisaccia | 1732/11/29 | X-XI | X-XI | 6.8 | 0.1 | | |
| | | 1930/07/23 | X | X | 6.7 | 0.1 | | |
| 49 | Melfi | 1851/08/14 | X | X | 6.5 | 0.1 | | |
| 50 | Irpinia Antithetic | | | | | | | |
| 51 | Irpinia | 1466/01/15 | VIII-IX | VIII-IX | 6.0 | 0.2 | 1980/11/23 | 6.8 |
| | | 1692/03/04 | VIII | VIII | 5.9 | 0.4 | | |
| | | 1694/09/08 | X | X | 6.7 | 0.1 | | |
| | | 1853/04/09 | IX | VIII | 5.6 | 0.2 | | |
| 52 | Volturara | | | | | | | |
| 53 | Alburni | | | | | | | |
| 54 | Caggiano-Diano Valley | 1561/07/31 | IX-X | X | 6.3 | 0.1 | | |
| 55 | Pergola-Maddalena | 1857/12/16 | | | 6.5 | | | |
| | | 1857/12/16 | | | 6.3 | | | |
| 56 | Agri | | | | | | | |
| 57 | Potenza | 1273/12/18 | VIII-IX | VIII-IX | 5.8 | 0.5 | 1990/05/05 | 5.8 |
| 58 | Palagianello | | | | | | | |
| 59 | Monte Alpi | | | | | | | |
| 60 | Maratea | | | | | | | |
| 61 | Mercure | 1708/01/26 | VIII-IX | VIII | 5.6 | 0.6 | 1998/09/09 | 5.5 |
| 62 | Pollino | | | | | | | |
| 63 | Castrovillari | | | | | | | |
| 64 | Rossano | 1836/04/25 | X | IX | 6.2 | 0.2 | | |

| | | | | | |
|---|---|---|---|---|---|
| 65 | Crati West | 1184/05/24 | IX | IX | 6.8 | 0.3 |
| | | 1870/10/04 | X | IX-X | 6.2 | 0.1 |
| | | 1886/03/06 | VII-VIII | VII-VIII | 5.6 | 0.3 |
| 66 | Crati East | 1767/07/14 | VIII-IX | VIII-IX | 5.9 | 0.2 |
| | | 1835/10/12 | X | IX | 5.9 | 0.3 |
| 67 | Lakes | 1638/06/08 | X | X | 6.8 | 0.1 |
| 68 | Fuscalto | 1832/03/08 | X | X | 6.6 | 0.1 |
| 69 | Piano Lago-Decollatura | | | | | |
| 70 | Catanzaro North | 1638/03/27 | | | 6.6 | |
| 71 | Catanzaro South | 1626/04/04 | X | IX | 6.1 | 0.4 |
| 72 | Serre | 1659/11/05 | X | X | 6.6 | 0.1 |
| | | 1743/12/07 | IX-X | VIII-IX | 5.9 | 0.2 |
| | | 1783/02/07 | X-XI | X-XI | 6.7 | 0.1 |
| | | 1791/10/13 | IX | IX | 6.1 | 0.1 |
| 73 | Vibo | | | | | |
| 74 | Sant'Eufemia Gulf | 1905/09/08 | X-XI | X-XI | 7 | 0.1 |
| 75 | Capo Vaticano | | | | | |
| 76 | Coccorino | 1928/03/07 | VIII | VII-VIII | 5.9 | 0.1 |
| 77 | Scilla | | | | | |
| 78 | Sant'Eufemia | 1894/11/16 | IX | IX | 6.1 | 0.1 |
| 79 | Cittanova-Armo | 1509/02/25 | IX | VIII | 5.6 | 0.4 |
| | | 1783/02/05 | XI | XI | 7.1 | 0.1 |
| 80 | Reggio Calabria | | | | | |
| 81 | Taormina | 1908/12/28 | XI | XI | 7.1 | 0.2 |
| 82 | Acireale | 1818/02/20 | IX-X | IX-X | 6.3 | 0.1 |
| 83 | Western Ionian | 1693/01/11 | XI | XI | 7.3 | 0.1 |
| 84 | Eastern Ionian | | | | | |
| 85 | Climiti | | | | | |
| 86 | Avola | | | | | |


Table 2 Earthquake-Source Association Adopted for Fault Sources. $I_{Max,}$ maximum
intensity; $I_0$, epicentral intensity; $M_w$, moment magnitude; and sD, standard deviation
of the moment magnitude. For references, see the supplemental files.