# Peer review of "model for peninsular Italy"

_Natural Hazards and Earth System Sciences, 2017_

## Referee Comment (RC2)

**Review of "Integrating faults and past earthquakes into a probabilistic seismic hazard model for peninsular Italy" by** Alessandro Valentini, Francesco Visini and Bruno Pace
Manuscript NHESS-2017-41

by Laurentiu Danciu
Swiss Seismological Service
ETH Zurich

General Comments

The manuscript provides a procedure to integrate active faults in a regional seismogenic source model for Italy. A database of active faults was compiled and fully parameterised for use together with observed seismicity (instrumental and historical) to forecast the spatial and temporal distribution of future seismicity. Earthquake recurrence models of the delineated active faults are model by two magnitude-frequency distributions: either a Characteristic Gaussian (CHG) or Truncated Gutenberg-Richter (TGR). Additionally, the seismicity off faults is described by a smoothed seismicity using a complete earthquake catalogue of the region. The two models are complementary not independent, thus the earthquake rates account for double-counting of earthquakes assigned to faults above specified threshold magnitude. Further, a novel weighting function to correct the earthquake rates in vicinity of fault sources is proposed and used. The resulting two seismic sources are eventually combined in a mixed source model representing the suitable activity rates in time and space. The authors conclude with a sensitivity analysis evaluating the impact of the two models of earthquake recurrence rates on the total seismic hazard.

The use of active faults in seismic hazard assessment has become extensive in the last decades due to efforts of data compilation and analysis. Active faults provides the information to extend the observational time of large magnitude earthquakes which often is not captured by the existing catalogues of observed seismicity. The current manuscript provides a step forward into this direction. The combination active faults and smoothed seismicity is not a novel procedure but rather state of practice. Overall, the manuscript is relatively well written, there are several misleading parts to be improved, highlighted in my detailed comments. The structure of the manuscript is consistent with the procedural steps and no major changes are required. The figures, tables and supplemental materials are clear and appropriated. There are some key references missing but this is not necessarily a criticism. The conclusions appear appropriate with the proposed procedure and analysed content. My comments follow the structure of the manuscript and summarised below:

1. First and foremost the authors should be clearly state that this is not an update of the seismic hazard model of Italy, and that the purpose of the study is to integrate the active faults in a hazard calculation. Moreover, the resulting seismogenic model presented in this study has limitations, such as the use only of shallow faults, but not the subduction and volcanic sources.

2. A definition of active fault in the context of the study must be introduced. The literature distinguishes between active faults in geological time, i.e. Quaternary or Neocene, capable of future reactivation. Moreover, the slip rate assumptions must be discussed. It is well accepted that large variability are associated with the slip-rate values, and some portion of slip-rate can be aseismic. Extension of this discussion must be introduced in the context of this study.

3. Further, the authors are aware of the 2013 European Seismic Hazard Model (ESHM13, Woesner et al 2015) developed within the SHARE Project. It might be worth discussing the two approaches side by side, as the ESHM13 is the first reference model to introduce active faults for Euro-Mediterranean Region.

4. There are several procedural steps that are not well explained in the document, such as the estimation of the activity rates for faults. Albeit, the main focus of the procedure is to implement active faults to seismic hazard, the activity rates are yet described as input to the FiSH code and the segment seismic moment conservation. In my opinion this is not enough. The key elements and assumptions for computing the activity rates of active faults needs more attention, supported with discussions of the sensitivity of the input parameters, i.e. the effect of slip rates to earthquake recurrence rates.

5. The role of each magnitude frequency distribution (MFD) for each fault is not clear as described in the current version. One might expect a logic tree of the two MFDs. This aspect needs to be emphasised in the introduction.

6. Maximum magnitude assigned to each fault based on empirical magnitude scaling relationships do not account for uncertainties of the fault size (subsurface length or area). From the current version of the manuscript it is not evident the error associated to the fault size in the fault dataset.

7. Also, one can argue that more recent magnitude scaling relationships can be used (e.g Leonardo et al 2010) but for those used, the role of aleatory uncertainty must be mentioned and quantified herein. The authors should describe the procedure implemented in the FiSH code because not everyone has access to that manuscript.

8. Five maximum magnitude values are described as being assigned to each fault. The way these five values are implemented in the final computational model is not clear. Are these values modelled in a logic tree?

9. A sensitivity analysis to the choice of the maximum magnitude may be necessary to explain the effect of maximum magnitude for the TRT. For the same slip rate increase of the maximum magnitude will result in a decrease of the recurrence of small events. This effect is due to the fact that the largest earthquake accounts for most of the seismic moment and this requires the subtraction of small events to maintain the seismic moment balance.

10. In a general way, the characteristic model implies a recurrence rate estimated on large past large-magnitude earthquakes recognised from past geological record and the time interval between events can be measured. How many of the faults have a geological record long enough to characterise the recurrence of the large magnitude events? In the current version of the manuscript the historical events are linked to the faults, thus the long-term representation of the fault activity is questionable.

11. Slip rates are averaged over successive geologically recognised earthquakes and prone to error in measurements, hence the uncertainties of the slip-rates needs to be quantified.

12. When combining active faults and background seismicity, it is mandatory a comparison of the seismic productivity (CHG and TRT) of the faults with the gridded seismicity in the vicinity of faults. Without such comparison it is difficult to assess the performance of the models.

13. Generally, evaluating the performance of seismogenic sources based on seismic hazard estimates is not recommended. The hazard estimates based on active faults only is misleading, as the active faults are incomplete in space, and not treated as independent models. Thus the model performance may be evaluated at the level of seismicity rates comparison, not for hazard estimates.

14. The authors should state clearly that a suitable seismogenic source model combines the active faults and the gridded seismicity as mixed model.

Section Specific Comments

L50:51: "In Europe, a working group…" In Europe, within the SHARE project (Giardini et al 2010) has introduced the use of active faults at the region level for the first time. I am surprised that the authors do not refer in their study to the fault source models for Italy, the DISS (Database of Individual Seismogenic Sources). What are the main similarities and differences between the two dataset? The authors may consider adding a reference and a discuss the two datasets to avoid confusion.

L63: 66 The uniform seismotectonic sources of the Italian hazard described by Stuchi et al (2011) are delineated considering the fault information where and when available. The more realistic pattern of ground motion due to faults it is questionable, because an area source delineated to describe a group of faults, it will produce a similar pattern with the individual faults. The major benefits of using the active faults is to extend the observational time to capture the recurrence of large magnitude events. The local pattern due to fault location might be controlled by other factors such as hanging wall, upper seismogenic depth, style of faulting. However, these effects are not evident if an inappropriate ground motion model is selected. Thus the seismic hazard pattern depends on both seismic source representation and ground motion models.

L72. The term models is misleading. A source model implies a complete source representation in space and time aimed at describing the seismogenic potential of the region. In the current context, the active faults are incomplete in space, they are not describing all the tectonics of the region - not volcanic, subduction or deep seismicity reported for the Italian territory. It has to be specified that these are individual seismic sources, but not independent models. The procedure proposed here is aiming at creating a "model" for an exercise of seismic hazard evaluation. Moreover, if the goal of the work is to provide a robust seismic hazard estimates, then the authors resolve the issues of model independence and completeness as well as to capture the epistemic uncertainties in the mixed source model.

L120: The time scale is a key aspect to evaluate the long-term representation of the seismic productivity of active faults. If a fault has moved in the recent geologically time , i.e Holocene, it might be considered as seismically active, if it moved in the far-off geologic time and has not moved again since then the fault might be judged to be an inactive fault. Hence, it might be of interest to specify the time scale and the definition of active faults on the present investigation. Yet, as mentioned before there is need to clarify the definition of fault activity or non activity.

L131:135. The sleep rate values for some faults are very low. Values of 0.3 mm/year are extremely low and the movement on these faults could also takes place as creep. Is the aseismic factor adjusting the slip rates? Are these slip-rates supported by historical seismicity observations, geological investigations and /or paleoseismicity studies?

L152: The name could be "Segmentation rules for delineating (or aggregating) fault sources"

L199: The role of aspect ratio must be discussed in greater extend than currently version. The extension along-strike dimensions of the faults seems to be constrained by this parameter.

L191: There are five Mmax values for each fault. How is the Mmax modelled in the hazard calculation?

L202: Introduce and explain the "segment seismic moment conservation"? The key assumptions and the input parameters of the recurrence rates must be described. Characterisation of the active faults is a key aspect of this approach, thus it requires more description. As mentioned before, the effect of maximum magnitude must be discussed. In the case of seismic moment balance, for a constant slip rate, the recurrence rates of small events are decreasing with increased magnitude.

L207:211: What is the rationale of the two MFDs? It is not evident why the two recurrence models are selected? In a general way, the characteristic earthquake is used to define an earthquake of a given magnitude and well identified recurrence time by geological evidences. The fault sources used here do not qualify for such model, for various reasons including the way they are constructed by linkage of various segments. A characteristic model will be appropriate for use on individual segment rather than a long composite fault. See discussions of Kagan (1993), that clearly states that the evidence of the characteristic earthquake hypothesis can be explained either by statistical bias or statistical artifact. Thus, it will be of great interest for the readers to specify the assumptions for the two MFDs.

L278: the number of Voronoi polygons is not clear to me. There are 3 to 50 polygons across the entire region? Each polygon is tectonic dependent? Please clarify.

L286: Who is parametrised the depth and the maximum magnitude for gridded seismicity? Are these parameters treated as aleatory or epistemic?

L382: For the purpose of an exercise one GMPE might have been justified. However, the focus of the study should be the comparison of the earthquake recurrence rates not the hazard estimates.

**Recommendation: Accept for publication with major revision**

---

## Referee Comment (RC1) · K. Vanneste (Referee) · 22 Mar 2017

**Review of manuscript NHESS-2017-41 "Integrating faults and past earthquakes into a probabilistic seismic hazard model for peninsular Italy" by Alessandro Valentini, Francesco Visini & Bruno Pace**

**Main comments**

This manuscript describes an approach to model seismic hazard in Italy using a combination of active fault data and gridded seismicity based on the instrumental and historical earthquake catalog. A database of active faults has been compiled, and important historical earthquakes have been assigned to their causative faults. Two models are considered for the magnitude-frequency distributions (MFDs) of the faults, either a truncated Gutenberg-Richter (TGR) MFD or a characteristic Gaussian (CHG) MFD. The gridded source model accounts for off-fault seismicity, and its MFD is computed in a way that it is complementary to the MFD of the fault source model (using a threshold magnitude, avoiding double-counting of earthquakes assigned to faults, and an additional weighting function that reduces gridded seismicity in the vicinity of faults). The authors explore the impact of the two MFD models, as well as the contribution of fault sources and gridded seismicity to the total hazard. They also define a preferred source model, in which the most appropriate MFD model for each fault is selected.

The approach to model fault sources is state of the art, and the integration of fault sources and gridded seismicity contains some innovative elements. The manuscript is mostly well written (with some exceptions, which are pointed out in the detailed comments below), the figures are clear, and the references are appropriate. The conclusions are supported by the results.

However, a number of improvements need to be made before the manuscript can be published. Below, I have listed a number of detailed comments. I summarize my main comments here:

- A major shortcoming is that the paper does not contain any reference to other published fault source models for Italy, notably DISS (Database of Individual Seismogenic Sources, http://diss.rm.ingv.it/diss/). At the very least, the authors should indicate how their fault source model relates to DISS, and what are the main differences (concepts and/or data).
- Although the authors refer to the SHARE project, and even use certain aspects of it, they do not compare their results to the fault-based hazard map (FSBG model) created in this project.
- Similarly, although a comparison with the current national hazard map is described in general terms, this comparison is not shown.
- In my opinion, it is also essential to show the summed MFDs of the different source models, and comparing those to each other and to the observed MFD based on the full catalog. Without this information, it is not possible to evaluate the performance of their model. Notably, it is indicated that the rate of M 5.5-6.0 earthquakes in the TGR end member is higher than in the CHG end member, but this is not shown.
- I have some doubt whether maximum magnitudes are correctly modelled, as it is indicated at some point that an earthquake assigned to a fault could have a magnitude larger than the magnitude range in the MFD for that fault, which should not be allowed.

- To improve clarity, the authors should more clearly explain in advance what they intend to do. Two main cases are:
  - They first describe the fault-source model and the distributed source model, and only later explain that these are not independent models, but are complementary, together accounting for all seismicity in Italy;
  - They first show hazard maps produced with the TGR and CHG MFD models, but only later explain that these are two end members, and that their preferred model is the Mixed model, in which a particular MFD model is assigned to each fault.

**Recommendation**

I recommend that this manuscript can be published after moderate revision.

**Detailed comments**

**Abstract**

L. 11: "many of active faults" → "many active faults"

L. 13-14: "modelled on a number of seismotectonic zones and the occurrence of earthquakes is assumed to be uniform" → "modelled using a number of seismotectonic zones in which the occurrence of earthquakes is assumed to be uniform";

L. 24-25: "earthquakes" → "magnitude-frequency distributions" (2x)

L. 30: "the spatial pattern of our model is far more detailed" → "the spatial pattern of the hazard maps obtained with our model is far more detailed". Unfortunately, this is not demonstrated in the paper, as there is no direct comparison with other hazard maps.

**1. Introduction**

L. 42: "in seismic hazard estimations": may be left out, it is obvious from the previous lines.

L. 52: "Combining seismic hazards from active faults with background sources" → "Combining active faults with background sources". I also note that the plural "seismic hazards" is used in other places in the manuscript, but it should be singular, as the paper deals with only one type of seismic hazard, namely ground-motion seismic hazard.

L. 58: Add reference for the SHARE FSBG model.

L. 62: "uniform on" → "uniform in"

L. 65-66: "…, obtaining more detailed and possibly more realistic patterns of ground motion, in order to improve the reliability …" → "…, in order to obtain more detailed and possibly more realistic patterns of ground motion, and to improve the reliability …"

L. 69: "can also be give insights" → "can also give insights"

**2.1 Fault Source Model**

L. 90-91: Add degree symbols to all dip and rake values.

L. 92: "thrust faults could be considered in a future study": Is there a particular reason for not including thrust faults in the present study? And for which areas in Italy will this have the largest impact?

L. 93: "seismogenic thickness" → "seismogenic layer" (a thickness has no upper and lower boundaries)

L. 101-102: "Slip rates control fault-based seismic hazards … and provide a time scale …": Strange phrasing. Slip rates do not provide a time scale. I'm not sure whether the authors mean to say that slip rates may be measured over different time scales or that slip rates may vary through time or both.

L. 112-124: This paragraph discusses slip rate variability through time, and states that slip rates have been determined for different time scales. However, (1) it is not clear how this time variability is handled in this study (it is not mentioned anymore further in the paper), and (2) Table 1 only lists minimum and maximum slip rates, without indication of the corresponding time scale. Is the time scale the same for all faults in this table?

L. 141: "the function with the lowest log-likelihood": Shouldn't this be the highest log-likelihood? Usually, one seeks the maximum likelihood, not the minimum likelihood.

L. 145-150: Is this an appropriate way to determine the overall standard deviation of the slip rate distribution in an area? I think it would be more appropriate to apply the Central Limit Theorem. If you consider each fault slip rate (x) as a sample from a population with mean μ and standard deviation σ, then μ can be found as $\mu_{\bar{x}}$ (mean value of the sample means), and σ as $\sqrt{n}\sigma_{\bar{x}}$ (with n the number of samples and $\sigma_{\bar{x}}$ the standard deviation of the sample means).

L. 160: "a fault source is …" → "a fault source is considered as …"

L. 166-169: there seems to be overlap between criterion ii (sharp bends) and criterion iv (bending ≥ 60°).

L. 171: "seismogenic thickness" → "changes in seismogenic thickness"

L. 180: "thinnest ST" → "smallest ST". Can you comment on the small ST value of 2.5 km? Is this in a volcanic zone?

L. 181: "Observed maximum magnitude data have been assigned to 47 fault sources". Is this based on Table 2?

L. 197-198: "a value that corresponds to the maximum observed magnitude (Mobs)". I'm not convinced it is correct to consider Mobs as one of the possible Mmax values, and treat it the same as the other estimations. In fact, the only thing we know for sure about Mmax is that it cannot be lower than Mobs. For that reason, Mobs is often used as a lower truncation of Mmax distributions (e.g.,

EPRI method for Stable Continental Regions). Not doing this can have strange consequences, as in lines 442-444, where it is stated "If an earthquake assigned to a fault source has a magnitude lower or higher than the bell curve of the CHG model distribution, …". However, the second case (observed magnitude higher than modelled Mmax distribution) should not be allowed in the PSHA model!

L. 199: "modifying the along-strike dimension if the rupture length exceeds the length predicted by the aspect ratio relationships". This is not very clear. Maybe rephrase as "reducing the fault length if the aspect ratio (W/L) is smaller than indicated by the relation between aspect ratio and rupture length for observed earthquake ruptures in the Abruzzo (Peruzza & Pace, 2002)".

L. 202: "we use the criterion of "segment seismic moment conservation"": is this a criterion or a concept, and can you briefly describe what it implies?

L. 206-207: "we use two magnitude-frequency distributions" → "we use two magnitude-frequency distribution models". I also recommend introducing the acronym MFD here, as the term is used frequently in the remainder of the manuscript.

L. 208: "Gaussian bell curve centred on the Mmax": Perhaps it is worth mentioning that this Gaussian curve applies to the incremental MFD values, not to the cumulative MFD values that are shown in Fig. 2c.

L. 209-211: It is not explained how the a- and b-values are determined for each fault when the TGR model is used. I assume this is done with the FiSH code, but it would be good to briefly describe the underlying concept (relation with slip rate).

L. 213: "the two above described magnitude-frequency distributions" → "the two MFD models described above".

**2.2 Distributed Source Model**

L. 218: "a number of earthquakes in the historical catalogue": Perhaps "important earthquakes"?

L. 228: "events not considered the mainshock" → "events not considered as mainshocks".

L. 233-234: "If the causative source of an earthquake is known, the impact of that earthquake does not need to be included in the seismicity smoothing process" → "If the causative fault of an earthquake is known, that earthquake does not need to be included in the seismicity smoothing procedure". It should be explicitly mentioned before that the fault and distributed source models are conceived as complementary source models, not as alternative source models (competing models in a logic tree). In the latter case, they should be independent.

L. 244: "where $n$ is the cumulative rate of earthquakes $ni$" → "where $n_i$ is the cumulative rate of earthquakes"

L. 263: I think the * symbol in the equation should be left out. If I understand correctly, rather than a multiplication, $\lambda(i_x, i_y)$ represents the seismicity rate in grid cell $(i_x, i_y)$.

L. 265: "jy" → "$i_y$"

L. 276-278: I don't understand the description of the Voronoi partition procedure: if the Italian territory is divided in a grid with 0.05° lon/lat spacing, then how can the number of grid cell centres be varied? Perhaps the centres of the grid cells represent the possible centres of Voronoi polygons, and you vary the number of Voronoi polygons from 3 to 50, for each case drawing 1000 random subsets of Nv grid cell centres?

L. 297: "β = 2/3 b": I think this should be "$\beta = b.\ln(10)$", which is ~2.3 b.

**2.3 Combining fault and distributed sources**

L. 299-300: It would be better to describe this concept before the two source model components are described (see general remark).

L. 307: Add some statement that this assumption is explained in more detail in the following paragraphs.

L. 311: "located across strike" → "arranged across strike"

L. 315: "neighbour faults" → "neighbouring faults"

L. 322: "has also shown similar" → "has shown similar"

L. 324-325: "from … to … to…" → "from … over … to …"

L. 330: "the spatial distribution … are" → "the spatial distribution … is"

L. 333: "appear to control to" → "appear to control"

L. 336: "Earthquakes" → "Large earthquakes"

L. 338-340: Is this valid for all types of faults or only for dip-slip faults?

L. 342: "distance along strike": I think this should be "across strike".

L. 352-353: "considerations to source models for fault sources and distributed seismicity" → "considerations to combine fault and distributed seismicity source models"

L. 360: Perhaps add that it is a linear function.

L. 363: Write the equation more completely:

$$Pe = \begin{cases} 0 \ (d \leq 1km) \\ \dfrac{1}{d} \ (1km < d \leq d_{max}) \\ 1 \ (d > d_{max}) \end{cases}$$

However, there is still a problem with the second line, which does the opposite of what is intended (going to 1 as d increases): instead of 1/d it should be $d/d_{max}$…

L. 366-367: What is the rationale for varying $d_{max}$ in function of slip rate?

L. 369-371: This is hard to understand. Maybe rephrase as "Because we considered two fault source models, one using only TGR MFDs and the other only CHR MFDs, and because the MFDs of distributed seismicity grid points in the vicinity of faults are modified with respect to the MFDs of these faults, we also obtain two different models of distributed seismicity."

In my opinion, it is also necessary at this point to show the summed MFDs of the different (sub)models, i.e. summed MFD of the TGR fault source model, of the CHR fault source model, of the TGR distributed source model, of the CHR distributed source model, and of the combined TGR and CHR source models.

L. 380: "in future" → "in the future".

**3. Results and discussion**

L. 382: "designed under the traditional Poisson hypothesis": Rephrase

L. 386: "well-known": this is not the most relevant property for choosing OpenQuake. Perhaps widely used, open-source, tested, …?

L. 390: "Bindi et al. 2014" → "Bindi et al. (2014)"

L. 391: "all GMPE" → "all GMPEs"

L. 401: "estimated by" → "obtained with"

L. 402: Explain more explicitly that the TGR and CHG fault source models are end members that are only used to explore the epistemic uncertainty, and that in the preferred fault source model a choice is made between the two MFD models for each fault.

L. 403-404: "Although both models have the same amount of seismic moment release": this has not been demonstrated.

L. 405-406: "all faults exhibit a 10% probability of exceedance in 50 years": Incorrect phrasing → "all faults contribute significantly to the seismic hazard".

L. 409-411: "The rates of earthquakes with magnitudes between 5.5 and approximately 6, …, are generally higher in the TGR model than in the CHG model": Please demonstrate by showing the summed MFDs.

L. 416-417: "threshold" → "lower value" (2x)

L. 424: "the fault source models contribute to …" → "the fault source models contribute most to …"

L. 434: "hazard maps with PGAs" → "hazard maps for PGA"

L. 436-437: "applying a magnitude frequency distribution to each fault" → "selecting the most appropriate MFD model (TGR or CHG) for each fault".

L. 443: "a magnitude lower or higher than the bell curve" → "a magnitude lower or higher than the magnitude range in the bell curve". See also my remark at lines 197-198: a higher magnitude should not be possible!

L. 448: "have been not yet mapped" → "have not yet been mapped".

L. 448: "The magnitude-frequency distribution" → "The MFD model".

L. 455: "sources that have been modelled using a magnitude-frequency distribution": all sources are modelled using an MFD → "sources that have been modelled using one or the other MFD model".

L. 462: "a particular magnitude-frequency distribution" → "a particular MFD model".

L. 465: "higher than those the CHG" → "higher than those of the CHG source model".

L. 467: "seismicity rate model" → "MFD model".

L. 468-471: It has not been explained exactly how the TGR MFDs have been constructed. See my remark at lines 209-211.

L. 475: "generally decrease with increases in the exceedance probability" → "generally decreases with increasing exceedance probability".

L. 505: Perhaps replace "TGR model" with a brief description like you do for the CHG model in the following line.

L. 513: "tests to the compatibility" → "tests of the compatibility".

L. 524: "The strength of our approach is" → "The strength of our approach lies in".

L. 527: "seismic hazards" → "seismic hazard".

**4. Conclusions**

L. 538-539: "110 faults with 86 fault sources" → "110 faults grouped into 86 fault sources".

L. 542-543: "two magnitude-frequency distributions" → "two MFD models".

L. 552: "overlapping the magnitude-frequency distribution" → "overlapping with the MFD".

L. 558-559: "pattern similar to that of the current national maps at the national scale, but some significant differences in hazard are present at the regional-to-local scale": this has not been discussed in the main text. It would be instructive to show both maps side by side and describe the comparison in some more detail in §3.

L. 560: "different magnitude-frequency distributions" → "different MFD models".

L. 563-565: See my comment for lines 409-411. It would also be interesting to compare the summed MFDs to the observed MFD based on the full catalog, to see which of the two MFD models is closer to the observations in this particular magnitude range (M 5.5 to ~6.0).

L. 568: "high values of probability of exceedance" → "high probabilities of exceedance".

L. 569: "equal the fault model one" → "equals that of the fault model".

L. 570-571: "applying a magnitude-frequency distribution to each fault" → "selecting the most appropriate MFD model for each fault".

**Figure captions**

Fig. 2, L. 824: "modified by" → "modified from"?

Fig. 2, L. 829-830: "the maximum subsurface fault length (MRLD, red line) and maximum rupture area (MRA, cyan line) …" → "MRLD (red line) and MRA (cyan line) correspond to estimates based on maximum subsurface fault length and maximum rupture area …"

Fig. 3, L. 841: "in correspondence of" → "corresponding to"

Fig. 6, L. 855: "versus c" → "versus correlation distance c"

Fig. 9 : Explain acronym "poe"

Fig. 12: How are the contributions of the component source models computed? The perfect symmetry between the contributions of the fault source model and the distributed source model gives me the impression that they do not correspond to the contributions one would obtain from a deaggregation.

---

## Author Response (AR1)

Ref: NHESS-2017-41

Dear Editor,

First of all, we warmly thank you for constructive comments. We followed your suggestions to revise the manuscript.

Regarding the two main points you highlighted:

- *the difference of your compilation with respect to DISS database (first point in Main Comments by RC1; L50-51 in Section Specific Comments by RC2).*
  We improved the manuscript (lines 123-149), giving an explanation of our choice about the fault database. In particular, we analysed the individual sources included in the DISS and spotted some issues that include: (i) the lack of updating of the geological information of some individual sources and (ii) the nonconformity between the input data used by DISS in Boxer and the latest historical seismicity (CPTI15) and macroseismic intensity (DBMI15) publications.
  Thus, we preferred to performed a full review of the fault database, compiling a fault source database as a synthesis of works published over the past twenty years, using all updated and available geological, paleoseismological and seismological data (see the supplemental files for a complete list of references).

- *the rationale and consequences of using end-member MFD models (L402 on by RC1;L207:211 by RC2).*
  We explained this in the introduction at L64-83. As we know, the choice of the "appropriate" MFD for each fault source is a difficult task because palaeoseismological studies are scarce, and it is often difficult to establish clear relationships between mapped faults and historical seismicity. Today, the discussion is still open and far from being solved with the available observations, including both seismological and/or geological/paleoseismological observations. What we did in this work, was to adopt two widely-used MFDs, a characteristic Gaussian model and a Truncated Gutenberg-Richter model, to explore the epistemic uncertainties. Finally, we considered also a Mixed model as a so-called "expert judgement" model. Obviously, this approach does not solve the issue, and the choice of MFD remains an open question in fault-based PSHA

Moreover, we used the last update version of the CPTI15, so some figures have been updated (Fig. 4,5,9,11,12 and 13).

Finally, the manuscript has been edited by mother-tongue American Journal Experts (AJE, www.aje.com) for proper English grammar and style.

Sincerely,

Alessandro Valentini (Corresponding author)

**Review of manuscript NHESS-2017-41 "Integrating faults and past earthquakes into a probabilistic seismic hazard model for peninsular Italy" by Alessandro Valentini, Francesco Visini & Bruno Pace**

**Main comments**

This manuscript describes an approach to model seismic hazard in Italy using a combination of active fault data and gridded seismicity based on the instrumental and historical earthquake catalog. A database of active faults has been compiled, and important historical earthquakes have been assigned to their causative faults. Two models are considered for the magnitude-frequency distributions (MFDs) of the faults, either a truncated Gutenberg-Richter (TGR) MFD or a characteristic Gaussian (CHG) MFD. The gridded source model accounts for off-fault seismicity, and its MFD is computed in a way that it is complementary to the MFD of the fault source model (using a threshold magnitude, avoiding double-counting of earthquakes assigned to faults, and an additional weighting function that reduces gridded seismicity in the vicinity of faults). The authors explore the impact of the two MFD models, as well as the contribution of fault sources and gridded seismicity to the total hazard. They also define a preferred source model, in which the most appropriate MFD model for each fault is selected.

The approach to model fault sources is state of the art, and the integration of fault sources and gridded seismicity contains some innovative elements. The manuscript is mostly well written (with some exceptions, which are pointed out in the detailed comments below), the figures are clear, and the references are appropriate. The conclusions are supported by the results.

However, a number of improvements need to be made before the manuscript can be published. Below, I have listed a number of detailed comments. I summarize my main comments here:

- A major shortcoming is that the paper does not contain any reference to other published fault source models for Italy, notably DISS (Database of Individual Seismogenic Sources, http://diss.rm.ingv.it/diss/). At the very least, the authors should indicate how their fault source model relates to DISS, and what are the main differences (concepts and/or data).

- *We will add these information in the section 2.1 "Fault Source Model" at line 84: "Although for the Italian territory there is already a database that contains the results of the investigations of the active tectonics during the past 20 years (Database of Individual Seismogenic Sources, DISS, http://diss.rm.ingv.it/diss/), made by three main categories of seismogenic sources: individual seismogenic sources, seismogenic areas, macroseismic sources, it does not work well to elaborate a PSHA model using individual seismogenic sources, as in this work. In*

*fact, the DISS Authors (Basili et al., 2008) say that the individual seismogenic sources database cannot guarantee the completeness of the sources themselves and are not meant to comprise a complete input dataset for probabilistic assessment of seismic hazard. For this reason, we are not restricted to just use of the DISS, but trough a synthesis of published works over the last twenty years (see supplements for complete references) we defined a database as complete as possible, in terms of individual seismogenic sources, and parameters to have input dataset for PSHA."*

- Although the authors refer to the SHARE project, and even use certain aspects of it, they do not compare their results to the fault-based hazard map (FSBG model) created in this project.
- Similarly, although a comparison with the current national hazard map is described in general terms, this comparison is not shown.

- *We attach in supplement a figure (Figure S1) showing the comparison among SHARE (FSBG) model, the current Italian national seismic hazard map (MPS04) and our model (Mixed model), using the same GMPE's. The new figure we'll be included in the manuscript, at Chapter 3. The figure shows how the impact of our fault sources input is more evident then the FSBG-Share model (the branch using fault sources and background) and the comparison with MPS04 confirm a similar pattern, but with some significant differences at the regional-to-local scale.*

- In my opinion, it is also essential to show the summed MFDs of the different source models, and comparing those to each other and to the observed MFD based on the full catalog. Without this information, it is not possible to evaluate the performance of their model. Notably, it is indicated that the rate of M 5.5-6.0 earthquakes in the TGR end member is higher than in the CHG end member, but this is not shown.

- *Thanks for your suggestion. We attach in supplement a figure (Figure S2) showing and comparing the summed MFD's of the fault source inputs (TGR, CHG, Mixed), the distributed source input, the total model (distributed + fault) and the CPTI15 catalogue, for Apennines and surrounding areas. This new figure highlights also the differences in the rate of M 5.5-6.0 earthquakes between TGR and CHG model. The new figure we'll be included in the revised version of the manuscript.*

- I have some doubt whether maximum magnitudes are correctly modelled, as it is indicated at some point that an earthquake assigned to a fault could have a magnitude larger than the magnitude range in the MFD for that fault, which should not be allowed.

- *What we wrote at lines 442-444 was a mistake: we never have a magnitude larger than the magnitude range in the MFD for a fault. So, the right sentence is: "if an earthquake*

*assigned to a fault source (see Table 2 for earthquake-source associations) has a magnitude lower than the magnitude range in the bell curve of the CHG model distribution, the TGR model is applied to that fault source." We'll update the text in the revised version of the manuscript.*

- To improve clarity, the authors should more clearly explain in advance what they intend to do. Two main cases are:

    - They first describe the fault-source model and the distributed source model, and only later explain that these are not independent models, but are complementary, together accounting for all seismicity in Italy;

    - *You are right, we'll write in the revised version of the manuscript, as you suggest, that the two models are not independent but complementary, both in magnitude and frequency distribution. Moreover, as also suggested by the second reviewer, the fault-source and distribute source are not 'models' s.s., so we'll rename them as 'input'.*

    - They first show hazard maps produced with the TGR and CHG MFD models, but only later explain that these are two end members, and that their preferred model is the Mixed model, in which a particular MFD model is assigned to each fault.

    - *Thanks for your suggestion. We'll be more clear in the introduction of the revised version of the manuscript that we consider the TGR and CHG MFD models as end members, and the Mixed model as a sort of an "expert judgment" model, useful for comparison analysis.*

**Detailed comments**

**Abstract**

L. 30: "the spatial pattern of our model is far more detailed" → "the spatial pattern of the hazard maps obtained with our model is far more detailed". Unfortunately, this is not demonstrated in the paper, as there is no direct comparison with other hazard maps.

*We'll show the differences between our approach and the others by a figure (Figure S1 in the supplement) where we compare our results with SHARE (FSBG) model and the current national hazard map (MPS04), using the same GMPE's. The new figure we'll be included in the manuscript, at Chapter 3.*

**1. Introduction**

L. 52: "Combining seismic hazards from active faults with background sources" → "Combining active faults with background sources". I also note that the plural "seismic hazards" is used in other places in the manuscript, but it should be singular, as the paper deals with only one type of seismic hazard, namely ground-motion seismic hazard.

*Thanks for your suggestion: we'll remove the plural.*

**2.1 Fault Source Model**

L. 92: "thrust faults could be considered in a future study": Is there a particular reason for not including thrust faults in the present study? And for which areas in Italy will this have the largest impact?

*We decided to not include thrust faults in the present study because for them we have to solve some problems, mainly connected to the definition of individual seismogenic source, not yet solved in Italy for such kind of structure. For example, for thrust faults we do not have a good knowledge of the geological slip rate as for normal active fault, we need to introduce a different way to make the segmentation and different segmentation rules, and maybe there is need to consider them as complex sources in OpenQuake. The areas in Italy where we think they will have the largest impact are NE sector of the Alps, Po Valley, offshore sector of the central Adriatic Sea and SW Sicily. In this paper we want to focus on the impact of the integration of faults and earthquakes data, without the assumption to be complete in terms of individual seismogenic source database, but on the contrary suggesting a way to integrate two incomplete database in the best way, without throwing data. We will add in the manuscript a phrase explaining our choices.*

L. 101-102: "Slip rates control fault-based seismic hazards … and provide a time scale …": Strange phrasing. Slip rates do not provide a time scale. I'm not sure whether the authors mean to say that slip rates may be measured over different time scales or that slip rates may vary through time or both.

*Thanks for your suggestion: we will rephrase this sentence as: "Slip rates control fault-based seismic hazard … and reflect the velocity of the mechanisms operating during continental deformation ..."*

L. 112-124: This paragraph discusses slip rate variability through time, and states that slip rates have been determined for different time scales. However, (1) it is not clear how this time variability is handled in this study (it is not mentioned anymore further in the paper), and (2) Table 1 only lists minimum and maximum slip rates, without indication of the corresponding time scale. Is the time scale the same for all faults in this table?

*Thanks for your suggestion: this paragraph is not clear and so we will re-write it in the revised version of the manuscript. The aim is to highlight that we are conscious of the problem of the possible slip term variability through time, but we are able to solve it with the data in our database. The assumption we do is that we use the minimum and maximum values of slip rate, determined in different ways and different time scales (see the numerous neotectonics, palaeseismological and seismotectonics cited papers), to calculate a mean value that we assume as representative of the long term behaviour (about last 15 ka for the Apennines).*

L. 141: "the function with the lowest log-likelihood": Shouldn't this be the highest log-likelihood? Usually, one seeks the maximum likelihood, not the minimum likelihood

*Yes, it is the highest log-likelihood. We'll correct in the revised version of the manuscript.*

L. 145-150: Is this an appropriate way to determine the overall standard deviation of the slip rate distribution in an area? I think it would be more appropriate to apply the Central Limit Theorem. If you consider each fault slip rate (x) as a sample from a population with mean μ and standard deviation σ, then μ can be found as $\mu_x$(mean value of the sample means), and σ as $\sqrt{n}\ \sigma_x$ (with n the number of samples and  the standard deviation of the sample means).

*Thanks for this suggestion. We applied the Central Limit Theorem for the three areas and the standard deviation is 0.11, 0.33 and 0.83 for Northern, Central-Southern, and Calabria-Sicilian area respectively. Instead using our approach we obtained 0.25, 0.29, and 0.35 for the three areas respectively. The obtained values for Northern and Calabrian-Sicilian areas are a little bit different, we think because the sample population is not enough large to apply the Central Limit Theorem; in fact n has to be > 30, while in our case n is equals 20 and 14 for the Northern and Calabrian-Sicilian area respectively. For this reason we decided to leave the standard deviation computed with our suggested approach.*

L. 166-169: there seems to be overlap between criterion ii (sharp bends) and criterion iv (bending ≥ 60°).

*Yes, you are right, we wrote in a wrong way. The ii criterion is "(ii) intersections with cross structures (often transfer faults) extending 4 km along strike….". We will correct the manuscript.*

L. 180: "thinnest ST" → "smallest ST". Can you comment on the small ST value of 2.5 km? Is this in a volcanic zone?

*No, it is not in a volcanic zone. The value of 2.5 km is due to the presence of "Alto Tiberina Fault". It is a structure well known in literature: a low angle normal fault acts to detachment*

*for the seismogenic faults located in the hanging-wall. We'll add a sentence in the revised manuscript at line 180 as: "with the thinnest ST is Monte Santa Maria Tiberina (id 9, ST = 2.5 km) due to the presence of east-dipping low angle normal fault, the Alto-Tiberina Fault (Boncio et al., 2000), located few kilometres west of the is Monte Santa Maria Tiberina fault."*

L. 181: "Observed maximum magnitude data have been assigned to 47 fault sources". Is this based on Table 2?

*Yes, it is. We have written it in the manuscript at line 181:" Observed maximum magnitude data have been assigned to 47 fault sources (based on Table 2)".*

L. 197-198: "a value that corresponds to the maximum observed magnitude (Mobs)". I'm not convinced it is correct to consider Mobs as one of the possible Mmax values, and treat it the same as the other estimations. In fact, the only thing we know for sure about Mmax is that it cannot be lower than Mobs. For that reason, Mobs is often used as a lower truncation of Mmax distributions (e.g.,EPRI method for Stable Continental Regions). Not doing this can have strange consequences, as in lines 442-444, where it is stated "If an earthquake assigned to a fault source has a magnitude lower or higher than the bell curve of the CHG model distribution, …". However, the second case (observed magnitude higher than modelled Mmax distribution) should not be allowed in the PSHA model.

*We partially agree with you. In some cases the observed Magnitude (Mobs) is useful to better constrain the potentiality of an individual seismogenic source, as some examples like Irpinia Fault (id 51 in the database) where the 1980 earthquake helps to better constrain the Mmax computed by only scaling relationships. Obviously it is important to avoid cases where there is an inconsistency between the fault geometry and the observed magnitude, and so our rationale was:*
   1) *we calculate the maximum expected magnitude (Mmax1), and the relative uncertainties, using only the scaling relationships (detail in Pace et al., 2016, FiSH paper);*
   2) *we compared the observed magnitude of the associated earthquakes in the catalogue (Mobs), and if the Mobs is contained in the range Mmax1 +-1 standard deviation, we consider the Mobs recalculating the Mmax (Mmax2) and the new uncertainties;*
   3) *if the Mobs is lower then Mmax1 we consider a GR behaviour for the source, without using the Mobs in the Mmax2 calculation;*
   4) *if the Mobs is larger then Mmax1 we review the fault geometry or the earthquake source association.*
*We'll improve the manuscript in order to better explain our rationale.*

L. 199: "modifying the along-strike dimension if the rupture length exceeds the length predicted by the aspect ratio relationships". This is not very clear. Maybe rephrase as "reducing the fault length if the aspect ratio (W/L) is smaller than indicated by the relation between aspect ratio and rupture length for observed earthquake ruptures in the Abruzzo (Peruzza & Pace, 2002)".

*Thanks for this suggestion. We'll rephrase as you suggest.*

L. 202: "we use the criterion of "segment seismic moment conservation"": is this a criterion or a concept, and can you briefly describe what it implies?

*We agree that a brief description could be useful. At line 203 we'll add a sentence as: "… which divides the seismic moment that corresponds to $M_{max}$ by the moment rate given a slip rate:*

$$T_{mean} = \frac{1}{Char\_Rate} = \frac{10^{1.5 M_{max} 9.1}}{\mu V L W}$$

*where $T_{mean}$ is the mean recurrence time in years, Char_Rate is the annual mean rate of occurrence, $M_{max}$ is the computed mean maximum magnitude, μ is the shear modulus, V is the average long-term slip rate, and L and W are the geometrical parameters of the fault, along-strike rupture length and down dip width respectively."*

L. 206-207: "we use two magnitude-frequency distributions" → "we use two magnitude-frequency distribution models". I also recommend introducing the acronym MFD here, as the term is used frequently in the remainder of the manuscript.

*Thanks for the suggestion: we'll introduce the acronym MFD in the abstract and replaced all "magnitude-frequency distribution" in the manuscript.*

L. 208: "Gaussian bell curve centred on the Mmax": Perhaps it is worth mentioning that this Gaussian curve applies to the incremental MFD values, not to the cumulative MFD values that are shown in Fig. 2c.

*We'll modify the sentence into: "symmetric Gaussian bell curve (applied to the incremental MFD values) centred on the Mmax of each fault, with a range of magnitudes equal to 1-sigma".*

L. 209-211: It is not explained how the a- and b-values are determined for each fault when the TGR model is used. I assume this is done with the FiSH code, but it would be good to briefly describe the underlying concept (relation with slip rate).

*We'll add a phrase to better explain how the a- and b-values have been determined: "For MFD, the b-value is constant and equal to 1.0 for all faults, obtained by the interpolation of the earthquakes in the CPTI15 catalogue, as the events on the single sources are*

*insufficient for statistics. However the a-values have been computed by Activity Rate FiSH code, balancing the total expected seismic moment rate with the seismic moment rate that was obtained by the pair $M_{max}$ and $T_{mean}$, evaluated by the fault geometry and the slip rate of each individual source (details in Pace et al., 2016)."*

**2.2 Distributed Source Model**

L. 233-234: "If the causative source of an earthquake is known, the impact of that earthquake does not need to be included in the seismicity smoothing process" → "If the causative fault of an earthquake is known, that earthquake does not need to be included in the seismicity smoothing procedure". It should be explicitly mentioned before that the fault and distributed source models are conceived as complementary source models, not as alternative source models (competing models in a logic tree). In the latter case, they should be independent.

*Thanks for this suggestion. We'll better explain before that we consider the two source models complementary but not alternative, and so not independent.*

L. 263: I think the * symbol in the equation should be left out. If I understand correctly, rather than a multiplication, $\lambda(i_x, i_y)$ represents the seismicity rate in grid cell $(i_x, i_y)$

*Yes, you are right, it was a typo.*

L. 276-278: I don't understand the description of the Voronoi partition procedure: if the Italian territory is divided in a grid with 0.05° lon/lat spacing, then how can the number of grid cell centres be varied? Perhaps the centres of the grid cells represent the possible centres of Voronoi polygons, and you vary the number of Voronoi polygons from 3 to 50, for each case drawing 1000 random subsets of Nv grid cell centres?

*To be more clear we'll modify the manuscript as:"… the Voronoi tessellation of space without tectonic dependency. The whole Italian territory has been divided into a grid with a longitude/latitude spacing of 0.05°, and the centres of the grid cells represent the possible centres of Voronoi poligons. We vary the number Voronoy poligons, Nv, from 3 to 50, generating 1000 tessellations for each Nv."*

L. 297: "β = 2/3 b": I think this should be "        = b. ln(10)", which is ~2.3 b.

*Yes thanks, it was an oversight. It is "= b. ln(10)" because we are taking into account the equation with magnitude and not seismic moment.*

**2.3 Combining fault and distributed sources**

L. 299-300: It would be better to describe this concept before the two source model components are described (see general remark).

*Thanks for the suggestion. We'll introduce this concept before in the manuscript.*

L. 307: Add some statement that this assumption is explained in more detail in the following paragraphs.

*Ok, at the end of the line 307 we'll add a sentence as:"… this assumption is explained in more detail further on."*

L. 338-340: Is this valid for all types of faults or only for dip-slip faults?

*It is valid only for dip-slip faults, and because we want be more general with this concept, we'll modify the lines 338-340 as: "Static stress changes produce areas of negative stress, also known as shadow zones, and positive stress zones".*

L. 360: Perhaps add that it is a linear function.

*Ok, we'll add it. We'll modify line 360 in: "we introduced a slip rate and a distance-weighting linear function.."*

L. 363: Write the equation more completely:

*We'll, thanks.*

However, there is still a problem with the second line, which does the opposite of what is intended (going to 1 as d increases): instead of $1/d$ it should be $d/d_{max}$…

*Thanks, you are right, we'll correct.*

L. 366-367: What is the rationale for varying $d_{max}$ in function of slip rate?

*We made a simple assumption, higher is the slip rate, higher is the deformation field and so higher is the value of $d_{max}$. We'll explain our rationale in the manuscript.*

L. 369-371: This is hard to understand. Maybe rephrase as "Because we considered two fault source models, one using only TGR MFDs and the other only CHR MFDs, and because the MFDs of distributed seismicity grid points in the vicinity of faults are modified with respect to the MFDs of these faults, we also obtain two different models of distributed seismicity."
In my opinion, it is also necessary at this point to show the summed MFDs of the different (sub)models, i.e. summed MFD of the TGR fault source model, of the CHR fault source model, of the TGR distributed source model, of the CHR distributed source model, and of the combined TGR and CHR source models.

*Thanks for the suggestion, we think that rephrasing as you suggested is clearer. As said in the previous comment, we'll add a new figure to show the MFD's of the different models.*

**3. Results and discussion**

L. 382: "designed under the traditional Poisson hypothesis": Rephrase

*We'll rephrase in: " To obtain PSH maps we assign the calculated expected seismicity rates, under Poisson hypothesis, to their pertinent geometries…"*

L. 386: "well-known": this is not the most relevant property for choosing OpenQuake. Perhaps widely used, open-source, tested, …?

*We'll remove "well-known" and add at line 387 before "The ground motion…" this sentence: "We used this software because it is an open source software developed recently by GEM with the purpose of providing seismic hazard and risk assessments. Moreover, it is widely recognized within the scientific community for its potential."*

L. 402: Explain more explicitly that the TGR and CHG fault source models are end members that are only used to explore the epistemic uncertainty, and that in the preferred fault source model a choice is made between the two MFD models for each fault.

*Thanks for your suggestion; we'll better explain our choices.*

L. 403-404: "Although both models have the same amount of seismic moment release": this has not been demonstrated.

*Here, we were discussing about the two fault source models. In this case the same amount of seismic moment release is an assumption that we made before to compute the MFD's, as before explained.*

L. 409-411: "The rates of earthquakes with magnitudes between 5.5 and approximately 6, …, are generally higher in the TGR model than in the CHG model": Please demonstrate by showing the summed MFDs.

*Will be shown in a new figure (now Figure S2 in the supplement).*

L. 443: "a magnitude lower or higher than the bell curve" → "a magnitude lower or higher than the magnitude range in the bell curve". See also my remark at lines 197-198: a higher magnitude should not be possible!

*We'll improve the manuscript, better describing our approach: see the answer in the general comments.*

L. 468-471: It has not been explained exactly how the TGR MFDs have been constructed. See my remark at lines 209-211.

*We'll add this information at line 209-211. See our reply at these lines.*

L. 505: Perhaps replace "TGR model" with a brief description like you do for the CHG model in the following line.

*Thanks for your comment, we agree. We'll add at line 505 a sentence as:" the Truncated Gutenberg-Richter model, where the maximum magnitude is the upper threshold and $M_w$ = 5.5 is the lower threshold for all faults…".*

**4. Conclusions**

L. 558-559: "pattern similar to that of the current national maps at the national scale, but some significant differences in hazard are present at the regional-to-local scale": this has not been discussed in the main text. It would be instructive to show both maps side by side and describe the comparison in some more detail in §3.

*See our reply at general comments and the new figure (now Figure S1 in the supplement). As suggested, the new figure we'll be included in the manuscript, at Chapter 3.*

L. 563-565: See my comment for lines 409-411. It would also be interesting to compare the summed MFDs to the observed MFD based on the full catalog, to see which of the two MFD models is closer to the observations in this particular magnitude range (M 5.5 to ~6.0).

*See our reply at general comments and the new figure (now Figure S2 in the supplement).*

**Figure captions**

Fig. 9 : Explain acronym "poe"

*In the caption we'll add this sentence: "The dashed lines represent the 2%, 10% and 81% probability of exceedance (poe) in 50 years."*

Fig. 12: How are the contributions of the component source models computed? The perfect symmetry between the contributions of the fault source model and the distributed source model gives me the impression that they do not correspond to the contributions one would obtain from a deaggregation.

*Yes, you're right it is not a deaggregation. It is the contribution of each source model in the total. For example, if the PGA value in a given point of the grid is: 0.15, 0.20 and 0.35 for the distributed, fault source and total respectively, the contribution will be 43% and 57% for the distributed and fault source respectively. Probably could be right to better explaining this in the manuscript, and so at line 482 we'll add a sentence as: "Note that the contributions are not given by deaggregation but are computed how the percentage of each source model in the PGA value of the total model."*

**Cited papers**

*Basili, R., G. Valensise, P. Vannoli, P. Burrato, U. Fracassi, S. Mariano, M. M. Tiberti, and E. Boschi. 2008. 'The Database of Individual Seismogenic Sources (DISS), version 3: Summarizing 20 years of research on Italy's earthquake geology', Tectonophysics, 453: 20-43.*

*Boncio, P., Brozzetti, F. and Lavecchia G. 2000. Architecture and seismotectonics of a regional Low-Angle Normal Fault zone in Central Italy. Tectonics, 19 (6), 1038-1055*

*Pace, B., F. Visini, and L. Peruzza. 2016. 'FiSH: MATLAB Tools to Turn Fault Data into Seismic- Hazard Models', Seismological Research Letters, 87: 374-86.*

**Review of manuscript NHESS-2017-41 "Integrating faults and past earthquakes into a probabilistic seismic hazard model for peninsular Italy" by Alessandro Valentini, Francesco Visini & Bruno Pace**
**by Laurentiu Danciu**
**Swiss Seismological Service**
**ETH Zurich**

**General Comments**

The manuscript provides a procedure to integrate active faults in a regional seismogenic source model for Italy. A database of active faults was compiled and fully parameterised for use together with observed seismicity (instrumental and historical) to forecast the spatial and temporal distribution of future seismicity. Earthquake recurrence models of the delineated active faults are model by two magnitude-frequency distributions: either a Characteristic Gaussian (CHG) or Truncated Gutenberg-Richter (TGR). Additionally, the seismicity off faults is described by a smoothed seismicity using a complete earthquake catalogue of the region. The two models are complementary not independent, thus the earthquake rates account for double-counting of earthquakes assigned to faults above specified threshold magnitude. Further, a novel weighting function to correct the earthquake rates in vicinity of fault sources is proposed and used. The resulting two seismic sources are eventually combined in a mixed source model representing the suitable activity rates in time and space. The authors conclude with a sensitivity analysis evaluating the impact of the two models of earthquake recurrence rates on the total seismic hazard.

The use of active faults in seismic hazard assessment has become extensive in the last decades due to efforts of data compilation and analysis. Active faults provides the information to extend the observational time of large magnitude earthquakes which often is not captured by the existing catalogues of observed seismicity. The current manuscript provides a step forward into this direction. The combination active faults and smoothed seismicity is not a novel procedure but rather state of practice. Overall, the manuscript is relatively well written, there are several misleading parts to be improved, highlighted in my detailed comments. The structure of the manuscript is consistent with the procedural steps and no major changes are required. The figures, tables and supplemental materials are clear and appropriated. There are some key references missing but this is not necessarily a criticism. The conclusions appear appropriate with the proposed procedure and analysed content. My comments follow the structure of the manuscript and summarised below:

1. First and foremost the authors should be clearly state that this is not an update of the seismic hazard model of Italy, and that the purpose of the study is to integrate the active faults in a hazard calculation. Moreover, the resulting seismogenic model presented in this study has limitations, such as the use only of shallow faults, but not the subduction and volcanic sources.

   *To clearly state that our model is not aimed to update seismic hazard model of Italy, we will add at line 70 the following statement: "In conclusion, even if the main purpose of this work is to integrate the active faults in a hazard calculation for the Italian territory, this work does not represent an official update of the seismic hazard model of the Italy".*
   *About the use of only shallow faults, but not the subduction and volcanic sources, we will more clear introduce this issue in the manuscript. In any case in this paper we want to focus on the*

*impact of the integration of faults and earthquakes data, without the assumption to be complete in terms of fault database, but on the contrary suggesting a way to integrate two incomplete database in the best way, without throwing data.*

2. A definition of active fault in the context of the study must be introduced. The literature distinguishes between active faults in geological time, i.e. Quaternary or Neocene, capable of future reactivation. Moreover, the slip rate assumptions must be discussed. It is well accepted that large variability are associated with the slip-rate values, and some portion of slip-rate can be aseismic. Extension of this discussion must be introduced in the context of this study.

*We agree that a definition of active fault in the context of the study is necessary. We will add at line 82 a phrase as: "For seismic hazard assessment an active fault is a structure that has evidence of activity in the late Quaternary (i.e. in the past 125 kyr), a demonstrable or potential capability of generating major earthquakes and capable of future reactivation (see Machette, 2000 for a discussion on terminology). The evidences of quaternary activity can be geomorphological and/or paleoseismological, when activation during instrumental seismic sequences and/or association to historical earthquakes are not available".*
*We will also extend discussion about slip rates assumptions for PSHA. In particular, we will more clear to state that we are assuming that slip-rates used are representative of seismic movements (no-aseismic factor). We think that investigating the impact of this assumption could be an issue of uncertainty-focused paper, for example by differentiating aseismic slip factor in respect to different tectonic contests.*

3. Further, the authors are aware of the 2013 European Seismic Hazard Model (ESHM13, Woesner et al 2015) developed within the SHARE Project. It might be worth discussing the two approaches side by side, as the ESHM13 is the first reference model to introduce active faults for Euro-Mediterranean Region.

*We prepared a new figure (Figure S1 in supplement) to compare our model, FSBG model proposed by SHARE and the Italian seismic hazard map MPS04, using the same GMPE's. A discussion about this comparison will be added in the "Results and Discussion" chapter.*

4. There are several procedural steps that are not well explained in the document, such as the estimation of the activity rates for faults. Albeit, the main focus of the procedure is to implement active faults to seismic hazard, the activity rates are yet described as input to the FiSH code and the segment seismic moment conservation. In my opinion this is not enough. The key elements and assumptions for computing the activity rates of active faults needs more attention, supported with discussions of the sensitivity of the input parameters, i.e. the effect of slip rates to earthquake recurrence rates.

*In order to explain more in detail the segment seismic moment conservation, we will modify part of the text by adding the following paragraph:*
*"… which divides the seismic moment that corresponds to Mmax by the moment rate given a slip rate:*

$$T_{mean} = \frac{1}{Char\_Rate} = \frac{10^{1.5M_{max}9.1}}{\mu VLW}$$

*where Tmean is the mean recurrence time in years, Char_Rate is the annual mean rate of occurrence, Mmax is the computed mean maximum magnitude, µ is the shear modulus, V is the average long-term slip rate, and L and W are the geometrical parameters of the fault, along-strike rupture length and down dip width respectively."*

*Moreover, to explain how magnitude frequency distribution of TGR is computed we will state that: " For MFD, the b-value is constant and equal to 1.0 for all faults, obtained by the interpolation of the earthquakes in the CPTI15 catalogue, as the events on the single sources are insufficient for statistics. However the a-values have been computed by Activity Rate FiSH code, balancing the total expected seismic moment rate with the seismic moment rate that was obtained by the pair Mmax and Tmean, evaluated by the fault geometry and the slip rate of each individual source (details in Pace et al., 2016)."*

5. The role of each magnitude frequency distribution (MFD) for each fault is not clear as described in the current version. One might expect a logic tree of the two MFDs. This aspect needs to be emphasised in the introduction.

   *Thanks for your suggestion. We'll clarify in the Introduction our choices, explaining that the TGR and CHG MFD are here used as end members, in order to explore the epistemic uncertainties, and we consider the Mixed model as a sort of an "expert judgment" model, useful for comparison analysis. As our model is not aimed to update seismic hazard model of Italy, we don't think we need to use a logic tree approach to produce a weighted model.*

6. Maximum magnitude assigned to each fault based on empirical magnitude scaling relationships do not account for uncertainties of the fault size (subsurface length or area). From the current version of the manuscript it is not evident the error associated to the fault size in the fault dataset.

   *In our work, the error associated to the fault size was not taken into account because there are no indications to quantify these errors from the published data used to obtain the active fault database. The error associated to the Mmax of the fault sources is only based on the errors of the used empirical relationships and observations.*

7. Also, one can argue that more recent magnitude scaling relationships can be used (e.g Leonard et al 2010) but for those used, the role of aleatory uncertainty must be mentioned and quantified herein. The authors should describe the procedure implemented in the FiSH code because not everyone has access to that manuscript.
8. *Five maximum magnitude values are described as being assigned to each fault. The way these five values are implemented in the final computational model is not clear. Are these values modelled in a logic tree?*

   *We will add a description of the procedure to estimate Mmax for faults after summarizing what has been done in Pace et al. (2016) FiSH code: "Because all the empirical relationships and observations are affected by uncertainties, a first code (MB) is designed to take these factors into*

*account and return a maximum magnitude value and a standard deviation. The uncertainties in the empirical scaling relationship are taken from the studies of Wells and Coppersmith (1994), Peruzza and Pace (2002) and Leonard (2010). Currently, the uncertainty in magnitude from seismic moment is fixed and set to 0.3, whereas the uncertainty in Mobs is defined by the catalogue. To combine the maximum magnitudes, MB draws a probability curve for each magnitude estimate by assuming a normal distribution. It is possible to define the number of standard deviations (σ) for truncating the normal distribution of magnitudes at both sides. MB successively sums the probability density curves and fits the summed curve to a normal distribution to obtain the mean of the maximum magnitude Mmax and its standard deviation. Therefore, Mmax represents an evaluation of the maximum rupture that is allowed by the fault geometry and the rheological properties".*

9.  A sensitivity analysis to the choice of the maximum magnitude may be necessary to explain the effect of maximum magnitude for the TGR. For the same slip rate increase of the maximum magnitude will result in a decrease of the recurrence of small events. This effect is due to the fact that the largest earthquake accounts for most of the seismic moment and this requires the subtraction of small events to maintain the seismic moment balance.

    *We agree with the topic here raised by the reviewer. Actually, the impact of uncertainties in Mmax and slip rate into PSHA is an important question, but we think it deserves a more extensive work to be exhaustively pointed out. We prepared a figure to show how varying these two parameters the seismicity rates can be distributed following a TGR model (Figure S3 in supplement). In our paper only the central values of the shown MFDs has been used. It is clear the final PSHA is substantially modified when Mmax and slip rate are changed, but, for the purpose of our work, this aspect is out of topic. We are exploring these (and other) aspects of fault-based approaches but, again, to be at least sufficiently analysed, they should be ingredients for a new work.*

10. In a general way, the characteristic model implies a recurrence rate estimated on large past large-magnitude earthquakes recognised from past geological record and the time interval between events can be measured. How many of the faults have a geological record long enough to characterise the recurrence of the large magnitude events? In the current version of the manuscript the historical events are linked to the faults, thus the long-term representation of the fault activity is questionable.

    *Thanks for your comment, we were not clear in explain how the mean recurrence times (Tmean) of the characteristic earthquake have been calculated. Similarly to TGR MFD we evaluated Mmax and Tmean by the fault geometry and the slip rate (not with the observed occurrences) of each individual source and we calculated the total expected seismic moment rate (eq. in the answer to comment 4). Then, we partitioned the total expected seismic moment rate in a range given by Mmax +- 1 standard deviation following a Gaussian bell distribution. We'll improve the manuscript to better explain this concept.*

11. Slip rates are averaged over successive geologically recognised earthquakes and prone to error in measurements, hence the uncertainties of the slip-rates needs to be quantified.

    *Uncertainties in slip rates estimates are given in the seismogenic sources database in the appendix. For our PSHA model we used the central value of the slip rate range given for each*

*fault. We are assuming that this value is representative of the average long term behaviour of the fault. Unfortunately, the state of the art of the knowledge of slip rates in Italy cannot allows to resolve a more detailed analysis of slip rate. However, varying slip rates in the currently range of uncertainty (as published in the papers cited in the appendix), we produced the figure S3 (in the supplement) to show the impact of these uncertainties on the activity rates.*

12. When combining active faults and background seismicity, it is mandatory a comparison of the seismic productivity (CHG and TRT) of the faults with the gridded seismicity in the vicinity of faults. Without such comparison it is difficult to assess the performance of the models.

    *Thanks for your suggestion. We attach in supplement a new figure (Figure S2) showing and comparing the summed MFD's of the fault source inputs (TGR, CHG, Mixed), the distributed source input, the final model (distributed + fault) and the CPTI15 catalogue. This new figure shows, in a sector of Italy where the faults are well defined, the behaviour of the activity rates as derived by our approach. The new figure we'll be included in the revised version of the manuscript.*

13. Generally, evaluating the performance of seismogenic sources based on seismic hazard estimates is not recommended. The hazard estimates based on active faults only is misleading, as the active faults are incomplete in space, and not treated as independent models. Thus the model performance may be evaluated at the level of seismicity rates comparison, not for hazard estimates.

    *Thanks for your comment, we agree it is important, in order to evaluate the performance of different seismic models for seismic hazard, a direct comparison of seismicity rates. For this reason we'll add in the manuscript the figure above described (Figure S2 in supplement). In any case we think it is interesting to show the impact of different seismogenic sources also in terms of seismic hazard maps.*

14. The authors should state clearly that a suitable seismogenic source model combines the active faults and the gridded seismicity as mixed model.

    *As also commented later, we agree that a model should include faults and distributed sources. We will clearly state that the mixed fault source is obtained by our judgment on the MFD assigned to each single fault, and that the mixed model combines this fault source input with the distributed sources input.*

**Section Specific Comments**

L50:51: "In Europe, a working group…" In Europe, within the SHARE project (Giardini et al 2010) has introduced the use of active faults at the region level for the first time. I am surprised that the authors do not refer in their study to the fault source models for Italy, the DISS (Database of Individual Seismogenic Sources). What are the main similarities and differences between the two dataset? The authors may consider adding a reference and a discuss the two datasets to avoid confusion.

*We mentioned SHARE project in our manuscript at line 58, and a new figure (S1 in supplement) compares the results. About the DISS, we will at line 84: "Although for the Italian territory there is already a database that contains the results of the investigations of the active tectonics during the past 20 years (Database of Individual Seismogenic Sources, DISS, http://diss.rm.ingv.it/diss/), made by three main categories of seismogenic sources: individual seismogenic sources, seismogenic areas, macroseismic sources, it does not work well to elaborate a PSHA model using individual seismogenic sources, as in this work. In fact, the DISS Authors (Basili et al., 2008) say that the individual seismogenic sources database cannot guarantee the completeness of the sources themselves and are not meant to comprise a complete input dataset for probabilistic assessment of seismic hazard. For this reason, we are not restricted to just use of the DISS, but trough a synthesis of published works over the last twenty years (see supplements for complete references) we defined a database as complete as possible, in terms of individual seismogenic sources, and parameters to have input dataset for PSHA."*

L63: 66 The uniform seismotectonic sources of the Italian hazard described by Stuchi et al (2011) are delineated considering the fault information where and when available. The more realistic pattern of ground motion due to faults it is questionable, because an area source delineated to describe a group of faults, it will produce a similar pattern with the individual faults. The major benefits of using the active faults is to extend the observational time to capture the recurrence of large magnitude events. The local pattern due to fault location might be controlled by other factors such as hanging wall, upper seismogenic depth, style of faulting. However, these effects are not evident if an inappropriate ground motion model is selected. Thus the seismic hazard pattern depends on both seismic source representation and ground motion models.

*We will modify from line 65: "…in order to obtain more detailed patterns of ground motion, extend the observational time to capture the recurrence of large magnitude events, and to improve the reliability of seismic hazard assessments." Moreover, we will add a new figure (Figure S1 in supplement) to compare the MPS04 and our PSHA model*

L72. The term models is misleading. A source model implies a complete source representation in space and time aimed at describing the seismogenic potential of the region. In the current context, the active faults are incomplete in space, they are not describing all the tectonics of the region - not volcanic, subduction or deep seismicity reported for the Italian territory. It has to be specified that these are individual seismic sources, but not independent models. The procedure proposed here is aiming at creating a "model" for an exercise of seismic hazard evaluation. Moreover, if the goal of the work is to provide a robust seismic hazard estimates, then the authors resolve the issues of model independence and completeness as well as to capture the epistemic uncertainties in the mixed source model.

*We agree with your comment, and so following your suggestion we'll remove the term "model" when we describe the fault source geometry, while we'll maintain the term "model" when we combine fault and distributed sources for the seismic hazard evaluations. In any case we want to highlight that the main aim of this work is how to combine fault and distributed sources in order to take into account and possibly overcome the incompleteness of the fault source database, without throwing data. We will add in the manuscript a phrase explaining our choices.*

L120: The time scale is a key aspect to evaluate the long-term representation of the seismic productivity of active faults. If a fault has moved in the recent geologically time , i.e Holocene, it might be considered as seismically active, if it moved in the far-off geologic time and has not moved again since then the fault might be judged to be an inactive fault. Hence, it might be of interest to specify the time scale and the definition of active faults on the present investigation. Yet, as mentioned before there is need to clarify the definition of fault activity or non activity.

*Please see our comment above on active fault definition (remark n. 2).*

L131:135. The slip rate values for some faults are very low. Values of 0.3 mm/year are extremely low and the movement on these faults could also takes place as creep. Is the aseismic factor adjusting the slip rates? Are these slip-rates supported by historical seismicity observations, geological investigations and /or paleoseismicity studies?

*These slip-rates are supported by historical seismicity observations, geological investigations and /or paleoseismicity studies as reported in the supplement files. Moreover we are assuming that the used slip-rates are representative of seismic movements (no-aseismic factor), as discussed above (remark n.2).*

L152: The name could be "Segmentation rules for delineating (or aggregating) fault sources"

*Thanks for the suggestion, we will modify it.*

L199: The role of aspect ratio must be discussed in greater extend than currently version. The extension along-strike dimensions of the faults seems to be constrained by this parameter.

*We will rephrase from line 199 as: "…by reducing the fault length if the aspect ratio (W/L) is smaller than indicated by the relation between aspect ratio and rupture length for observed earthquake ruptures as derived by Peruzza and Pace (2002)."*

L191: There are five Mmax values for each fault. How is the Mmax modelled in the hazard calculation?

*Please, see the comment to remark n. 8*

L202: Introduce and explain the "segment seismic moment conservation"? The key assumptions and the input parameters of the recurrence rates must be described. Characterisation of the active faults is a key aspect of this approach, thus it requires more description. As mentioned before, the effect of maximum magnitude must be discussed. In the case of seismic moment balance, for a constant slip rate, the recurrence rates of small events are decreasing with increased magnitude.

*We will introduce and explain better this issue. Please, see the replies to remarks n. 2, 4, and 9.*

L207:211: What is the rationale of the two MFDs? It is not evident why the two recurrence models are selected? In a general way, the characteristic earthquake is used to define an earthquake of a given magnitude and well identified recurrence time by geological evidences. The fault sources used here do not qualify for such model, for various reasons including the way they are constructed by linkage of various segments. A characteristic model will be appropriate for use on individual segment rather than a long composite fault. See discussions of Kagan (1993), that clearly states that the evidence of the characteristic earthquake hypothesis can be explained either by statistical bias or statistical artifact. Thus, it will be of great interest for the readers to specify the assumptions for the two MFDs.

*We agree that it is difficult to define an appropriate MFD (e.g. characteristic earthquake) for individual source using the available geological data, and important project as UCERF3 didn't solve the same doubts. In any case our fault source database have been developed to be representative of the maximum single earthquake rupture, and not long composite faults, by using restrictive segmentation rules described in chapter 2.1.2. Moreover, the two MFD are used as end members, in order to explore the epistemic uncertainties, and we consider the Mixed model as a sort of an "expert judgment" model, useful for comparison analysis.*

L278: the number of Voronoi polygons is not clear to me. There are 3 to 50 polygons across the entire region? Each polygon is tectonic dependent? Please clarify.

*We will modify the manuscript from the line 276: "… the Voronoi tessellation of space without tectonic dependency. The whole Italian territory has been divided into a grid with a longitude/latitude spacing of 0.05°, and the centres of the grid cells represent the possible centres of Voronoi polygons. We vary the number Voronoy polygons, $N_v$, from 3 to 50, generating 1000 tessellations for each $N_v$."*

L286: Who is parametrised the depth and the maximum magnitude for gridded seismicity? Are these parameters treated as aleatory or epistemic?

*The parameters have been taken from SHARE project, as written at lines 285-291. We did not explore the variability of these parameters.*

L382: For the purpose of an exercise one GMPE might have been justified. However, the focus of the study should be the comparison of the earthquake recurrence rates not the hazard estimates.

*We believe that the use of these GMPE's is correct, as they have been developed for Active Crust regions. Comparing model in terms of rates is for sure a valid approach. However, as the aim of our work is a PSH model, we believe that comparing different model (using the same GMPE's) can be useful. In any case we'll add in the manuscript a figure comparing the results also in terms of activity rates (Figure S2 in supplement).*

**Cited papers**

Basili, R., G. Valensise, P. Vannoli, P. Burrato, U. Fracassi, S. Mariano, M. M. Tiberti, and E. Boschi. 2008. 'The Database of Individual Seismogenic Sources (DISS), version 3: Summarizing 20 years of research on Italy's earthquake geology', *Tectonophysics*, 453: 20- 43.

Leonard, M. (2010). Earthquake fault scaling: Self-consistent relating of rupture length, width, average displacement, and moment release. *Bulletin of the Seismological Society of America*, *100*(5A), 1971-1988.

Machette, M.N., 2000, Active, capable, and potentially active faults; a paleoseismic perspective, *J. Geodyn.* **29**, 387–392.

Pace, B., F. Visini, and L. Peruzza. 2016. 'FiSH: MATLAB Tools to Turn Fault Data into Seismic-Hazard Models', Seismological Research Letters, 87: 374-86.

Peruzza, L., and B. Pace. 2002. 'Sensitivity analysis for seismic source characteristics to probabilistic seismic hazard assessment in central Apennines (Abruzzo area) '. Bollettino di Geofisica Teorica ed Applicata 43, 79–100.

Wells, D. L., and K. J. Coppersmith. 1994. 'New Empirical Relationships among Magnitude, Rupture Length, Rupture Width, Rupture Area, and Surface Displacement', Bulletin of the Seismological Society of America, 84: 974-1002.

Woessner, J., D. Laurentiu, D. Giardini, H. Crowley, F. Cotton, G. Grunthal, G. Valensise, R. Arvidsson, R. Basili, M. B. Demircioglu, S. Hiemer, C. Meletti, R. W. Musson, A. N. Rovida, K. Sesetyan, M. Stucchi, and SHARE Consortium. 2015. 'The 2013 European Seismic Hazard Model: key components and results', Bulletin of Earthquake Engineering, 13: 3553-96

**Integrating faults and past earthquakes into a probabilistic seismic hazard**

**model for peninsular Italy**

Alessandro Valentini[1], Francesco Visini[2] and Bruno Pace[1]

[1] DiSPUTer, Università degli Studi "Gabriele d'Annunzio", Chieti, Italy

[2] Istituto Nazionale di Geofisica e Vulcanologia, L'Aquila, Italy

**Abstract**

[revised manuscript text omitted]

*2.1.1 Slip rates*

Slip rates control fault-based seismic hazards (Main, 1996, Roberts et al., 2004; Bull
et al., 2006; Visini and Pace, 2014) and reflect the velocities of the mechanisms that
operate during continental deformation (e.g., Cowie et al., 2005). Moreover, long-
term observations of faults in various tectonic contexts have shown that slip rates
vary in space and time (e.g., Bull et al., 2006; Nicol et al., 2006, 2010, McClymont et
al., 2009; Gunderson et al., 2013; Benedetti et al., 2013, D'Amato et al., 2016), and
numerical simulations (e.g., Robinson et al., 2009; Cowie et al., 2012; Visini and
Pace, 2014) suggest that variability mainly occurs in response to interactions
between adjacent faults. Therefore, understanding the temporal variability in fault slip
rates is a key point in understanding the earthquake recurrence rates and their
variability.

In this work, we used the mean of the minimum and maximum slip rate values listed
in Table 1 and assumed that it is representative of the long-term behaviour (over the
past 15 ky in the Apennines). These values were derived from approximately 65
available neotectonics, palaeoseismology and seismotectonics papers (see the
supplemental files). To evaluate the long-term slip rate, which is representative of the
average slip behaviour, and its variability over time, we used slip rates determined in
different ways and at different time scales (e.g., at the decadal scale based on
geodetic data or at longer scales based on the displacement of Holocene or Plio-
Pleistocene horizons). Because a direct comparison of slip rates over different time
intervals obtained by different methods may be misleading (Nicol et al., 2009), we
cannot exclude the possibility that epistemic uncertainties could affect the original

Alessandro 28/8/y 14:32
**Commenta [7]:** After Detailed Comments L92 by RC1.

Authors 28/8/y 12:12

Authors 28/8/y 12:12

Authors 28/8/y 12:12

Authors 28/8/y 12:12

Authors 28/8/y 12:12

Authors 28/8/y 12:12

Authors 28/8/y 12:12

Authors 28/8/y 12:12

Authors 28/8/y 12:12

Authors 28/8/y 12:12

Authors 28/8/y 12:12

Authors 28/8/y 12:12

Authors 28/8/y 12:12

Authors 28/8/y 12:12

Authors 28/8/y 12:12

Authors 28/8/y 12:12

Authors 28/8/y 12:12

Alessandro 28/8/y 14:35
**Commenta [8]:** After: Detailed Comment L112-124 by RC1 and General Comments number 11 by RC2.

[revised manuscript text omitted]

Although incorrect to consider MObs a possible $M_{max}$ value and treat it the same as
other estimations, in some cases, it was useful to constrain the seismogenic
potentials of individual seismogenic sources. As an example, for the *Irpinia Fault* (id
51 in Tables 1 and 2), the characteristics of the 1980 earthquake (Mw~6.9) can be
used to evaluate $M_{max}$ via comparison with the $M_{max}$ derived from scaling
relationships. In such cases, we (i) calculated the maximum expected magnitude ($M_{max1}$) and the relative uncertainties using only the scaling relationships and (ii) compared the maximum of observed magnitudes of the earthquakes potentially associated with the fault. If MObs was within the range of $M_{max}$ ± 1 standard deviation, we considered the value and recalculated a new $M_{max}$ ($M_{max2}$) with a new uncertainty. If MObs was larger than $M_{max1}$, we reviewed the fault geometry and/or the earthquake-source association.

Because all the empirical relationships, as well as observed historical and recent magnitudes of earthquakes, are affected by uncertainties, the *MomentBalance* (MB) portion of the FiSH code (Pace et al., 2016) was used to account for these uncertainties. MB computes a probability density function for each magnitude derived from empirical relationships or observations and summarizes the results as a maximum magnitude value with a standard deviation. The uncertainties in the empirical scaling relationship are taken from the studies of Wells and Coppersmith (1994), Peruzza and Pace (2002) and Leonard (2010). Currently, the uncertainty in magnitude associated with the seismic moment is fixed and set to 0.3, whereas the catalogue defines the uncertainty in MObs. Moreover, to combine the evaluated maximum magnitudes, MB creates a probability curve for each magnitude by assuming a normal distribution (Fig. 2). We assumed an untruncated normal distribution of magnitudes at both sides. MB successively sums the probability density curves and fits the summed curve to a normal distribution to obtain the mean of the maximum magnitude $M_{max}$ and its standard deviation.

Thus, a unique $M_{max}$ with a standard deviation is computed for each source, and this value represents the maximum rupture that is allowed by the fault geometry and the rheological properties.

Finally, to obtain the mean recurrence time of $M_{max}$ (i.e., $T_{mean}$), we use the criterion of "segment seismic moment conservation" proposed by Field et al. (1999). This criterion divides the seismic moment that corresponds to $M_{max}$ by the moment rate for given a slip rate:

$$T_{mean} = \frac{1}{Char\_Rate} = \frac{10^{1.5\,M_{max}\,9.1}}{\mu VLW} \quad (1)$$

Alessandro 28/8/y 14:41
**Commenta [11]:** After Detailed Comment L197-198 by RC1.

Authors 28/8/y 12:12

Authors 28/8/y 12:12

Authors 28/8/y 12:12

Authors 28/8/y 12:12

Authors 28/8/y 12:12

Alessandro 28/8/y 14:45
**Commenta [12]:** After General Comments number 7 and 8 and Section Specific Comments L191 by RC2.

Authors 28/8/y 12:12

Authors 28/8/y 12:12
**Spostato (inserimento) [1]**

where $T_{mean}$ is the mean recurrence time in years, Char_Rate is the annual mean rate of occurrence, $M_{max}$ is the computed mean maximum magnitude, μ is the shear modulus, V is the average long-term slip rate, and L and W are geometrical parameters of the fault along-strike rupture length and downdip width, respectively. This approach was used for both MFDs in this study, and, in particular, we evaluated $M_{max}$ and $T_{mean}$ based on the fault geometry and the slip rate of each individual source. Additionally, we calculated the total expected seismic moment rate using equation 1. Then, we partitioned the total expected seismic moment rate based on a range given by $M_{max}$ ± 1 standard deviation following a Gaussian distribution. After the fault source is entered as input, the seismic moment rate is calculated, $M_{max}$ (Fig. 2b) and $T_{mean}$ are defined for each source, we computed the MFDs of expected seismicity. For each fault source, we use two "end-member" MFD models: (i) a *Characteristic Gaussian* (*CHG*) model, a symmetric Gaussian curve (applied to the incremental MFD values) centred on the $M_{max}$ value of each fault with a range of magnitudes equal to 1-sigma, and (ii) a *Truncated Gutenberg-Richter* (*TGR, Ordaz, 1999; Kagan, 2002*) model, with $M_{max}$ as the upper threshold and $M_w$ = 5.5 as the minimum threshold for all sources. The b-values are constant and equal to 1.0 for all faults, and they are obtained by the interpolation of earthquake data from the CPTI15 catalogue, 
[revised manuscript text omitted]

Authors 28/8/y 12:12

Alessandro 28/8/y 14:50
Commenta [17]: After: Main Comment 2 and 3 by RC1 and General Comment number 3 by RC2.

Authors 28/8/y 12:12

Authors 28/8/y 12:12

Authors 28/8/y 12:12

Authors 28/8/y 12:12

Authors 28/8/y 12:12

Authors 28/8/y 12:12

Authors 28/8/y 12:12

Authors 28/8/y 12:12

Authors 28/8/y 12:12

Authors 28/8/y 12:12

Authors 28/8/y 12:12

Authors 28/8/y 12:12

Authors 28/8/y 12:12

Authors 28/8/y 12:12

Authors 28/8/y 12:12

Authors 28/8/y 12:12

Authors 28/8/y 12:12

Authors 28/8/y 12:12

Authors 28/8/y 12:12

Authors 28/8/y 12:12

Authors 28/8/y 12:12

Authors 28/8/y 12:12

Authors 28/8/y 12:12

[revised manuscript text omitted]

Unknown
Alessandro 28/8/y 15:00

This work (Mixed Model)

Alessandro 28/8/y 15:02
**Commenta [19]:** After Main Comment number 2 and 3 and Detailed Comment L30 and L558-559 by RC1; and General Comment number 3 and and Section Specific Comment L63-68 by RC2.

| ID | Fault Sources | L (km) | Dip (°) | Upper (km) | Lower (km) | SR$_{min}$ (mm/yr) | SR$_{max}$ (mm/yr) |
|---|---|---|---|---|---|---|---|
| 1 | Lunigiana | 43.8 | 40 | 0 | 5 | 0.28 | 0.7 |
| 2 | North Apuane Transfer | 25.5 | 45 | 0 | 7 | 0.33 | 0.83 |
| 3 | Garfagnana | 26.9 | 30 | 0 | 4.5 | 0.35 | 0.57 |
| 4 | Garfagnana Transfer | 47.1 | 90 | 2 | 7 | 0.33 | 0.83 |
| 5 | Mugello | 21.0 | 40 | 0 | 7 | 0.33 | 0.83 |
| 6 | Ronta | 19.3 | 65 | 0 | 7 | 0.17 | 0.5 |
| 7 | Poppi | 17.1 | 40 | 0 | 4.5 | 0.33 | 0.83 |
| 8 | Città di Castello | 22.9 | 40 | 0 | 3 | 0.25 | 1.2 |
| 9 | M.S.M. Tiberina | 10.5 | 40 | 0 | 2.5 | 0.25 | 0.75 |
| 10 | Gubbio | 23.6 | 50 | 0 | 6 | 0.4 | 1.2 |
| 11 | Colfiorito System | 45.9 | 50 | 0 | 8 | 0.25 | 0.9 |
| 12 | Umbra Valley | 51.1 | 55 | 0 | 4.5 | 0.4 | 1.2 |
| 13 | Vettore-Bove | 35.4 | 50 | 0 | 15 | 0.2 | 1.05 |
| 14 | Nottoria-Preci | 29.0 | 50 | 0 | 12 | 0.2 | 1 |
| 15 | Cascia-Cittareale | 24.3 | 50 | 0 | 13.5 | 0.2 | 1 |
| 16 | Leonessa | 14.9 | 55 | 0 | 12 | 0.1 | 0.7 |
| 17 | Rieti | 17.6 | 50 | 0 | 10 | 0.25 | 0.6 |
| 18 | Fucino | 82.3 | 50 | 0 | 13 | 0.3 | 1.6 |
| 19 | Sella di Corno | 23.1 | 60 | 0 | 13 | 0.35 | 0.7 |
| 20 | Pizzoli-Pettino | 21.3 | 50 | 0 | 14 | 0.3 | 1 |
| 21 | Montereale | 15.1 | 50 | 0 | 14 | 0.25 | 0.9 |
| 22 | Gorzano | 28.1 | 50 | 0 | 15 | 0.2 | 1 |
| 23 | Gran Sasso | 28.4 | 50 | 0 | 15 | 0.35 | 1.2 |
| 24 | Paganica | 23.7 | 50 | 0 | 14 | 0.4 | 0.9 |
| 25 | Middle Aternum Valley | 29.1 | 50 | 0 | 14 | 0.15 | 0.45 |
| 26 | Campo Felice-Ovindoli | 26.2 | 50 | 0 | 13 | 0.2 | 1.6 |
| 27 | Carsoli | 20.5 | 50 | 0 | 11 | 0.35 | 0.6 |
| 28 | Liri | 42.5 | 50 | 0 | 11 | 0.3 | 1.26 |
| 29 | Sora | 20.4 | 50 | 0 | 11 | 0.15 | 0.45 |
| 30 | Marsicano | 20.0 | 50 | 0 | 13 | 0.25 | 1.2 |
| 31 | Sulmona | 22.6 | 50 | 0 | 15 | 0.6 | 1.35 |
| 32 | Maiella | 21.4 | 55 | 0 | 15 | 0.7 | 1.6 |
| 33 | Aremogna C.Miglia | 13.1 | 50 | 0 | 15 | 0.1 | 0.6 |
| 34 | Barrea | 17.1 | 55 | 0 | 13 | 0.2 | 1 |
| 35 | Cassino | 24.6 | 60 | 0 | 11 | 0.25 | 0.5 |
| 36 | Ailano-Piedimonte | 17.6 | 60 | 0 | 12 | 0.15 | 0.35 |
| 37 | Matese | 48.3 | 60 | 0 | 13 | 0.2 | 1.9 |
| 38 | Bojano | 35.5 | 55 | 0 | 13 | 0.2 | 0.9 |
| 39 | Frosolone | 36.1 | 70 | 11 | 25 | 0.35 | 0.93 |
| 40 | Ripabottoni-San Severo | 68.3 | 85 | 6 | 25 | 0.1 | 0.5 |
| 41 | Mattinata | 42.3 | 85 | 0 | 25 | 0.7 | 1 |
| 42 | Castelluccio dei Sauri | 93.2 | 90 | 11 | 22 | 0.1 | 0.5 |
| 43 | Ariano Irpino | 30.1 | 70 | 11 | 25 | 0.35 | 0.93 |
| 44 | Tammaro | 25.0 | 60 | 0 | 13 | 0.35 | 0.93 |
| 45 | Benevento | 25.0 | 55 | 0 | 10 | 0.35 | 0.93 |
| 46 | Volturno | 15.7 | 60 | 1 | 13 | 0.23 | 0.57 |
| 47 | Avella | 20.5 | 55 | 1 | 13 | 0.2 | 0.7 |
| 48 | Ufita-Bisaccia | 59.0 | 64 | 1.5 | 15 | 0.35 | 0.93 |
| 49 | Melfi | 17.2 | 80 | 12 | 22 | 0.1 | 0.5 |
| 50 | Irpinia Antithetic | 15.0 | 60 | 0 | 11 | 0.2 | 0.53 |

Authors 28/8/y 12:12

| ID | Name | L | Dip | Upper | Lower | SRmin | SRmax |
|---|---|---|---|---|---|---|---|
| 51 | Irpinia | 39.7 | 65 | 0 | 14 | 0.3 | 2.5 |
| 52 | Volturara | 23.7 | 60 | 1 | 13 | 0.2 | 0.35 |
| 53 | Alburni | 20.4 | 60 | 0 | 8 | 0.35 | 0.7 |
| 54 | Caggiano-Diano Valley | 46.0 | 60 | 0 | 12 | 0.35 | 1.15 |
| 55 | Pergola-Maddalena | 50.6 | 60 | 0 | 12 | 0.20 | 0.93 |
| 56 | Agri | 34.9 | 50 | 5 | 15 | 0.8 | 1.3 |
| 57 | Potenza | 17.8 | 90 | 15 | 21 | 0.1 | 0.5 |
| 58 | Palagianello | 73.3 | 90 | 13 | 22 | 0.1 | 0.5 |
| 59 | Monte Alpi | 10.9 | 60 | 0 | 13 | 0.35 | 0.9 |
| 60 | Maratea | 21.6 | 60 | 0 | 13 | 0.46 | 0.7 |
| 61 | Mercure | 25.8 | 60 | 0 | 13 | 0.2 | 0.6 |
| 62 | Pollino | 23.8 | 60 | 0 | 15 | 0.22 | 0.58 |
| 63 | Castrovillari | 10.3 | 60 | 0 | 15 | 0.2 | 1.15 |
| 64 | Rossano | 14.9 | 60 | 0 | 22 | 0.5 | 0.6 |
| 65 | Crati West | 49.7 | 45 | 0 | 15 | 0.84 | 1.4 |
| 66 | Crati East | 18.4 | 60 | 0 | 8 | 0.75 | 1.45 |
| 67 | Lakes | 43.6 | 60 | 0 | 22 | 0.75 | 1.45 |
| 68 | Fuscalto | 21.1 | 60 | 2 | 22 | 0.75 | 1.45 |
| 69 | Piano Lago-Decollatura | 25.0 | 60 | 1 | 15 | 0.23 | 0.57 |
| 70 | Catanzaro North | 29.5 | 80 | 3 | 20 | 0.75 | 1.45 |
| 71 | Catanzaro South | 21.3 | 80 | 3 | 20 | 0.75 | 1.45 |
| 72 | Serre | 31.6 | 60 | 0 | 15 | 0.7 | 1.15 |
| 73 | Vibo | 23.0 | 80 | 0 | 15 | 0.75 | 1.45 |
| 74 | Sant'Eufemia Gulf | 24.8 | 40 | 1 | 11 | 0.11 | 0.3 |
| 75 | Capo Vaticano | 13.7 | 60 | 0 | 8 | 0.75 | 1.45 |
| 76 | Coccorino | 13.3 | 70 | 3 | 11 | 0.75 | 1.45 |
| 77 | Scilla | 29.7 | 60 | 0 | 13 | 0.8 | 1.5 |
| 78 | Sant'Eufemia | 19.2 | 60 | 0 | 13 | 0.75 | 1.45 |
| 79 | Cittanova-Armo | 63.8 | 60 | 0 | 13 | 0.45 | 1.45 |
| 80 | Reggio Calabria | 27.2 | 60 | 0 | 13 | 0.7 | 2 |
| 81 | Taormina | 38.7 | 30 | 3 | 13 | 0.9 | 2.6 |
| 82 | Acireale | 39.4 | 60 | 0 | 15 | 1.15 | 2.3 |
| 83 | Western Ionian | 50.1 | 65 | 0 | 15 | 0.75 | 1.45 |
| 84 | Eastern Ionian | 39.3 | 65 | 0 | 15 | 0.75 | 1.45 |
| 85 | Climiti | 15.7 | 60 | 0 | 15 | 0.75 | 1.45 |
| 86 | Avola | 46.9 | 60 | 0 | 16 | 0.8 | 1.6 |

Table 1 Geometric Parameters of the Fault Sources. L, along-strike length; Dip,
inclination angle of the fault plane; Upper and Lower, the thickness bounds of the
local seismogenic layer; SRmin and SRmax, the slip rates assigned to the sources
using the references available (see the supplemental files); and *ID,* the fault number
identifier.

Authors 28/8/y 12:12
... [117]
Authors 28/8/y 12:12
... [119]

| ID | Fault Sources | Historical Earthquakes | | | | | Instrumental Earthquakes | |
|---|---|---|---|---|---|---|---|---|
| | | yyyy/mm/dd | $I_{Max}$ | $I_0$ | $M_w$ | sD | yyyy/mm/dd | $M_w$ |
| 1 | Lunigiana | 1481/05/07 | VIII | VIII | 5.6 | 0.4 | | |
| | | 1834/02/14 | IX | IX | 6.0 | 0.1 | | |
| 2 | North Apuane Transfer | 1837/04/11 | X | IX | 5.9 | 0.1 | | |
| 3 | Garfagnana | 1740/03/06 | VIII | VIII | 5.6 | 0.2 | | |
| | | 1920/09/07 | X | X | 6.5 | 0.1 | | |
| 4 | Garfagnana Transfer | | | | | | | |
| 5 | Mugello | 1542/06/13 | IX | IX | 6.0 | 0.2 | | |
| | | 1919/06/29 | X | X | 6.4 | 0.1 | | |
| 6 | Ronta | | | | | | | |
| 7 | Poppi | | | | | | | |
| 8 | Città di Castello | 1269 | | | 5.7 | | | |
| | | 1389/10/18 | IX | IX | 6 | 0.5 | | |
| | | 1458/04/26 | VIII-IX | VIII-IX | 5.8 | 0.5 | | |
| | | 1789/09/30 | IX | IX | 5.9 | 0.1 | | |
| 9 | M.S.M. Tiberina | 1352/12/25 | IX | IX | 6.3 | 0.2 | | |
| | | 1917/04/26 | IX-X | IX-X | 6.0 | 0.1 | | |
| 10 | Gubbio | | | | | | 1984/04/29 | 5.6 |
| 11 | Colfiorito System | 1279/04/30 | X | IX | 6.2 | 0.2 | 1997/09/26 | 5.7 |
| | | 1747/04/17 | IX | IX | 6.1 | 0.1 | 1997/09/26 | 6 |
| | | 1751/07/27 | X | X | 6.4 | 0.1 | | |
| 12 | Umbra Valley | 1277 | | VIII | 5.6 | 0.5 | | |
| | | 1832/01/13 | X | X | 6.4 | 0.1 | | |
| | | 1854/02/12 | VIII | VIII | 5.6 | 0.3 | | |
| 13 | Vettore-Bove | | | | | | 2016/10/30 | 6.5 |
| 14 | Nottoria-Preci | 1328/12/01 | X | X | 6.5 | 0.3 | 1979/09/19 | 5.8 |
| | | 1703/01/14 | XI | XI | 6.9 | 0.1 | | |
| | | 1719/06/27 | VIII | VIII | 5.6 | 0.3 | | |
| | | 1730/05/12 | IX | IX | 6.0 | 0.1 | | |
| | | 1859/08/22 | VIII-IX | VIII-IX | 5.7 | 0.3 | | |
| | | 1879/02/23 | VIII | VIII | 5.6 | 0.3 | | |
| 15 | Cascia-Cittareale | 1599/11/06 | IX | IX | 6.1 | 0.2 | | |
| | | 1916/11/16 | VIII | VIII | 5.5 | 0.1 | | |
| 16 | Leonessa | | | | | | | |
| 17 | Rieti | 1298/12/01 | X | IX-X | 6.3 | 0.5 | | |
| | | 1785/10/09 | VIII-IX | VIII-IX | 5.8 | 0.2 | | |
| 18 | Fucino | 1349/09/09 | IX | IX | 6.3 | 0.1 | | |
| | | 1904/02/24 | IX | VIII-IX | 5.7 | 0.1 | | |
| | | 1915/01/13 | XI | XI | 7 | 0.1 | | |
| 19 | Sella di Corno | | | | | | | |
| 20 | Pizzoli-Pettino | 1703/02/02 | X | X | 6.7 | 0.1 | | |
| 21 | Montereale | | | | | | | |
| 22 | Gorzano | 1639/10/07 | X | IX-X | 6.2 | 0.2 | | |
| | | 1646/04/28 | IX | IX | 5.9 | 0.4 | | |
| 23 | Gran Sasso | | | | | | | |
| 24 | Paganica | 1315/12/03 | VIII | VIII | 5.6 | 0.5 | 2009/06/04 | 6.3 |
| | | 1461/11/27 | X | X | 6.5 | 0.5 | | |
| 25 | Middle Aternum Valley | | | | | | | |
| 26 | Campo Felice-Ovindoli | | | | | | | |
| 27 | Carsoli | | | | | | | |
| 28 | Liri | | | | | | | |
| 29 | Sora | 1654/07/24 | X | IX-X | 6.3 | 0.2 | | |
| 30 | Marsicano | | | | | | | |
| 31 | Sulmona | | | | | | | |
| 32 | Maiella | | | | | | | |
| 33 | Aremogna C.Miglia | | | | | | | |
| 34 | Barrea | | | | | | 1984/05/07 | 5.9 |
| 35 | Cassino | | | | | | | |
| 36 | Ailano-Piedimonte | | | | | | | |
| 37 | Matese | 1349/09/09 | X-XI | X | 6.8 | 0.2 | | |


| # | Name | Date | | | | | Date 2 | |
|---|------|------|---|---|---|---|--------|---|
| 38 | Bojano | 1805/07/26 | X | X | 6.7 | 0.1 | | |
| 39 | Frosolone | 1456/12/05 | XI | XI | 7 | 0.1 | | |
| 40 | Ripabottoni-San Severo | 1627/07/30 | X | X | 6.7 | 0.1 | 2002/10/31 | 5.7 |
| | | 1647/05/05 | VII-VIII | VII-VIII | 5.7 | 0.4 | | |
| | | 1657/01/29 | IX-X | VIII-IX | 6.0 | 0.2 | | |
| 41 | Mattinata | 1875/12/06 | VIII | VIII | 5.9 | 0.1 | | |
| | | 1889/12/08 | VII | VII | 5.5 | 0.1 | | |
| | | 1948/08/18 | VII-VIII | VII-VIII | 5.6 | 0.1 | | |
| 42 | Castelluccio dei Sauri | 1361/07/17 | X | IX | 6 | 0.5 | | |
| | | 1560/05/11 | VIII | VIII | 5.7 | 0.5 | | |
| | | 1731/03/20 | IX | IX | 6.3 | 0.1 | | |
| 43 | Ariano Irpino | 1456/12/05 | | | 6.9 | 0.1 | | |
| | | 1962/08/21 | IX | IX | 6.2 | 0.1 | | |
| 44 | Tammaro | 1688/06/05 | XI | XI | 7 | 0.1 | | |
| 45 | Benevento | | | | | | | |
| 46 | Volturno | | | | | | | |
| 47 | Avella | 1499/12/05 | VIII | VIII | 5.6 | 0.5 | | |
| 48 | Ufita-Bisaccia | 1732/11/29 | X-XI | X-XI | 6.8 | 0.1 | | |
| | | 1930/07/23 | X | X | 6.7 | 0.1 | | |
| 49 | Melfi | 1851/08/14 | X | X | 6.5 | 0.1 | | |
| 50 | Irpinia Antithetic | | | | | | | |
| 51 | Irpinia | 1466/01/15 | VIII-IX | VIII-IX | 6.0 | 0.2 | 1980/11/23 | 6.8 |
| | | 1692/03/04 | VIII | VIII | 5.9 | 0.4 | | |
| | | 1694/09/08 | X | X | 6.7 | 0.1 | | |
| | | 1853/04/09 | IX | VIII | 5.6 | 0.2 | | |
| 52 | Volturara | | | | | | | |
| 53 | Alburni | | | | | | | |
| 54 | Caggiano-Diano Valley | 1561/07/31 | IX-X | X | 6.3 | 0.1 | | |
| 55 | Pergola-Maddalena | 1857/12/16 | | | 6.5 | | | |
| | | 1857/12/16 | | | 6.3 | | | |
| 56 | Agri | | | | | | | |
| 57 | Potenza | 1273/12/18 | VIII-IX | VIII-IX | 5.8 | 0.5 | 1990/05/05 | 5.8 |
| 58 | Palagianello | | | | | | | |
| 59 | Monte Alpi | | | | | | | |
| 60 | Maratea | | | | | | | |
| 61 | Mercure | 1708/01/26 | VIII-IX | VIII | 5.6 | 0.6 | 1998/09/09 | 5.5 |
| 62 | Pollino | | | | | | | |
| 63 | Castrovillari | | | | | | | |
| 64 | Rossano | 1836/04/25 | X | IX | 6.2 | 0.2 | | |

| | | | $I_{Max}$ | $I_0$ | $M_w$ | sD |
|---|---|---|---|---|---|---|
| 65 | Crati West | 1184/05/24 | IX | IX | 6.8 | 0.3 |
| | | 1870/10/04 | X | IX-X | 6.2 | 0.1 |
| | | 1886/03/06 | VII-VIII | VII-VIII | 5.6 | 0.3 |
| 66 | Crati East | 1767/07/14 | VIII-IX | VIII-IX | 5.9 | 0.2 |
| | | 1835/10/12 | X | IX | 5.9 | 0.3 |
| 67 | Lakes | 1638/06/08 | X | X | 6.8 | 0.1 |
| 68 | Fuscalto | 1832/03/08 | X | X | 6.6 | 0.1 |
| 69 | Piano Lago-Decollatura | | | | | |
| 70 | Catanzaro North | 1638/03/27 | | | 6.6 | |
| 71 | Catanzaro South | 1626/04/04 | X | IX | 6.1 | 0.4 |
| 72 | Serre | 1659/11/05 | X | X | 6.6 | 0.1 |
| | | 1743/12/07 | IX-X | VIII-IX | 5.9 | 0.2 |
| | | 1783/02/07 | X-XI | X-XI | 6.7 | 0.1 |
| | | 1791/10/13 | IX | IX | 6.1 | 0.1 |
| 73 | Vibo | | | | | |
| 74 | Sant'Eufemia Gulf | 1905/09/08 | X-XI | X-XI | 7 | 0.1 |
| 75 | Capo Vaticano | | | | | |
| 76 | Coccorino | 1928/03/07 | VIII | VII-VIII | 5.9 | 0.1 |
| 77 | Scilla | | | | | |
| 78 | Sant'Eufemia | 1894/11/16 | IX | IX | 6.1 | 0.1 |
| 79 | Cittanova-Armo | 1509/02/25 | IX | VIII | 5.6 | 0.4 |
| | | 1783/02/05 | XI | XI | 7.1 | 0.1 |
| 80 | Reggio Calabria | | | | | |
| 81 | Taormina | 1908/12/28 | XI | XI | 7.1 | 0.2 |
| 82 | Acireale | 1818/02/20 | IX-X | IX-X | 6.3 | 0.1 |
| 83 | Western Ionian | 1693/01/11 | XI | XI | 7.3 | 0.1 |
| 84 | Eastern Ionian | | | | | |
| 85 | Climiti | | | | | |
| 86 | Avola | | | | | |

Table 2 Earthquake-Source Association Adopted for Fault Sources. $I_{Max}$, maximum
intensity; $I_0$, epicentral intensity; $M_w$, moment magnitude; and sD, standard deviation
of the moment magnitude. For references, see the supplemental files.